# Meningeal lymphatic dysfunction exacerbates traumatic brain injury pathogenesis

Ashley C. Bolte [1,2,3,4], Arun B. Dutta [3,5], Mariah E. Hurt[1], Igor Smirnov[1], Michael A. Kovacs[1,2,3,4], Celia A. McKee[1], Hannah E. Ennerfelt[1,6], Daniel Shapiro[1], Bao H. Nguyen[5], Elizabeth L. Frost[1], Catherine R. Lammert[1,6], Jonathan Kipnis [1,6] & John R. Lukens [1,3,4,6 ✉]

Traumatic brain injury (TBI) is a leading global cause of death and disability. Here we demonstrate in an experimental mouse model of TBI that mild forms of brain trauma cause severe deficits in meningeal lymphatic drainage that begin within hours and last out to at least one month post-injury. To investigate a mechanism underlying impaired lymphatic function in TBI, we examined how increased intracranial pressure (ICP) influences the meningeal lymphatics. We demonstrate that increased ICP can contribute to meningeal lymphatic dysfunction. Moreover, we show that pre-existing lymphatic dysfunction before TBI leads to increased neuroinflammation and negative cognitive outcomes. Finally, we report that rejuvenation of meningeal lymphatic drainage function in aged mice can ameliorate TBI-induced gliosis. These findings provide insights into both the causes and consequences of meningeal lymphatic dysfunction in TBI and suggest that therapeutics targeting the meningeal lymphatic system may offer strategies to treat TBI.

[1] Center for Brain Immunology and Glia (BIG), Department of Neuroscience, University of Virginia, Charlottesville, VA 22908, USA. [2] Department of Microbiology, Immunology and Cancer Biology, University of Virginia, Charlottesville, VA 22908, USA. [3] Medical Scientist Training Program, University of Virginia, Charlottesville, VA 22908, USA. [4] Immunology Training Program, University of Virginia, Charlottesville, VA 22908, USA. [5] Department of Biochemistry and Molecular Genetics, University of Virginia, Charlottesville, VA 22908, USA. [6] Neuroscience Graduate Program, University of Virginia, Charlottesville, VA 22908, USA. ✉email: Jrl7n@virginia.edu

Traumatic brain injury (TBI) affects millions of people worldwide each year[1]. TBI can cause debilitating impairments in motor function, cognition, sensory function, and mental health. In addition, mounting evidence indicates that having a history of TBI markedly increases the risk of developing numerous other neurological disorders later in life including chronic traumatic encephalopathy (CTE), Alzheimer's disease, anxiety, depression, and amyotrophic lateral sclerosis[2–6]. Despite being a prevalent and pressing global medical issue, the pathoetiology of TBI remains incompletely understood and improved treatment options are desperately needed.

Tissue damage and cellular stress resulting from TBI incites activation of the immune system, which is intended to aid in the disposal of neurotoxic material and coordinate tissue repair[7,8]. While immune responses can play beneficial roles in TBI, unchecked and/or chronic immune activation following brain trauma can lead to secondary tissue damage, brain atrophy, and eventual neurological dysfunction[7–12]. Notably, the inability to properly dispose of danger/damage-associated molecular patterns (DAMPs) such as protein aggregates, necrotic cells, and cellular debris is widely thought to be a pivotal driver of both persistent and maladaptive immune activation in numerous neurological disorders[13,14]. In the case of central nervous system (CNS) injury, inefficient removal of DAMPs has been proposed to perpetuate neuroinflammation and incite secondary CNS pathology and neurological complications[8,13]. However, we currently lack complete knowledge of the drainage pathways that the brain relies on to dispose of DAMPs and resolve tissue damage following TBI.

Emerging studies over the last few years have shown that the meningeal lymphatics are centrally involved in the drainage of macromolecules, cellular debris, and immune cells from the brain to the periphery during homeostasis[15,16]. The anatomy and function of this CNS drainage pathway are just now being defined[15–21], and its role in many neurological diseases, including TBI, has not been elucidated. In recently published work, it was shown that these lymphatic vessels drain cerebrospinal fluid (CSF), interstitial fluid (ISF), CNS-derived molecules, and immune cells from the brain and meninges to the deep cervical lymph nodes (dCLN)[15,16,19,22]. Importantly, studies using in vivo magnetic resonance imaging (MRI) techniques have also identified the existence of meningeal lymphatic vessels in both humans and nonhuman primates[23,24]. More recent studies have also shown that the meningeal lymphatic system is critical for clearing amyloid beta, extracellular tau, and alpha synuclein from the brain, and that disruption of this drainage system can promote the accumulation of these neurotoxic DAMPs in the brain[18,25,26]. Whether meningeal lymphatic dysfunction plays a role in TBI remains poorly understood.

Here, we explore how the meningeal lymphatics are impacted following TBI, and how possessing defects in this drainage system before brain trauma influences TBI pathogenesis. We find that TBI results in compromised meningeal lymphatic drainage that can last out to at least one month post-injury. Mechanistically, we report that increased intracranial pressure (ICP), which is commonly observed in TBI, can impair meningeal lymphatic drainage. We further show that pre-existing deficits in meningeal lymphatic function predispose the brain to exacerbated neuroinflammation and cognitive deficits following brain trauma. Lastly, we demonstrate that prophylactic recuperation of meningeal lymphatic drainage capabilities in aged mice with viral delivery of VEGF-C mitigates gliosis in TBI.

## Results

**TBI causes meningeal lymphatic dysfunction.** To investigate whether meningeal lymphatic drainage function is impacted by TBI, we employed a mild closed-skull model of TBI that uses a stereotaxic electromagnetic impactor to deliver a single hit to the right inferior temporal lobe in the piriform region of the brain. This model is ideal for studying CNS lymphatic function because it does not rely on a craniotomy to perform the TBI and also does not result in a direct impact to the lymphatic vessels (Fig. 1a). Following TBI, injured mice had an average righting time of 300 s post-injury, compared to the average of 30 s for the sham mice (Fig. 1b), however TBI mice performed as well as sham mice in a series of behavioral tasks assessing injury-associated deficits in balance, motor coordination, reflex, and alertness (Fig. 1c). In addition, brain injury in this model did not affect performance on an accelerating rotarod (Fig. 1d) and brains from TBI mice only showed modest increases in measures of gliosis (Iba1 and GFAP staining) (Supplementary Fig. 1a, b). Together, these data indicate that this injury paradigm is relatively mild and does not result in noticeable behavioral deficits immediately after injury.

In order to assess whether meningeal lymphatic drainage is altered in this model of TBI, 0.5 μm fluorescent beads were injected intra-cisterna magna (i.c.m.) at various time points after injury and the dCLN, meninges, and brain were harvested 2 h after injection to assess for the presence of beads (Fig. 1e). Interestingly, when we examined cleared dCLN using confocal microscopy, we observed a substantial decrease in bead drainage to the dCLN in TBI mice as early as 2 h post-injury (Fig. 1f, g). Moreover, meningeal lymphatic drainage function remained significantly impaired out to one month post-injury (Fig. 1f, g) and it was not until two months post-injury that bead drainage function returned to pre-injury levels (Fig. 1f, g). Analysis of the brain and meningeal whole mounts revealed that the beads were taken up along the transverse sinuses (TSs) as previously published[17], and were also detected around the fourth ventricle and in the cerebellum, both of which are areas close to the i.c.m. injection site (Supplementary Fig. 2a, b). However, we were unable to detect any appreciable number of beads in the systemic circulation even after TBI (Supplementary Fig. 2c, d). To evaluate whether severity of the TBI plays a role in the duration of meningeal lymphatic drainage deficits, we performed TBI in the same location but with a higher impact velocity (6.2 meters (m) per second (s)) than the TBI used for the rest of the studies (5.2 m/s). At one month post injury, we found that there was a noticeable trend towards decreased drainage in the mice that had received the higher velocity injury (Supplementary Fig. 3a, b), suggesting that meningeal lymphatic drainage capacity is also likely affected by injury severity.

To determine whether the uptake of CSF into the meningeal lymphatic vasculature was altered in TBI, we examined the hotspots of CSF uptake from the sub-arachnoid space along the TSs[15,20]. Fluorescently labeled Lyve-1 antibody was injected into the CSF i.c.m. at either 2 or 24 h after TBI, and then the meninges were harvested to examine uptake of CSF contents in the hotspots 15 min after injection (Fig. 1h–j and Supplementary Fig. 4a–c). Analysis of the meningeal whole mounts revealed that mice that had received TBI 2 h prior demonstrated significantly decreased uptake of fluorescently labeled Lyve-1 antibodies at the hotspots when compared to mice that underwent a sham procedure (Fig. 1h, i). Moreover, Lyve-1 antibody did not travel as far along the lymphatics lining the TS in the TBI mice at both 2 and 24 h post brain injury (Fig. 1j and Supplementary Fig. 4c). Taken together, these findings indicate that even mild forms of TBI can result in meningeal lymphatic dysfunction and that these deficits persist out to one month post-injury. Moreover, we find that disruption in CNS drainage post-TBI is associated with impaired uptake of CSF at meningeal lymphatic hotspots.

**TBI induces morphological changes to lymphatic vasculature.** In order to determine whether the functional deficits in meningeal

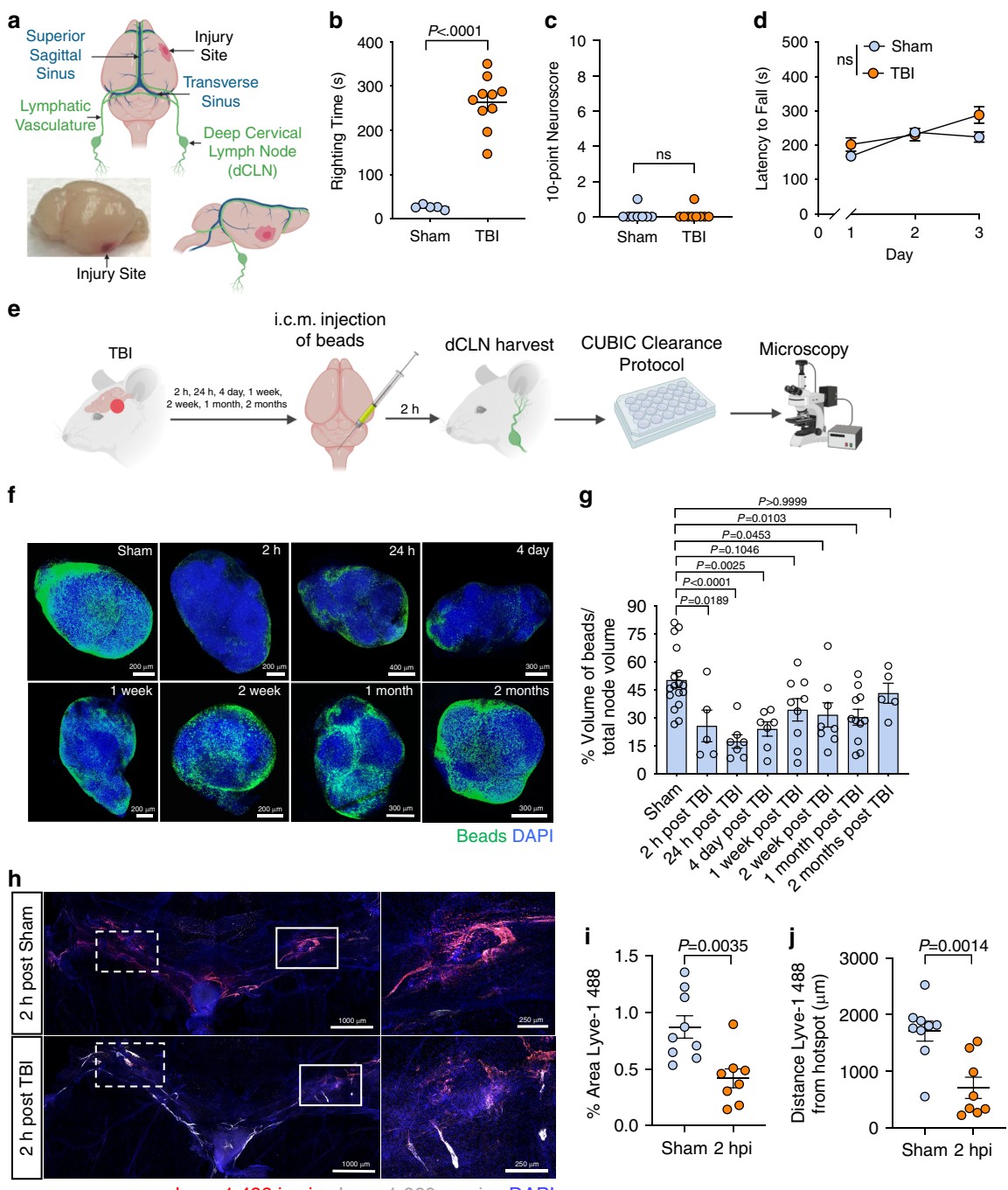

**Fig. 1 TBI leads to impairments in meningeal lymphatic drainage. a** Location of injury site in relation to the CNS lymphatic vasculature. **b** Righting times of TBI and sham mice (Sham $n = 5$, TBI $n = 10$; representative data from 10 independent experiments). **c** The 10-point gross neuroscore test 1 h after TBI (Sham $n = 8$, TBI $n = 9$; representative data from two independent experiments). **d** The accelerating rotarod behavioral test was used to assess motor function the first 3 days after TBI ($n = 10$ mice per group; representative data from two independent experiments). **e** Schematic of the experimental layout where mice received TBI and then were injected intra-cisterna magna (i.c.m.) with 0.5 μm fluorescent beads. **f** Representative images and **g** quantification of bead accumulation in the cleared dCLN at 2 h, 24 h, 4 days, 1 week, 2 weeks, 1 month, and 2 months post TBI. Each data point represents an average of the 2 dCLNs from an individual mouse (Sham $n = 17$, 2 h $n = 5$, 24 h $n = 7$, 4 day $n = 7$, 1 week $n = 9$, 2 weeks $n = 8$, 1 month $n = 11$, 2 months $n = 5$; pooled data from five independent experiments). **h** Representative images of meningeal whole mounts 2 h after TBI stained for Lyve-1 488 (in vivo, red), Lyve-1 660 (ex vivo, gray), and DAPI (blue). Solid boxes show zoomed insets of the hotspots along the transverse sinus on the right. Dashed boxes indicate the other hotspots not featured in the inset. **i**) Percent area of Lyve-1 488 (in vivo, red) coverage at 2 h post TBI or sham, and **j** distance traveled of Lyve-1 488 staining along the transverse sinus 15 min after injection (Sham $n = 9$, TBI $n = 8$; pooled data from two independent experiments). All $n$ values refer to the number of mice used and the error bars depict mean ± s.e.m. $P$ values were calculated by two-tailed unpaired Student's $t$-test (**b**, **c**, **i**, **j**), repeated-measures two-way ANOVA with Bonferroni's post hoc test (**d**), and one-way ANOVA with Bonferroni's multiple comparison test (**g**). hpi, hour(s) post injury; h, hour(s); mo, month(s); wk, week(s). Source data (**b–d**, **g**, **i–j**) are provided as a Source data file.

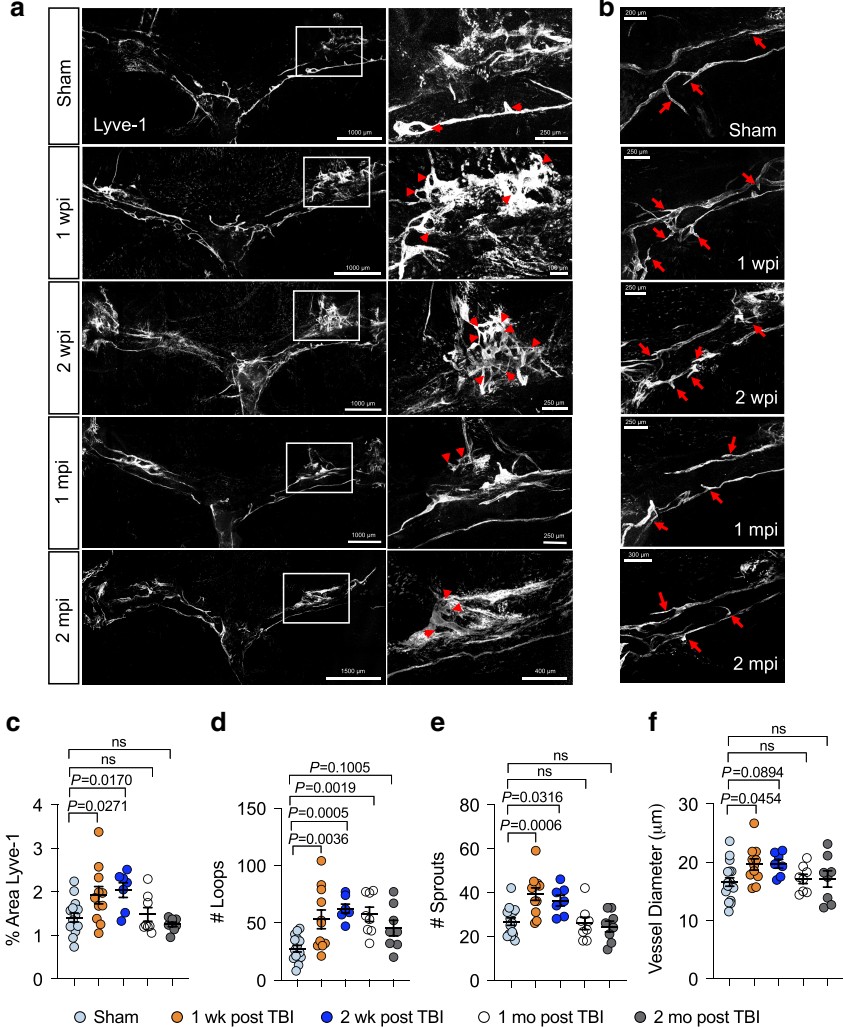

**Fig. 2 TBI causes changes in meningeal lymphatic vasculature morphology.** Mice received TBI or sham treatment and then meningeal whole mounts were harvested 1 week, 2 weeks, 1 month, and 2 months later. **a** Representative images depicting transverse sinuses (left) and meningeal lymphatic vasculature loops near lymphatic hotspots (right) and (**c**) quantification of the percent area coverage of Lyve-1 antibody staining and (**d**) the number of loops in meningeal whole mounts. Solid boxes show zoomed insets of the hotspots along the transverse sinus on the right. Red arrow heads in the insets of (**a**) denote meningeal lymphatic vasculature loops. **b** Representative images depicting meningeal lymphatic vasculature sprouts along the transverse sinuses and (**e**) quantification of the number of sprouts found in meningeal whole mounts. Red arrows in (**b**) denote meningeal lymphatic vasculature sprouts. **f** Quantification of the diameters of the meningeal lymphatic vessels. Each data point represents an independent mouse and is an average of 70 measurements along the transverse and superior sagittal sinuses per mouse. Data in (**c–f**): Sham $n = 15$, 1 week $n = 11$, 2 weeks $n = 7$, 1 month $n = 8$, 2 months $n = 8$; pooled data from four independent experiments. All $n$ values refer to the number of mice used and the error bars depict mean ± s.e.m. $P$ values were calculated by a one-way ANOVA with Dunnett's multiple corrections test (**c–f**). mo month(s), mpi month(s) post injury, wk week(s), wpi week(s) post injury. Source data (**c–f**) are provided as a Source data file.

lymphatic drainage were also accompanied by morphological changes, we assessed the meningeal lymphatic vasculature at different time points post injury. At one week post TBI, there was an overall increase in Lyve-1 percent area coverage in the dorsal meningeal lymphatics (Fig. 2a, c), which are known to be especially sensitive to growth factors in comparison to the larger and less labile collecting ducts at the base of the skull[19,20]. To further assess changes in Lyve-1 coverage and structure, we quantified the number of loops and sprouts within the dorsal lymphatic vasculature, as these morphological changes are believed to represent an overall higher lymphatic vasculature complexity and may also indicate lymphangiogenesis[27]. Interestingly, we observed a significant increase in the number of capillary loops and sprouts in TBI mice one week following head trauma (Fig. 2a, b, d, e). We also quantified the lymphatic diameter, another measure for lymphatic growth[18], and saw an increase in TBI mice at one week

post injury as compared to the sham group (Fig. 2f). These same alterations continued out to two weeks after TBI, where we saw significantly increased Lyve-1 percent area coverage, loop formation, and sprout numbers, as well as a trend toward increased lymphatic vessel diameter (Fig. 2a–f). By one and two months post injury, many of these measures of lymphangiogenesis had returned to baseline levels, although loop numbers remained significantly increased at one month post injury, taking longer to return to pre-injury levels (Fig. 2a–f). In addition, when we injected Lyve-1 antibody into the CSF i.c.m. one week post TBI or sham, we saw that there were no apparent differences in CSF uptake at the hotspots on the TSs between sham and TBI mice (Supplementary Fig. 4d–f), indicating that the lymphangiogenesis seen at one week post injury may have helped to restore proper CSF uptake into the meningeal lymphatic vasculature. Overall, these data indicate that TBI induces morphological changes in the

meningeal lymphatic vasculature and that these morphological changes occur maximally at one and two weeks post injury. Moreover, the results indicate that the lymphangiogenesis seen after a mild TBI is likely not permanent.

**Increased ICP contributes to CNS lymphatic dysfunction.** Elevated ICP is a major driver of mortality after TBI and is associated with negative clinical outcomes[28,29]. Because the lymphatic drainage deficits were substantial even 2 h after injury, we next examined whether there were associated changes in ICP after TBI. Two hours after injury, TBI mice exhibited markedly increased ICP as compared to sham mice (Fig. 3a, b). At later time points post-TBI, ICP levels stabilized at slightly higher (~5–7.5 mmHg) levels than baseline; however, the differences in ICP at these later time points were not found to be statistically significant (Fig. 3a, b).

Because the meningeal lymphatic vasculature is not associated with smooth muscle, it is especially vulnerable to changes in pressure and brain swelling inside the fixed skull[16,19]. Therefore, we speculated that an acute rise in ICP might lead to disruptions in meningeal lymphatic drainage. To specifically test this hypothesis, we subjected mice to bilateral internal jugular vein ligation (JVL, Fig. 3c), which is known to transiently increase ICP in both humans and mice[17,30,31]. Consistent with previous findings, we observed that JVL substantially increased ICP to an average of 10 mmHg 3 h after surgical ligation, and that the ICP normalized by 24 h post ligation (Fig. 3d)[17,30,31]. This reflected a similar acute spike in ICP that is seen at 2 h after TBI (Fig. 3a). To investigate what effect this rise in ICP has on meningeal lymphatic drainage function, we injected beads and assessed drainage to the dCLN at 3 and 24 h after JVL. We found that there was significantly less drainage to the dCLN 3 h after bilateral JVL (Fig. 3e, f) and that there was reduced bead accumulation around the meningeal lymphatic vasculature of ligated mice (Fig. 3h, i), indicating that there are deficits in the uptake of CSF contents in the CNS lymphatic vasculature. Moreover, we found that even after pressure normalized at 24 h (Fig. 3d), there was still a prolonged period of decreased lymphatic drainage as seen by the diminished uptake of beads into the meningeal lymphatics (Fig. 3j) and a trend towards decreased beads in the dCLN at 24 h post injury (Fig. 3g). Collectively, these findings suggest that increased ICP is capable of provoking meningeal lymphatic dysfunction.

**Prior lymphatic defects worsen TBI-induced inflammation.** TBI is an especially serious condition in the elderly and in individuals sustaining repetitive brain injuries[32–38]. For instance, similar injuries result in more severe pathology and neurological impairment in the elderly than in other age groups[32,36,39,40]. Mounting evidence also suggests that repetitive TBI can have devastating consequences that includes increasing one's risk of developing CTE, psychiatric disorders, and other forms of neurological disease later in life[34,41]. However, why TBI leads to worsened neurological disease in the elderly and following repetitive brain trauma remains poorly understood. Interestingly, it has recently been shown that CNS lymphatic drainage function significantly declines during aging[18,20,42]. Moreover, as we demonstrated in Fig. 1, a single mild head injury can result in pronounced disruptions in meningeal lymphatic function. This led us to question whether pre-existing meningeal lymphatic dysfunction contributes to exacerbated disease following TBI, and if this might help to explain the increased severity of TBI-associated disease seen in repetitive TBI and the elderly.

Therefore, to formally investigate how antecedent meningeal lymphatic deficits affect outcomes after TBI, we utilized a

pharmacological approach to selectively ablate the meningeal lymphatic vessels before head injury. Visudyne, a photoconvertible drug that has been shown to effectively ablate lymphatic vasculature[17,18,43], was injected i.c.m. into the CSF and allowed to travel into the CNS lymphatics. A nonthermal 689-nm laser was then aimed through the skull to selectively photoablate the meningeal lymphatics. Visudyne photoablation has been shown to selectively target lymphatic endothelial cells and results in the loss of lymphatic endothelial cell markers including Lyve-1, Prox1, and Podoplanin, while largely sparing the surrounding blood vasculature[17,43]. Indeed, other studies show that this photoablation procedure has no impact on blood flow or blood oxygenation four days after treatment when measured by photoacoustic imaging[17]. After performing this procedure, mice were then rested for one week before receiving either sham treatment or brain injury (Supplementary Fig. 5a). Photoablation after Visudyne injection (Visudyne + laser) resulted in a significant decrease in area of the meninges covered by Lyve-1-expressing lymphatic vessels in comparison to mice that received vehicle and laser treatment (Vehicle + laser) or Visudyne without laser treatment (Visudyne) (Supplementary Fig. 5b, c). Consistent with previous reports[17,18,43], the area of CD31+Lyve-1− blood vasculature was unchanged between all experimental groups, both when measured throughout the entire meninges and also in the region surrounding the sites of photoablation (Supplementary Fig. 5b,d,e). Overall, this indicates that the meningeal lymphatic vasculature was ablated without visible changes to the blood vasculature.

In order to determine how pre-existing meningeal lymphatic dysfunction affects subsequent TBI, we performed bulk RNA sequencing 24 h and 1 week post injury in mice that had undergone either Visudyne ablation or a sham procedure one week prior. At 24 h, while the sham groups clustered close together in the principal component analysis, the groups of mice that received TBI clustered separately (Fig. 4a). Moreover, the group that received both TBI and ablation clustered separately from the group that received TBI alone (Fig. 4a). Not surprisingly, at this early time point post brain injury, the injury itself caused the largest changes in differentially regulated gene expression (Fig. 4b, c). We found 1573 upregulated and 1409 downregulated genes (FDR < 0.1), when comparing TBI mice to the mice that underwent a sham procedure. We found similar numbers of upregulated (1511) and downregulated (1363) genes when comparing injured and uninjured mice that underwent meningeal lymphatic ablation (FDR < 0.1). To understand how pre-existing lymphatic dysfunction affects subsequent TBI, we compared the two groups that had received TBI, where one group received ablation before brain injury (Ablated + TBI) and the other received a sham procedure (Not Ablated + TBI). Interestingly, we saw that the group that had pre-existing lymphatic dysfunction before TBI had a significant upregulation in complement-related genes just 24 h post injury (Fig. 4d–f). Upregulation of the complement pathway, while important for coordinating the innate immune response and for pruning synapses in development, can be highly detrimental for brain health later in life. For instance, over-activation of the complement system has been shown to be an early harbinger for neuronal loss and cognitive decline[44–46]. One quarter of the top 20 most significantly upregulated genes in the Ablated + TBI group in comparison to the Not Ablated + TBI group were related to the complement pathway, including *C1qc*, *Itgam*, *C4b*, *Ctsh*, and *Irf7* (Fig. 4e). To further assess the overall changes in the complement system, we created a heat map of the top 20 most significantly differentially expressed complement cascade-related genes using the Broad Hallmark gene sets[47]. We found that there was substantial upregulation of many genes related to the complement pathway

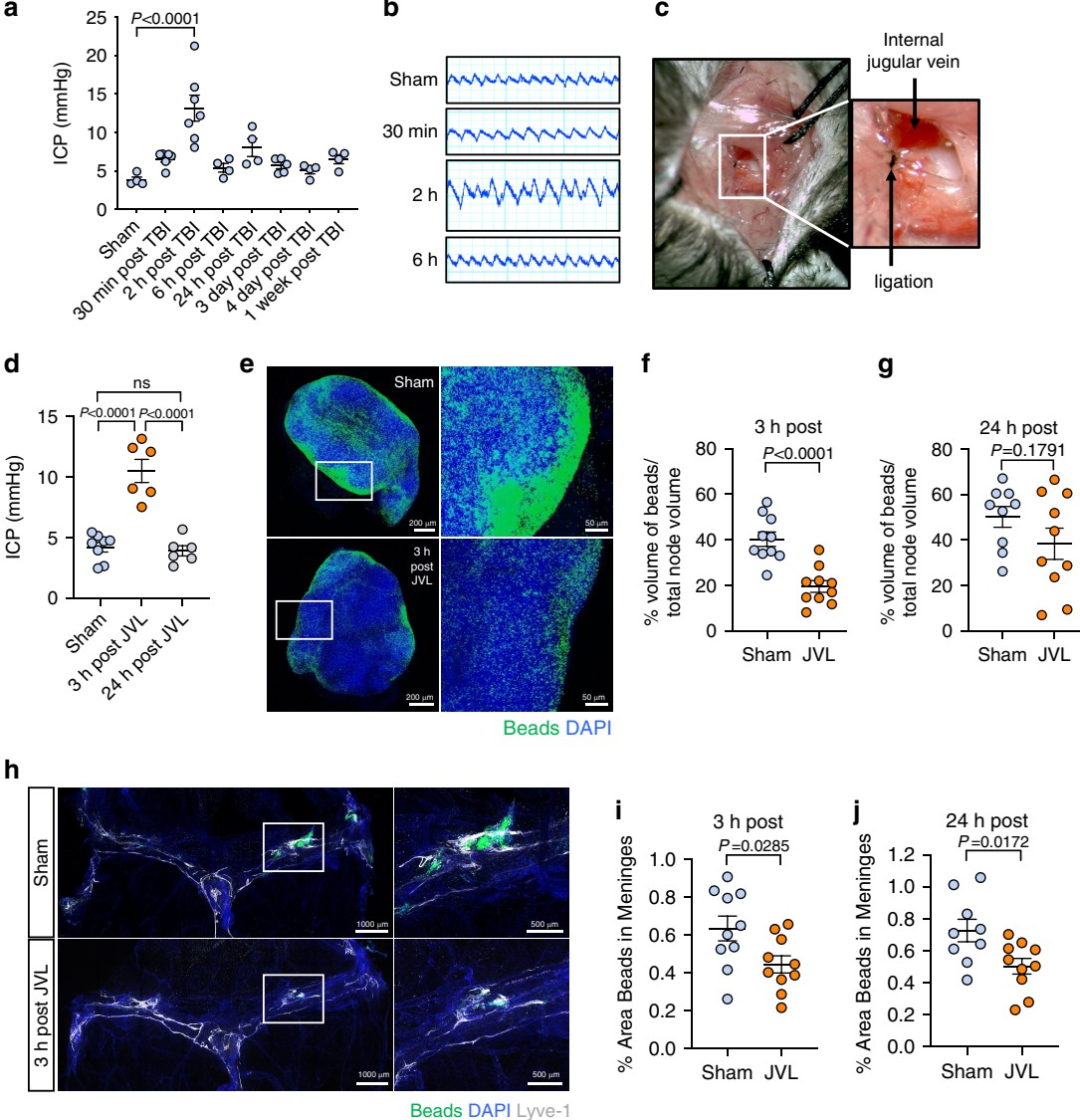

**Fig. 3 Increases in intracranial pressure disrupt CNS lymphatic drainage. a** Measurements of intracranial pressure (ICP) and **b** representative pressure readings were collected at various time points after TBI (Sham $n = 4$, 30 min $n = 6$, 2 h $n = 7$, 6 h $n = 4$, 24 h $n = 4$, 3 days $n = 5$, 4 days $n = 4$, 1 week $n = 4$; pooled data from three independent experiments). **c** Representative images of internal jugular vein ligation (JVL). **d** ICP readings from mice that underwent bilateral JVL or a sham procedure 3 or 24 h prior (Sham $n = 8$, 3 h $n = 6$, 24 h $n = 6$; pooled data from three independent experiments). **e–j** The internal jugular vein was ligated bilaterally and then 0.5 µm fluorescent beads were injected i.c.m. 3 h later. dCLN and meninges were then harvested from mice 2 h after bead injection. **e** Representative images of dCLN and graph showing drainage of beads (**f**) 3 h and (**g**) 24 h after jugular venous ligation. Each data point represents an average of the 2 dCLNs from an individual mouse (3 h: Sham $n = 10$, JVL $n = 10$, 24 h: Sham $n = 9$, JVL $n = 10$; pooled data from two independent experiments). Solid boxes of the node images on the left show zoomed insets of the images on the right. **h** Representative images of meningeal whole-mounts with 0.5 µm beads (green) stained with DAPI (blue) and Lyve-1 660 (gray) and graph depicting percent area of bead coverage (**i**) 3 h and (**j**) 24 h post-JVL. Solid box shows a zoomed inset of the hotspot along the transverse sinus on the right (3 h: Sham $n = 10$, JVL $n = 10$, 24 h: Sham $n = 9$, JVL $n = 10$; pooled data from two independent experiments). All $n$ values refer to the number of mice used and the error bars depict mean ± s.e.m. $P$ values calculated by one-way ANOVA with Bonferroni's multiple comparison test (**a**) and Tukey's multiple comparison test (**d**) and two-tailed unpaired Student's $t$-test (**f**, **g**, **i**, **j**). ICP intracranial pressure, JVL jugular venous ligation, h hour(s), min minute(s). Source data (**a**, **d**, **f–g**, **i–j**) are provided as a Source data file.

in the Ablated + TBI group as compared to the Not Ablated + TBI group (Fig. 4f).

To further address the question of how pre-existing lymphatic dysfunction affects subsequent TBI, we decided to take a closer look at the inflammatory pathways known to affect brain health. We found many of the upregulated genes in Ablated + TBI mice compared to Not Ablated + TBI mice were enriched in innate immune pathways (Fig. 4g). Given this increase in inflammation

with pre-existing lymphatic dysfunction before TBI, we were interested in determining whether neuronal health was affected at this early time point. Interestingly, we found multiple down-regulated genes that are critical for neuronal health including *Arc*, *Homer1*, *Homer2*, and *Bdnf* (Fig. 4h). The most significantly downregulated gene in the Ablated + TBI group when compared to the Not Ablated + TBI group was *Arc* (Fig. 4d, e, h). *Arc*, an immediate early gene, is known to be critical for maintaining

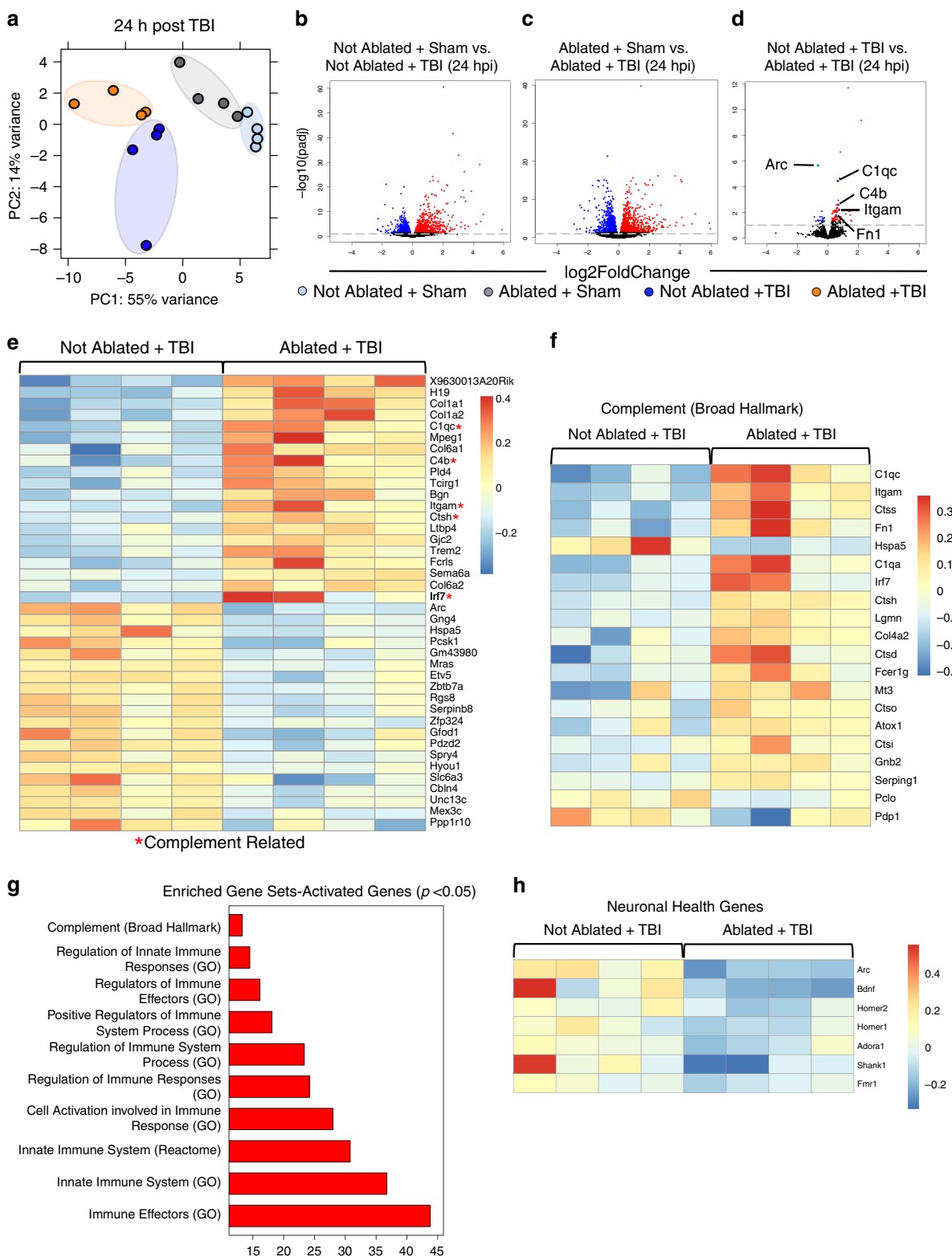

neuronal health[48]. Interestingly, even small reductions in *Arc* expression are known to be highly detrimental for learning capabilities, memory, and brain health[49].

To determine whether the changes seen in the Not Ablated + TBI vs. Ablated + TBI comparison could be attributed to the ablation procedure, we investigated the changes observed between the Not Ablated + Sham group and the Ablated + Sham group.

We found that although ablation alone resulted in 338 differentially expressed genes, only 29 of these were shared with the TBI + ablation group (Supplementary Fig. 6a, b, d). Shared signatures between these two comparisons include an upregulation in genes important for collagen production (*Col1a1, Col1a2*, and *Col6a2*) and an upregulation in *H19* (important for noncoding RNA synthesis[50]) (Supplementary Table 1). Interestingly, we found that

**Fig. 4 Pre-existing meningeal lymphatic dysfunction alters gene expression 24 h after TBI.** Mice were subjected to an injection of Visudyne or vehicle i.c.m. and 15 min later a red laser was directed at 5 spots along the sinuses through the skull. After a week of rest, mice received TBI or a sham procedure. 24 h after injury, RNA was isolated from homogenized brains. Bulk RNA sequencing was performed on four experimental groups with four samples per group. **a** Principal component (PC) analysis showing clustering of samples. **b–d** Volcano plots illustrate the number of significantly differentially expressed genes (FDR < 0.1). Blue data points represent significantly downregulated genes and red data points represent significantly upregulated genes. Individual genes are highlighted in (**d**), where select downregulated genes are marked green and select upregulated genes are marked purple. **e** Heatmap representation of the top 20 most significantly upregulated and downregulated (FDR < 0.1) genes in the Not Ablated + TBI vs. Ablated + TBI groups. The red star (*) indicates genes associated with the complement signaling cascade. **f** Heatmap representation of the 20 most significantly differentially expressed Broad Hallmark Complement-related genes in the Not Ablated + TBI vs. Ablated + TBI comparison. **g** Gene set enrichment analysis of upregulated genes in Ablated + TBI mice compared to Not Ablated + TBI mice (uncorrected $p < 0.05$). **h** Heatmap representation of genes associated with neuronal health in the Ablated + TBI group as compared to the Not Ablated + TBI group. FDR and $P$ values were calculated with DEseq2 using the Wald test for significance following fitting to a negative binomial linear model and the Benjamini–Hochberg procedure to control for false discoveries. hpi hours post injury, padj adjusted $p$-value.

---

genes related to complement (*C1qc, C4b, Ctsh, Irf7*, and *Itgam*) and neuronal health (*Arc*) were less significantly changed by orders of magnitude, or not changed in the Not Ablated + Sham vs. Ablated + Sham comparison in relation to the Not Ablated + TBI vs. Ablated + TBI comparison (Supplementary Table 1). Therefore at 24 h post injury, while there are changes from the ablation alone, there are important differences in the differentially regulated gene signatures and significance of these changes when compared to those seen in the Not Ablated + TBI vs. Ablated + TBI comparison (Supplementary Table 1). Overall, these findings indicate that possessing pre-existing deficits in meningeal lymphatic function before TBI leads to elevated expression of genes associated with neuroinflammation and complement signaling at 24 h post-injury. Moreover, we find that possessing unresolved defects in this meningeal drainage system before head injury adversely affects the expression of neuronal health genes.

We also performed bulk RNA sequencing at 1 week post injury to investigate the effects of pre-existing lymphatic dysfunction at a more distant time point after TBI. At 1 week post injury, both the Not Ablated + Sham and the Ablated + Sham groups clustered together, indicating that the photoablation alone had very little effect at one week post injury (Fig. 5a). Moreover, there were only 11 differentially regulated genes in the Not Ablated + Sham vs. Ablated + Sham group, indicating that at one week post injury, the ablation procedure itself is not playing a significant role in changing gene expression (Supplementary Fig. 6a, c). Interestingly, both the group with TBI alone and the group with pre-existing lymphatic dysfunction before TBI (Ablated + TBI) clustered separately (Fig. 5a). While the TBI alone still results in many differentially regulated genes, we find that there are over 200 differentially regulated genes in the TBI group with pre-existing lymphatic dysfunction (Ablated + TBI) when compared to TBI alone (FDR < 0.1, Fig. 5a, b). A gene set enrichment analysis with the pathways in the Reactome database for differentially expressed genes between TBI mice with pre-existing lymphatic dysfunction (Ablated + TBI) and those with TBI alone (Not Ablated + TBI) showed that the most highly enriched pathways were related to the innate and adaptive immune system and cytokine signaling (Fig. 5c)[51]. We also used the Biocarta database[52] to determine which signaling pathways were most highly enriched in our dataset. We found that genes differentially expressed between the TBI mice with ablated meningeal lymphatics (Ablated + TBI) and TBI mice with normal lymphatics (Not Ablated + TBI) are enriched in pathways important for innate immunity (e.g., complement pathway), leukocyte recruitment (e.g., CCR5), and cytokine signaling (e.g., IL-1 and TNF signaling) (Fig. 5d).

We were next interested in investigating whether the TBI mice with pre-existing lymphatic dysfunction (Ablated + TBI) shared common genes associated with neurodegenerative or psychiatric diseases. To this end, we used the GWAS Catalog[53], a database of genome-wide association studies, to find known gene associations with several common neurodegenerative and psychiatric diseases including Parkinson's disease, schizophrenia, amyotrophic lateral sclerosis (als), Alzheimer's disease, bipolar disease, depression, multiple sclerosis (ms), and obsessive compulsive disorder (ocd). We found that the differentially expressed genes between TBI mice with pre-existing lymphatic dysfunction and mice with TBI alone were also associated with these diseases, indicating that these mice share signatures with various neurological disease states (Fig. 5e, f).

**Unresolved drainage defects worsen cognitive decline in TBI.** The findings from the RNA-seq experiments motivated us to investigate whether there are different long-term outcomes in the TBI mice with pre-existing meningeal lymphatic dysfunction (Ablated + TBI) compared to mice with TBI alone (Not Ablated + TBI). We looked at levels of Iba1 (labels microglia and CNS infiltrating monocytes/macrophages) and GFAP (labels reactive astrocytes) immunoreactivity 2 weeks after injury or sham treatment, with or without ablation, as aggravated gliosis often correlates with worsened clinical outcomes in TBI[54–57]. We found that possessing defects in the meningeal lymphatic system before TBI (Ablated + TBI) results in a significantly higher percent area of GFAP coverage (Fig. 6a, b). While we saw an overall increase in Iba1 immunoreactivity in the Ablated + TBI group compared to the Not Ablated + Sham group, we did not see a significant increase when compared to TBI alone (Fig. 6a, c). These initial results suggest that more severe neuroinflammation can unfold if TBI occurs in the context of pre-existing meningeal lymphatic dysfunction.

Because we saw a trend towards increased percent area Iba1 staining in Ablated + TBI mice (Fig. 6c), we wanted to more closely examine Iba1 cell morphology in each of the experimental groups at 2 weeks post injury. We performed a Sholl analysis and found that Iba1+ cells from the pre-existing lymphatic dysfunction TBI group (Ablated+TBI) had a lower number of dendritic branches at greater distances from the soma, indicating a less ramified, more highly activated state than in mice with TBI alone (Not Ablated + TBI), or the other control groups (Fig. 6d, e). In addition, the Ablated + TBI group had a higher number of Iba1+ cells (Fig. 6f), and trends towards lower dendrite length, dendrite branch points, and dendrite volume compared to the other control groups (Supplementary Fig. 7a–c). These data suggest that brain macrophages (Iba1 + cells) in TBI mice with pre-existing lymphatic dysfunction (Ablated + TBI) are more highly activated and less ramified than Iba1+ cells in the other control groups.

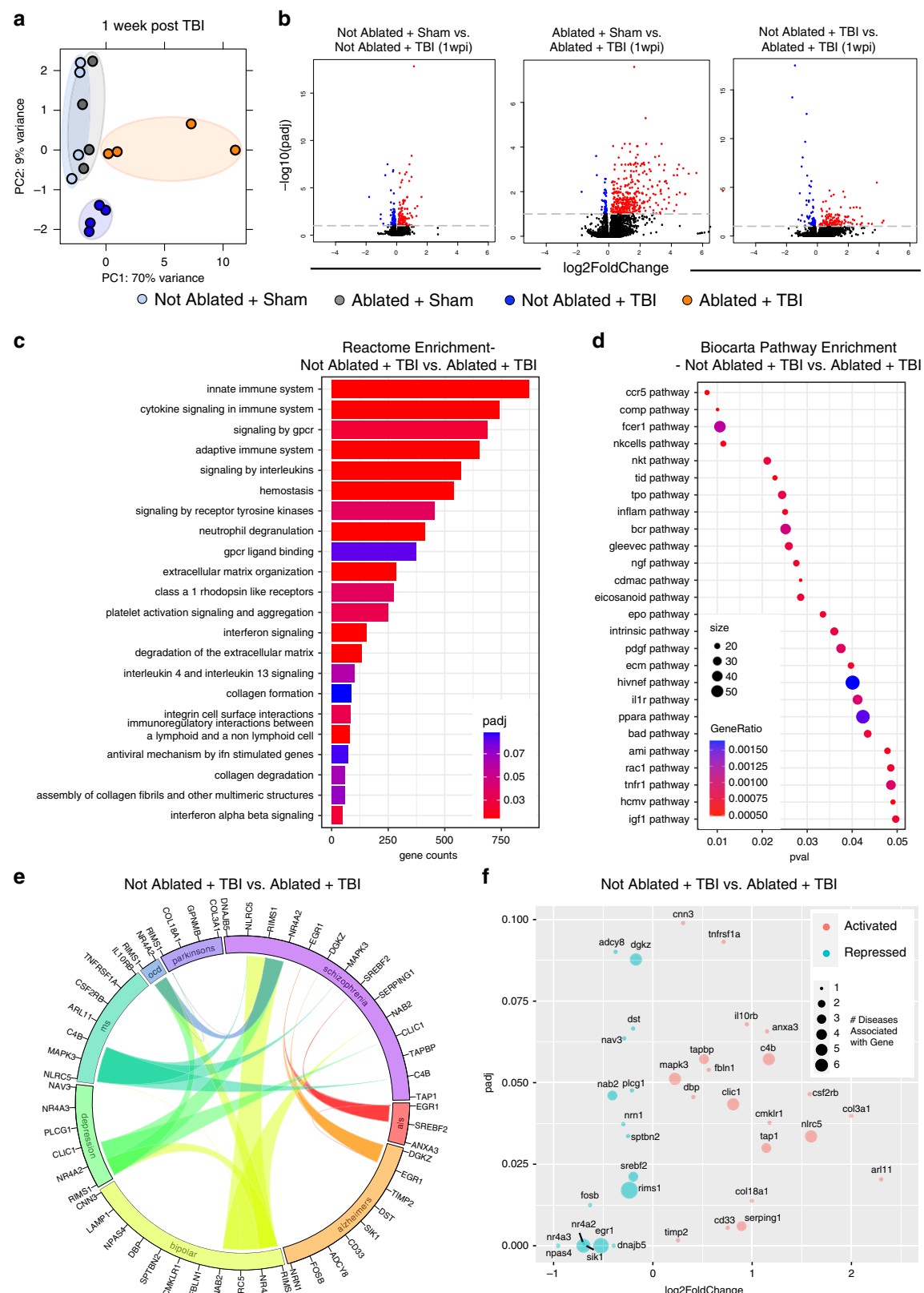

Next, we sought to investigate whether possessing deficits in meningeal lymphatic function before TBI negatively affects cognitive performance, which is a common consequence of TBI in humans and animal models. Since others have shown that similar injuries to this region of the brain can lead to cognitive impairments[58,59], we decided to investigate how preexisting

lymphatic dysfunction affects memory and motor learning following TBI. Interestingly, while all four experimental groups exhibited similar performance on the accelerating rotarod at 24 h post-TBI, the mice possessing deficits in meningeal lymphatic function before TBI (Ablated + TBI) showed impaired motor learning over the three days of the accelerating rotarod test

**Fig. 5 TBI leads to elevated expression of disease-associated genes when the brain possesses pre-existing lymphatic deficits.** Mice were subjected to an injection of Visudyne or vehicle i.c.m. and 15 min later a red laser was directed at 5 spots along the sinuses through the skull. After a week of rest, mice received TBI or a sham procedure. One week after injury, RNA was isolated from homogenized brains. Bulk RNA sequencing was performed on these four experimental groups with four individual mice per group. **a** Principal component (PC) analysis showing clustering of samples. **b** Volcano plots illustrate the number of significantly differentially expressed genes (FDR < 0.1). Blue data points represent significantly downregulated genes and red data points represent significantly upregulated genes. **c, d** Gene set enrichment analysis using the (**c**) Reactome database and (**d**) Biocarta pathway database shows enrichment of immune-related pathways with differentially expressed genes between TBI mice with pre-existing lymphatic dysfunction compared to mice with TBI alone. **e** Circos plot depicting differentially expressed genes in TBI mice with pre-existing lymphatic dysfunction compared to mice with TBI alone (FDR < 0.1) associated with neurodegenerative or psychiatric diseases. The proportion of the circle's circumference allocated to each disease represents the number of genes associated with that disease that are also differentially expressed in the Not Ablated + TBI vs. Ablated + TBI comparison. The lines connecting genes within the circle indicate which genes were shared amongst disease signatures. **f** Scatterplot showing the adjusted $P$ value and the expression changes of genes shown in (**e**). padj, adjusted $p$-value; pval, $p$-value. wpi week(s) post injury. FDR and $P$ values were calculated with DEseq2 using the Wald test for significance following fitting to a negative binomial linear model and the Benjamini–Hochberg procedure to control for false discoveries.

(Fig. 6g). Indeed, the percent performance increase over three days in the Ablated + TBI group was lower than any of the other control groups indicating that undergoing meningeal lymphatic photoablation before TBI results in impaired motor learning (Fig. 6g). Likewise, mice that underwent meningeal lymphatic photoablation before TBI also performed worse in the novel location recognition test (NLRT) at two weeks post-brain injury, suggesting an impairment in memory (Fig. 6h, i). Taken together, these results indicate that TBI can cause exacerbated neuroinflammation and cognitive dysfunction when the brain possesses pre-existing defects in meningeal lymphatic function.

**Lymphatic rejuvenation mitigates TBI-driven inflammation.** To better understand the functional consequences of impaired meningeal lymphatic function in TBI, we next explored whether boosting CNS lymphatic drainage is effective in attenuating TBI disease pathogenesis. Multiple recent studies have shown that aging leads to severe impairments in meningeal lymphatic function[18,20,42]. Moreover, it is also known that even mild-to-moderate forms of brain trauma can have especially devastating consequences in elderly individuals[32,33,35,36,40]. Therefore, we were interested in investigating whether recuperating meningeal lymphatic function in aged mice would be effective in limiting TBI disease pathogenesis. To this end, we utilized viral delivery of VEGF-C, which has previously been shown to successfully increase the diameter of the meningeal lymphatic vessels[15,18,21]. Importantly, delivery of VEGF-C to the meningeal lymphatic vessels in aged mice has also been reported to rejuvenate meningeal lymphatic draining function[18]. Accordingly, we subjected aged mice (18–24 months of age) or young mice (8–10 weeks of age) to an i.c.m injection with either AAV1-CMV-mVEGF-C or control AAV1-CMV-eGFP. We observed stable expression of the viral vector along the TS and superior sagittal sinus (SSS) in the aged meninges 1 month after injection (Fig. 7a). Two weeks after viral vector delivery, we subjected these mice to TBI or sham treatment. Not surprisingly, the aged mice who experienced head trauma had a significantly longer loss-of-consciousness and performed significantly worse on the neuroscore behavioral tests than their younger counterparts with the same injury parameters (Fig. 7b, c). There were, however, no differences in righting time or neuroscore between the aged TBI groups that had received AAV1-CMV-mVEGF-C or AAV1-CMV-eGFP delivery (Fig. 7b, c). Mice that received sham procedures instead of a TBI and either AAV1-CMV-mVEGF-C or AAV1-CMV-eGFP delivery also showed no differences in the righting times or neuroscore behavioral tests, although predictably, both measurements were lower than in mice that had received TBI (Supplementary Fig. 8a, b and Fig. 7b, c). Two weeks after injury and one month after viral vector delivery, we

measured GFAP and Iba1 immunoreactivity to assess gliosis and determine whether treatment with VEGF-C improved neuroinflammatory measures in the injured brains. Interestingly, we saw that aged TBI mice that received viral-mediated VEGF-C treatment (Aged- VEGFC + TBI) had significantly lower levels of Iba1 in the hemisphere contralateral to the injury site, and a trend towards lower levels of Iba1 in the hemisphere ipsilateral to the injury site when compared to aged TBI mice that had received the control viral vector (Aged- GFP + TBI) (Fig. 7d–f). Notably, the levels of Iba1 in the aged TBI mice that were treated with VEGF-C (Aged- VEGFC + TBI) were more similar to the levels of Iba1 immunoreactivity in young TBI mice (Young- GFP + TBI) (Fig. 7d–f and Supplementary Fig. 8d, f). In contrast, VEGF-C pretreatment was not found to influence GFAP immunoreactivity in aged mice following head trauma (Fig. 7d, g, h and Supplementary Fig. 8c, e). These data indicate that boosting meningeal lymphatic function after TBI may aid in decreasing levels of Iba1 gliosis in aged mice.

## Discussion

Despite TBI being a significant medical issue, the biological factors that promote CNS pathology and neurological dysfunction following head trauma remain poorly characterized. Recently, the meningeal lymphatic system was identified as a critical mediator of drainage from the CNS. In comparison to other peripheral organs, our understanding of how defects in lymphatic drainage from the CNS contribute to disease is limited. Here, we report that meningeal lymphatic function is impaired after TBI and that this disruption begins almost immediately and may take months to fully return to pre-injury levels. We further show that ICP is significantly elevated at two hours post-brain injury, and that this increase in ICP can promote hotspot dysfunction in the uptake of CSF from the subarachnoid space. Moreover, we find that increased ICP is sufficient to cause meningeal lymphatic dysfunction. Our work also provides evidence that pre-existing lymphatic dysfunction, as may occur with repetitive head injury and aging, predisposes the brain to more severe neuroinflammation and cognitive deficits.

Several recent studies in other models of disease have shown that the meningeal lymphatic system is critical for modulating immune responses and inflammation in the CNS[17,18,21]. Whether CNS lymphatic drainage is involved in promoting or resolving inflammation is likely specific to individual disease settings. For instance, mounting evidence indicates that lymphatic drainage plays a role in promoting autoimmunity by facilitating drainage of brain antigens to the peripheral dCLN. In the context of experimental autoimmune encephalomyelitis (EAE), ablation of the meningeal lymphatic vasculature was found to decrease disease severity by limiting CD4$^+$ T cell infiltration into the spinal

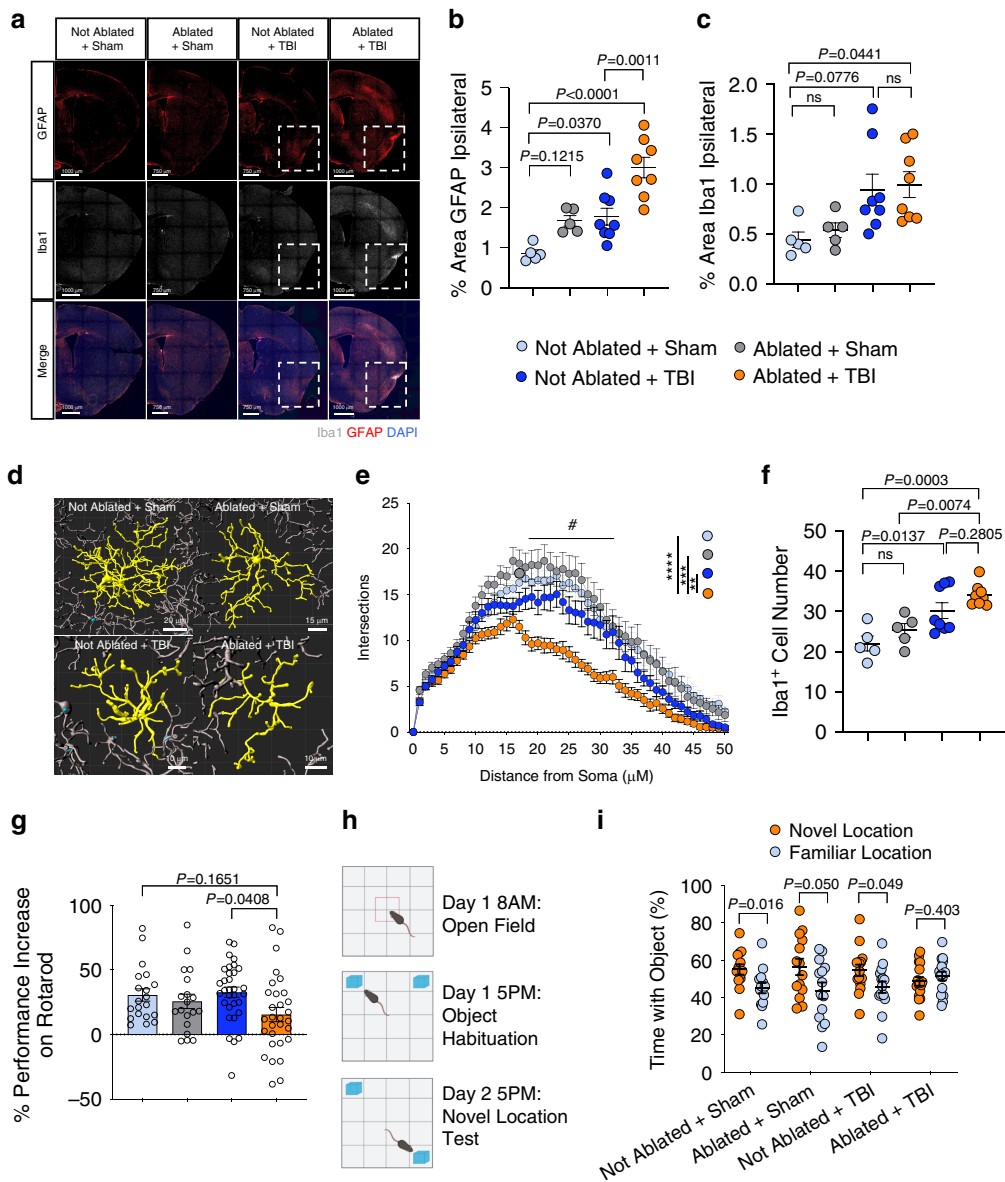

**Fig. 6 Prior lymphatic defects lead to exacerbated TBI-induced inflammation and cognitive decline.** Mice were subjected to meningeal lymphatic photoablation or a control procedure and then to TBI or a sham procedure 1 week later. **a–f** Brains were harvested 2 weeks after TBI. **a** Representative images of the brain hemisphere ipsilateral to the injury and quantification of the percent area of (**b**) GFAP (red) and (**c**) Iba1 (gray) immunoreactivity (Sham groups $n = 5$, TBI groups $n = 8$). Dashed boxes denote the injury site. **d** Representative reconstructions of morphology of Iba1+ cells and (**e**) Sholl analysis. Each data point represents the number of Iba1 + branches intersecting with a radius of 0–50 μm from the soma, calculated by the average of 20 microglia per group (4 Iba1 + cells per section, 5 mice per group). Number sign (#) indicates that all control groups were significantly different from the Ablated + TBI group from 18–32 μm; ****$P < 0.0001$, ***$P = 0.0003$, **$P = 0.0034$ (calculated at 23 μm). **f** Quantification of the average number of Iba1+ cells per field of view (Sham groups $n = 5$, TBI groups $n = 8$). **g** The percent performance increase on the accelerating rotarod from day 1 to day 3 post TBI or sham procedure (Not Ablated + Sham $n = 20$, Not Ablated + TBI $n = 32$, Ablated + Sham $n = 19$, Ablated + TBI $n = 30$; pooled data from four independent experiments). **h** NLRT experimental schematic and **i** percent time the mouse spent investigating each object over the total time investigating both objects (Not Ablated + Sham $n = 14$, Not Ablated + TBI $n = 15$, Ablated + Sham $n = 14$, Ablated + TBI $n = 16$; pooled data from two independent experiments). Each point represents the percent time one mouse spent with either the novel location object (orange) or the familiar location object (blue). All other $n$ values refer to the number of mice used and the error bars depict mean ± s.e.m. $P$ values calculated by two-way ANOVA with Tukey's multiple comparison correction (**b**, **c**, **e**, **f**, **g**) and mixed two-tailed unpaired Student's t-test with Holm-Sidak multiple comparison correction (**i**). Source data (**b**–**c**, **e**–**g**, **i**) are provided as a Source data file.

cord[17]. Consistent with a disease-promoting role for CNS lymphatic drainage in EAE, other studies have shown that lymphangiogenesis near the cribriform plate is a hallmark of disease progression and suggest that this may augment the peripheral adaptive immune response to myelin peptides[60]. In other instances, drainage of macromolecules and protein aggregates from the brain through the meningeal lymphatics is essential to maintain CNS health, as was recently shown to be the case in mouse models of Alzheimer's disease[18]. In this study by Da Mesquita et al., it was shown that blocking lymphatic drainage with Visudyne photoablation results in the accumulation of amyloid beta aggregates in the meninges and hippocampus.

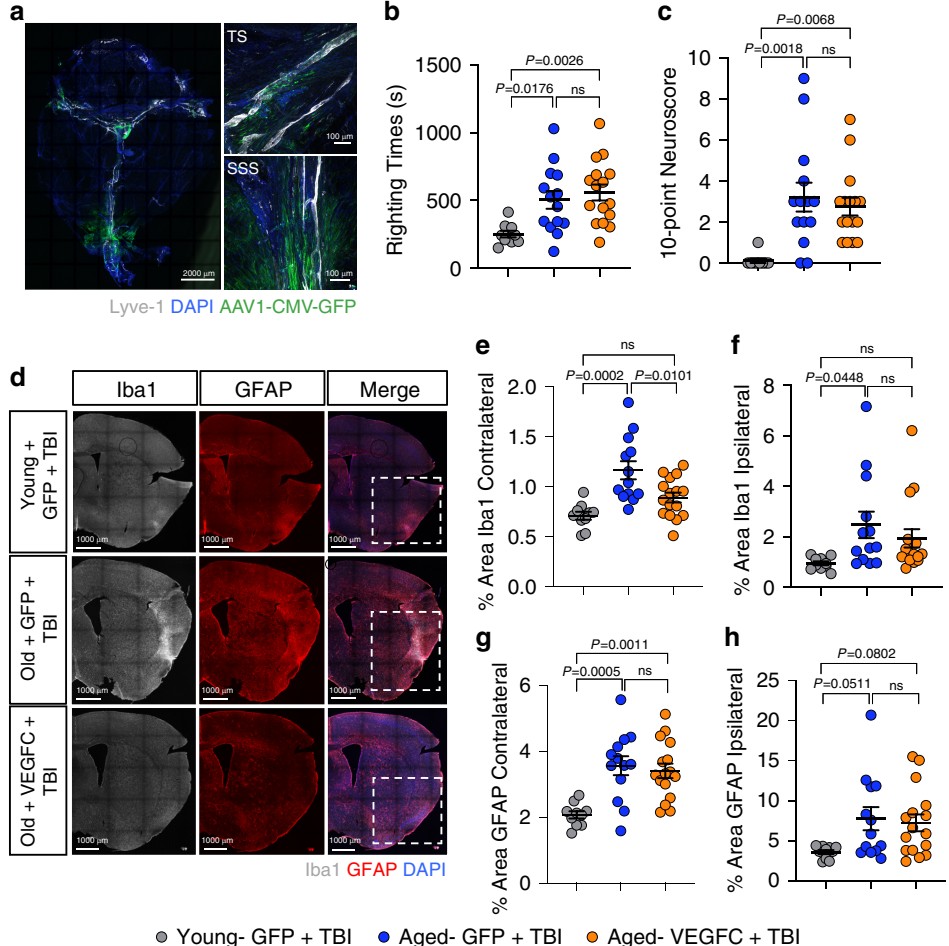

**Fig. 7 VEGF-C treatment of aged mice results in decreased neuroinflammation after TBI.** Aged mice (18–20 months old) received 2 µl of artificial CSF containing $10^{13}$ genome copies per ml of either AAV1-CMV-mVEGF-C or control AAV1-CMV-eGFP by i.c.m. injection to rejuvenate meningeal lymphatic drainage function. Young mice (8–10 weeks of age) received 2 µl of artificial CSF containing $10^{13}$ genome copies per ml of control AAV1-CMV-eGFP by i.c.m. injection. Mice were rested for two weeks and then were subjected to either TBI or sham procedures. Two weeks after injury, brains were harvested, sectioned and stained for markers of gliosis. **a)** Representative image of a meningeal whole mount 1 month after viral vector administration showing Lyve-1 (gray), DAPI (blue), and AAV1-CMV-eGFP (green). **b** Righting times (Young-GFP + TBI $n = 10$, Aged-GFP + TBI $n = 14$, Aged-VEGFC + TBI $n = 16$; pooled data from two independent experiments) and **c** 10-point gross neuroscore of mice at 1 h after TBI (Young-GFP + TBI $n = 9$, Aged-GFP + TBI $n = 14$, Aged-VEGFC + TBI $n = 16$; pooled data from two independent experiments). **d** Representative images of hemisphere ipsilateral to the site of injury stained with Iba1 (gray), GFAP (red), and DAPI (blue). Dashed boxes in merge column indicate lesion site. **e–h** Quantification of the percent area of (**e–f**) Iba1 and (**g–h**) GFAP immunoreactivity in the hemispheres contralateral and ipsilateral to the site of injury. (Young-GFP + TBI $n = 10$, Aged-GFP + TBI $n = 13$, Aged-VEGFC + TBI $n = 16$; pooled data from two independent experiments). All $n$ values refer to the number of mice used and the error bars depict mean ± s.e.m. $P$ values calculated by one-way ANOVA with Tukey's multiple comparison correction (**b, c, e, f, g, h**). SSS superior sagittal sinus, TS transverse sinus. Source data (**b–c, e–h**) are provided as a Source data file.

Furthermore, recent evidence also suggests that having proper meningeal lymphatic drainage is critical for mounting protective immune responses to invasive and hard-to-target brain cancers[21].

Neuroinflammation and gliosis can often persist for months or even years post-brain trauma[7,9], and interventions aimed at attenuating this neuroinflammation have proven successful in modulating cognitive outcomes[61,62]. Yet, the physiological processes involved in the prolonged inflammatory state of the TBI brain remain poorly defined. Based on our findings presented here, it is possible that impaired drainage of DAMPs such as amyloid beta, necrotic cells, and cellular debris from the brain could incite prolonged immune activation in the injured brain. Therefore, therapeutic approaches that promote functional recovery of the meningeal lymphatic system could provide strategies to help curtail the persistent neuroinflammation that is a hallmark of TBI.

An acute rise in ICP after head trauma is a poor prognostic indicator in TBI patients and is estimated to account for nearly half of all TBI mortalities[28,29]. We found that the rise in ICP seen after TBI or jugular vein ligation results in decreased meningeal lymphatic drainage, indicating that changes in the CNS environment can rapidly impact lymphatic function. While acute rises in ICP after injury caused by brain edema and swelling may result in decreased drainage[28], it still remains to be seen how longer-lasting alterations in ICP affect the meningeal lymphatic vasculature. It is possible that preventing exorbitant rises in ICP after brain injury may allow for a more succinct immune response due to more efficient drainage of neurotoxic DAMPs from the CNS.

The mechanism of how CSF is taken up into meningeal lymphatics and removed from the CNS is still widely debated. Before the recent characterization of the meningeal lymphatic system, it was thought that arachnoid granulations were the main mechanism

through which CSF was removed from the brain[63]. The arachnoid granulations are areas of the arachnoid layer in the meninges that project through the meningeal layers into the venous sinuses, where they are thought to absorb CSF. However, evidence for arachnoid granulations in mice has been scarce[63]. With the recent findings that the meningeal lymphatic system is capable of transporting CSF to the periphery, the term hotspots was coined to describe areas along this lymphatic network in which CSF appears to be taken more readily into the lymphatic pathways. These hotspots have been reported both in the dorsal and basal meningeal lymphatic network[17,20]. How these hotspots take up CSF and solutes from the CNS is still not clear. It has been proposed that the lymphatics may send extensions into the sub-arachnoid space where CSF can be taken up, but this has not been further substantiated[17]. Further research is needed to understand the anatomy of the lymphatic vasculature at the hotspots and to determine which macromolecules and cells are capable of trafficking through the meningeal lymphatic vasculature in homeostasis and various disease states. Furthermore, additional studies are also needed to delineate the contributions of the recently identified basal CNS lymphatics in TBI pathogenesis[20].

Because clearance of ISF containing solutes and macromolecules from the brain parenchyma relies on perivascular routes, termed the glymphatic system[59,64–66], we anticipate that changes in ICP would also affect glymphatic function. Indeed, our findings are consistent with data showing that the glymphatic system is impaired after TBI[59,67]. The glymphatic system and lymphatic system are inherently linked[64,68]. The glymphatic system is responsible for transport of ISF to the CSF surrounding the brain, and the lymphatic system takes up CSF/ISF for transport into the periphery[59,66,68]. While decreased ISF transport from the parenchyma to the sub-arachnoid space may lead to decreased lymphatic drainage[59,65], it has also been shown that impaired lymphatic drainage decreases recirculation of macromolecules through the glymphatic route[18]. Our findings indicate that, in addition to the previously described glymphatic dysfunction, there is also lymphatic dysfunction after TBI. By directly targeting the meningeal lymphatic vasculature through pharmacologic photoablation in our studies, we highlight that pre-existing lymphatic dysfunction results in increased TBI-mediated neuroinflammation and cognitive dysfunction. In addition, injections into the cisterna magna, as was done in our studies, largely bypass the need for glymphatic clearance, as the meningeal lymphatic network takes up CSF directly from this compartment. Therefore, impairment in both systems likely coalesce to cause defective clearance of toxins, protein aggregates, and macromolecules from the brain after TBI. Preventing the rapid rise in ICP seen after TBI may provide a route in which to address both the glymphatic and lymphatic dysfunction that persists after brain trauma; however, future studies are needed to definitively test this.

Interestingly, TBI has been strongly linked to an increased risk of developing numerous other neurological disorders later in life including CTE, Alzheimer's disease, amyotrophic lateral sclerosis, and various psychiatric disorders[2–6]. Our understanding of how brain trauma contributes to the development of these other neurological disorders at the mechanistic level is currently limited. Like TBI, the majority of these CNS disorders are also characterized by neuroinflammation and impaired clearance of DAMPs (e.g., protein aggregates and neurotoxic debris). One could envision that TBI-induced disruptions in meningeal lymphatic function and the resulting buildup of DAMPs in the brain could set off a series of events that ultimately lead to other forms of neurological disease down the road, although future studies are needed to formally test this hypothesis. One approach to test this hypothesis is to investigate whether rejuvenation of meningeal

lymphatic function after TBI limits the risk of disease sequelae later in life. As a proof of principle, we show here that rejuvenating meningeal lymphatic vasculature in aged mice with viral delivery of VEGF-C is effective in preventing excessive Iba1 gliosis following brain trauma.

For reasons that remain poorly understood, sustaining a second head injury before the brain has recuperated from prior head trauma can have devastating consequences. Furthermore, it has been shown that repetitive TBIs result in more serious long-term outcomes when compared to a single TBI[37,38,41,69,70]. Indeed, recent reports in the scientific literature and media have highlighted several high-profile cases of repetitive TBI and its devastating consequences that can include CTE and suicide[34,41,71]. Improved understanding of what makes the injured brain more vulnerable to more severe pathology and neurological demise following secondary head trauma will lead to improved treatment practices. Our findings presented in this paper suggest that disruptions in meningeal lymphatic function may contribute at some level to the more severe neuroinflammation and neurological dysfunction commonly observed in repetitive TBI. The lack of defined guidelines for when individuals can safely return to high-risk activities following TBI is a significant problem for caregivers. Our work suggests that evaluating meningeal lymphatic drainage recovery post-injury might provide clinicians with a much-needed empirical test to inform athletes and military personnel of when it is safe to return to action. However, improved diagnostics must first be developed in order to accurately measure meningeal lymphatic function in humans.

While our studies on pre-existing lymphatic dysfunction before TBI provide important insights into how lymphatic dysfunction may contribute to a higher neuroinflammatory state, additional studies are required to more fully understand the role that each CNS-resident cell population is playing in this inflammatory environment. While we see an increase in GFAP immunoreactivity in the brain after Ablation + TBI as compared to other control groups two weeks after injury (Fig. 6a-b), this same change is not apparent in the sequencing data at one week post injury. We would expect the inflammatory environment to be dynamically changing throughout this time and believe that following up on these studies with other techniques including single cell RNA sequencing would be incredibly valuable to understand how cells such as microglia and astrocytes are changing in response to injury, and how they might be influencing the inflammatory environment after TBI.

TBI is an especially serious threat to health in the elderly, where it is a leading cause of death and disability[35,36,39]. Even though the elderly only account for 10% of all TBI cases, over 50% of all TBI-related death occurs in individuals over the age of 65[72]. It has been extensively shown that similar injuries result in more severe pathology and neurological impairment in the elderly than in other age groups[32,39]; however, the cause for this is currently not well understood. Interestingly, several recent studies have shown that the meningeal lymphatics are impaired with aging[18,20,42]. Moreover, boosting lymphatic function with VEGF-C treatment can mitigate the cognitive deficits seen in aged mice[17]. In our studies presented here, we show that photoablation of meningeal lymphatic vessels before head injury leads to more severe neuroinflammatory outcomes and decreased performance in cognitive tests following TBI. Therefore, it is feasible that aging-associated deterioration of CNS lymphatic function may contribute at some level to the especially devastating consequences of TBI in the elderly. Indeed, our data demonstrating that viral delivery of VEGF-C to aged mice improves neuroinflammatory outcomes after TBI provides further evidence that the lymphatic system may be involved in the particularly devastating

outcomes the elderly experience after TBI. Furthermore, it suggests that boosting meningeal lymphatic drainage may serve as a viable therapeutic option to limit TBI pathogenesis.

Overall, the work described here provides insights into how the meningeal lymphatic system is impacted by TBI and also how pre-existing defects in this drainage system can predispose the brain to exacerbated neuroinflammation and cognitive outcomes following brain injury. We show that even mild forms of TBI can result in pronounced defects in meningeal lymphatic function that can last for weeks post-injury. Mechanistically, we demonstrate that closed-skull TBI is associated with elevated ICP and that this can contribute to disruptions in meningeal lymphatic drainage function. Finally, we provide evidence that boosting meningeal lymphatic function through delivery of VEGF-C may serve to decrease neuroinflammation after TBI. Importantly, improved understanding of the contributions of the meningeal lymphatic system in brain injury and recovery may help provide opportunities for therapeutic approaches to treat TBI.

## Methods

**Mice.** All mouse experiments were performed in accordance with the relevant guidelines and regulations of the University of Virginia and approved by the University of Virginia Animal Care and Use Committee. C57BL/6J mice were obtained from Jackson Laboratories. Mice were housed and behavior was conducted in specific pathogen-free conditions under standard 12-h-light/dark cycle conditions in rooms equipped with control for temperature ($21 \pm 1.5\,°C$) and humidity ($50 \pm 10\%$). Mice matched for sex and age were assigned to experimental groups and all adult mice used were between 8 and 10 weeks of age. Males and females were used for drainage and lymphangiogenesis studies, as sex has not been shown to influence lymphatic drainage at baseline[18]. Both males and females were also used to study the effects of increased ICP on lymphatic flow. Males were used for all pre-existing lymphatic dysfunction studies (Figs. 4–6) for consistency with behavioral readouts and for consistency within the RNA sequencing data, as sex can influence both of these readouts. Male mice were also used for the aged mice experiments. All aged mice (used for experiments in Fig. 7 and Supplementary Fig. 8) were between 18 and 24 months of age and were obtained from Jackson Laboratories and the National Institute on Aging (NIA) Aged Rodent Colonies.

**Traumatic brain injury.** This injury paradigm was adapted from the published Hit and Run model[58]. Mice were anesthetized by 4% isoflurane with 0.3 kPa $O_2$ for 2 min and then the right preauricular area was shaved. The mouse was placed prone on an $8 \times 4 \times 4$-inch foam bed (type E bedding, open-cell flexible polyurethane foam with a density of ~0.86 pounds per cubic feet and a spring constant of ~4.0 N/m purchased from Foam to Size, Ashland VA) with its nose in a nosecone delivering 1.5% isoflurane. The head was otherwise unsecured. The device used to deliver TBI was a Controlled Cortical Impact Device (Leica Biosystems, 39463920). A 3 mm impact probe was attached to the impactor device which was secured to a stereotaxic frame and positioned at 45 degrees from vertical. In this study, we used a strike depth of 2 mm, 0.1 s of contact time and an impact velocity of 5.2 m (m) per second (s). An impact velocity of 6.2 m/s was used for the TBI severity studies in Supplementary Fig. 3. The impactor was positioned at the posterior corner of the eye, moved 3 mm towards the ear and adjusted to the specified depth using the stereotaxic frame. A cotton swab was used to apply water to the injury site and the tail in order to establish contact sensing. To induce TBI, the impactor was retracted and dispensed once correctly positioned. The impact was delivered to the piriform region of the brain. Following impact, the mouse was placed supine on a heating pad and allowed to regain consciousness. After anesthesia induction, the delivery of the injuries took less than 1 min. The time until the mouse returned to the prone position was recorded as the righting time. Upon resuming the prone position, mice were returned to their home cages to recover on a heating pad for six hours with soft food. For sham procedures, mice were anesthetized by 4% isoflurane with 0.3 kPa $O_2$ for 2 min and then the right preauricular area was shaved. The mouse was placed prone on a foam bed with its nose secured in a nosecone delivering 1.5% isoflurane. The impactor was positioned at the posterior corner of the eye, moved 3 mm towards the ear and adjusted to the specified depth using the stereotaxic frame. A cotton swab was used to apply water to the injury site and the tail in order to establish contact sensing. Then, the impactor was adjusted to a height where no impact would occur, and was retracted and dispensed. Following the sham procedure, the mouse was placed supine on a heating pad and allowed to regain consciousness. Mice were allowed to recover on the heating pad in their home cages for 6 h with soft food before being returned to the housing facilities.

**Intra-cisterna magna injections.** Mice were anaesthetized by intraperitoneal (i.p.) injection of a mixed solution of ketamine (100 mg/kg) and xylazine (10 mg/kg) in sterile saline. The skin of the neck was shaved and cleaned with iodine and 70%

ethanol, and ophthalmic solution (Puralube Vet Ointment, Dechra) was placed on the eyes to prevent drying. The head of the mouse was secured in a stereotaxic frame and an incision in the skin was made at midline. The muscle layers were retracted and the cisterna magna exposed. Using a Hamilton syringe (coupled to a 33-gauge needle), the volume of the desired solution was injected into the CSF-filled cisterna magna compartment. For the bead experiments, 2 µl of FluoSpheres carboxylate 0.5 µm-beads 505/515 (Invitrogen, F8813) in artificial CSF (597316, Harvard Apparatus UK) were injected at a rate of 2 µl/min. For Visudyne experiments, 5 µl of Visudyne (verteporforin for injection, Valeant Ophthalmics) was injected at a rate of 2.5 µl/min. For Lyve-1 labeling experiments, 2 µl of anti-mouse Lyve-1 488 (Invitrogen, 53044382, undiluted) was injected at a rate of 2 µl/min. For experiments with aged mice, 2 µl of artificial CSF containing $10^{13}$ genome copies per ml of AAV1-CMV-mVEGF-C, or control AAV1-CMV-eGFP (AAV1, adeno-associated virus serotype 1; CMV, cytomegalovirus promoter; eGFP, enhanced green fluorescent protein; purchased from Vector BioLabs, Philadelphia), were injected into the cisterna magna CSF at a rate of 2 µl/min. The needle was inserted into the cisterna magna through retracted muscle in order to prevent backflow upon needle removal. The neck skin was then sutured, after which the mice were subcutaneously injected with ketoprofen (1 mg/kg) and allowed to recover on a heating pad until fully awake.

**Pharmacologic meningeal lymphatic vessel ablation.** Visudyne treatment was adapted from published protocols[17,18,43]. Selective ablation of the meningeal lymphatic vessels was achieved by i.c.m. injection and transcranial photoconversion of Visudyne (verteporfin for injection, Valeant Ophthalmics). Visudyne was reconstituted following the manufacturer's instructions and 5 µl was injected i.c.m. following the procedure described above in the intra-cisterna magna injections methods section. After 15 min, a midline incision was created in the skin to expose the skull and visudyne was photoconverted by pointing a 689-nm-wavelength non-thermal red light (Coherent Opal Photoactivator, Lumenis) to five different locations above the intact skull (1 at the injection site, 1 at the SSS, 1 at the confluence of the sinuses and 2 at the TSs). This experimental group is labeled as Ablated or Visudyne + laser. Each location was irradiated with a light dose of 50 J/cm² at an intensity of 600 mW/cm² for a total of 83 s. Controls were injected with the same volume of Visudyne (without the photoconversion step; labeled as Visudyne or Not Ablated) or sterile saline plus laser treatment (labeled as Vehicle + laser). The scalp skin was then sutured, after which the mice were subcutaneously injected with ketoprofen (1 mg/kg) and allowed to recover on a heating pad until fully awake.

**ICP measurements.** ICP was measured according to published protocols[17,18]. Mice were anaesthetized by i.p. injection with ketamine (100 mg/kg) and xylazine (10 mg/kg) in saline and the skin was incised to expose the skull. A 0.5-mm diameter hole was drilled in the skull above the left parietal lobe. Using a stereotaxic frame, a pressure sensor catheter (model SPR100, Millar) was inserted perpendicularly into the cortex at a depth of 1 mm. To record changes in ICP, the pressure sensor was connected to the PCU-2000 pressure control unit (Millar). For measurements in mice after TBI (30 min, 2 h, 6 h, 24 h, 3 day, 4 day, and 1 week post injury) or after jugular venous ligation (3 and 24 h) ICP was recorded for 6 min after stabilization of the signal and the average pressure was calculated over the last 3 min of recording. Mice were euthanized following the procedure.

**Jugular venous ligation.** Mice were anaesthetized by i.p. injection with ketamine (100 mg/kg) and xylazine (10 mg/kg) in saline. The left and right preauricular area and the skin between the ears was shaved and prepped with iodine and 70% ethanol. Ophthalmic ointment (Puralube Vet Ointment, Dechra) was applied to the eyes to prevent drying. The mouse was secured onto a surgical plane in the lateral position and a lateral incision was created between the two mouse ears. The incision site was retracted to reveal the left temporalis muscle. The left temporalis muscle was retracted to reveal the infratemporal fossa, where the left internal jugular vein can be identified. The left internal jugular vein was ligated using 8-0 Nylon Suture (AD surgical, XXS-N808T6), and then the same procedure was performed on the opposite side to ligate the right jugular vein. The incision was then sutured and the mice were subcutaneously injected with ketoprofen (1 mg/kg) and were allowed to recover on the heating pad until awake. Sham mice received the incision and the jugular veins were exposed bilaterally, but they did not undergo ligation. The ICP on these mice was recorded as described in the ICP measurements methods section 3 and 24 h after ligation.

**RNA extraction and sequencing.** For RNA extraction, the brain hemisphere ipsilateral to the brain injury was harvested, the cerebellum and olfactory bulbs were removed, and the hemisphere was immediately snap-frozen in dry ice and stored at −80 °C until further use. After defrosting on ice, samples were mechanically homogenized in 500 µl extraction buffer comprised of T-PER Tissue Protein Extraction Reagent (78510, Thermo Scientific) supplemented with Phos-STOP phosphatase inhibitor cocktail (04906845001, Roche) and cOmplete protease inhibitor cocktail (11873580001, Roche). 100 µl of the homogenate was transferred to a tube containing 1.1 ml TRIzol Reagent (15596018, Life Technologies) and vortexed. Two hundred microliters of chloroform (BP1145-1, Fisher Scientific) were added to the samples, and samples were vortexed and allowed to incubate for

5 min at room temperature. Samples were then spun at 18,400 RCF at 4 °C for 15 min. The top aqueous phase was transferred into a new Eppendorf tube and 300 μl isopropanol (I9516, Sigma) were added, vortexed, and allowed to incubate at room temperature for 10 min. Samples were then spun down at 13,500 RCF at 4 °C for 12 min. The RNA pellet was washed two times with 70% ethanol and resuspended in DNAse/RNAse free water. Sample quality and RNA concentration were assessed using the NanoDrop 2000 Spectrophotometer (Thermo Scientific) and samples were frozen at −80 °C until further use. For sequencing, total RNA samples were sent to GENEWIZ for library preparation and paired end sequencing.

**RNA-seq analysis**. The raw sequencing reads (FASTQ files) were aligned to the UCSC mm10 mouse genome build using the splice-aware read aligner HISAT2[73]. Samtools was used for quality control filtering[74]. Reads were sorted into feature counts with HTSeq[75]. DESeq2 was used to normalize the raw counts based on read depth and perform principal component analysis and differential expression analysis[76]. The p-values were corrected with the Benjamini–Hochberg procedure to limit false positives arising from multiple testing. The gene set collections from MSigDB were used for differential gene set enrichment analysis[47]. The analysis itself was performed using the Seq2Pathway, fgsea, tidyverse, and dplyr software packages. Heatmaps were generated using the pheatmap R package [https://github.com/raivokolde/pheatmap] while other plots were made with the lattice (http://lattice.r-forge.r-project.org/) or ggplot2 [https://ggplot2.tidyverse.org] packages. The GWAS Catalog was used to find genes associated with neurodegenerative or psychiatric diseases [https://www.ebi.ac.uk/gwas/home][53], and the circos plot including these data was generated using the circlize R package[77]. All code used for analysis is available at [https://github.com/arun-b-dutta/TBI_Lymphatics_RNAseq-Analysis]. Raw and processed sequencing data can be accessed through the Gene Expression Omnibus (GEO) at [https://www.ncbi.nlm.nih.gov/geo/query/acc.cgi?acc=GSE155063].

**Behavioral testing**. All behavioral experiments were carried out during daylight hours (except the NLRT, which was performed at 5:00 PM) in a blinded fashion.

**Gross neuroscore**. The gross neuroscore was performed following a published protocol with modifications[78]. Briefly, mice performed 10 individual tasks to assess behaviors including seeking/exploring tendencies, the startle reflex, and balance/motor coordination. The ability to cross different width beams, to react to a loud noise, to balance on a beam, and to explore the surroundings were assessed and scored by a blinded experimenter 1 h after TBI. If the mouse was able to adequately perform the task, a score of 0 was given. If the mouse failed to adequately perform the test, a score of 1 was given. Scores for the 10 tasks were summed for a total minimum score of 0 and a total maximum score of 10.

**Rotarod test**. The mice were transported to the behavior room and allowed to habituate for 1 h before each day of testing. The experimental apparatus used in this test contained five separate compartments on a rotating rod to accommodate five mice per trial (MED Associates Inc, ENV-575M). The rod was programmed to turn starting at 4 rotations per minute (rpm) and to accelerate to 40 rpm over a span of 5 min. Each mouse was placed on the rod and allowed to ambulate until it either fell off, hung without effort on the rod for a total of 5 rotations, or reached the trial endpoint (6 min). When a mouse fell from the rotarod, it disrupted a laser sensor to stop recording. The time spent on the rod and the speed at which the mouse fell or the trial ended was recorded (RotaRod Version 1.4.1, MED associates inc). Three trials were performed each day for 3 days. The 3 trials per day were averaged, and latency to fall and percent performance increase were calculated based off of the average time of trial per mouse per day.

**Novel location recognition test (NLRT)**. The NLRT was performed following a published protocol with modifications[18]. The mice were transported to the behavior room and allowed to habituate for 1 h before each trial of the test. The experimental apparatus used in this study was a square box made of opaque white plastic (35 cm × 35 cm). The mice were first habituated to the square apparatus for 10 min by allowing for free exploration within the open field. Eight hours later, after 5:00PM, two identical objects were then positioned in the two far corners of the arena at distances of 5 cm away from the adjacent arena wall (familiar locations). Mice were then placed in the arena facing the wall furthest away from the objects and allowed to explore the arena and objects for 10 min. Time spent investigating the objects was measured and was considered the training phase of the test. After 24 h, the mice were placed in the same box but one object was moved down to a diagonal position (novel location). The time spent exploring the objects in the familiar or novel location was measured for 10 min and was considered the test phase. Exploration of an object was recorded when the mouse approached an object and touched it with its vibrissae, snout, or forepaws and was measured using a video tracking software (Noldus Ethovision XT). The preference for either the novel or familiar object was calculated as the percent of time the mouse spent with one object divided by the total time the mouse spent investigating either object.

**Tissue collection**. Mice were euthanized with $CO_2$ and then transcardially perfused with 20 ml 1x PBS. dCLNs were dissected and drop-fixed in 4% paraformaldehyde (PFA) for 2 h at 4 °C and then the CUBIC clearance protocol was performed as described below in the dCLN clearance methods sections[79]. For meningeal whole-mount collection, skin and muscle were stripped from the outer skull and the skullcap was removed with surgical scissors and fixed in 4% PFA for 12 h at 4 °C. Then the meninges (dura mater and arachnoid mater) were carefully dissected from the skullcaps with Dumont #5 forceps (Fine Science Tools). Meningeal whole-mounts were then moved to PBS and 0.05% azide at 4 °C until further use. Brains were removed and kept in 4% PFA for 24 h and cryoprotected with 30% sucrose for 3 days. A 4 mm coronal section of brain tissue that surrounded the site of the lesion was removed using a brain sectioning device and then frozen in Tissue-Plus OCT compound (Thermo Fisher Scientific). Fixed and frozen brains were sliced (50-μm thick sections) with a cryostat (Leica) and kept in PBS + 0.05% azide at 4 °C until further use.

**dCLN clearance**. dCLN clearance was performed following the published CUBIC protocol with modifications[79]. In brief, nodes were incubated in 50% reagent 1 (prepared 1:1 with dH2O) for 1 day at 37 °C, shaking, with DAPI (1:1000). Nodes were then transferred to reagent 1 for 1 day at 37 °C, shaking, with DAPI (1:1000). Nodes were washed two times in PBS + 0.01% sodium azide for 2 h and overnight with DAPI (1:1000) at 37 °C. Then nodes were incubated with 50% reagent 2 (prepared 1:1 with dH2O) for 1 day at 37 °C with DAPI (1:1000). Finally, nodes were incubated with reagent 2 for 1 day at 37 °C. Nodes were placed in eight well chambers (155411, Thermo Fisher) with mineral oil and imaged with confocal microscopy.

**Immunohistochemistry, imaging, and quantification**. For immunofluorescence staining, floating brain sections and meningeal whole-mounts in PBS and 0.05% azide were blocked with either 2% donkey serum or 2% goat serum, 1% bovine serum albumin, 0.1% triton, 0.05% tween-20, and 0.05% sodium azide in PBS for 1.5 h at room temperature. This blocking step was followed by incubation with appropriate dilutions of primary antibodies: anti-Lyve-1–eFluor 660 & eFluor 488 (eBioscience, clone ALY7, 1:200), anti-CD31 (Millipore Sigma, MAB1398Z, clone 2H8, 1:200), anti-Iba1 (Abcam, ab5076, 1:300) and anti-GFAP (Thermo Fisher Scientific, 2.2B10, 1:1000) in the same solution used for blocking overnight at 4 °C or for 3 h at room temperature. Meningeal whole-mounts or brain tissue sections were then washed three times for 10 min at room temperature in PBS and 0.05% tween-20, followed by incubation with the appropriate goat or donkey Alexa Fluor 594 or 647 anti-rat, -goat (Thermo Fisher Scientific, 1:1000) or -Armenian hamster (Jackson ImmunoResearch, 1:1000) IgG antibodies for 2 h at RT in the same solution used for blocking. The sections or whole-mounts were then washed 3 times for 10 min at RT before incubation for 10 min with 1:1000 DAPI in PBS. The tissue was then transferred to PBS and mounted with ProLong Gold antifade reagent (Invitrogen, P36930) on glass slides with coverslips. Slide preparations were stored at 4 °C and imaged using a Lecia TCS SP8 confocal microscope and LAS AF software (Leica Microsystems) within one week of staining. Quantitative analysis of the acquired images was performed using Fiji software. For the assessment of gliosis in the injured and uninjured brains, two representative brain sections from the site of the lesion (approximately −0.74 to 0 bregma) or the corresponding area in sham animals were fully imaged and at least 5 animals were included per experimental group. The full brain section was adjusted for brightness/contrast uniformly for each experiment (Experiments in Fig. 6a–c: b/c- GFAP-32/190, b/c- Iba1-51/158, experiments in Fig. 7d-h: b/c- GFAP- 41/224, b/c- Iba1-40/174), and the percent area of coverage of each immunohistochemical markers was calculated per hemisphere for each brain section. Each hemisphere was traced, and then the threshold was uniformly set for each experiment to select for stained cells (Experiments in Fig. 6a–c: thresh- GFAP-90/255, b/c- Iba1-115/255, Experiments in Fig. 7d-h: b/c- GFAP- 115/255, b/c- Iba1-139/255). The mean percent area fraction was calculated using Microsoft Excel. For Supplementary Fig. 1, high magnification images (×20 and ×63) were taken directly adjacent to the site of the injury and the entire hemisphere ipsilateral to the injury was quantified for the levels of gliosis at each timepoint as specified above. For lymph nodes, the percent volume of microbead coverage in cleared dCLN was assessed by creating a 3D reconstruction of the node and then calculating the volume covered by beads divided by the total volume of the node using Fiji. The right and the left dCLN percent volume were averaged together for each mouse. For assessment of meningeal lymphatic vessel coverage and complexity, images of meningeal whole-mounts were acquired using a confocal microscope and Fiji was used for quantifications. The entire meningeal whole-mount (overlying both the injured and uninjured hemisphere) was traced and used for quantification. The percent area coverage of Lyve-1 was used to determine the coverage of the lymphatic vessels. When applicable, the same images were used to assess the percentage of field coverage by Lyve-1⁻CD31⁺ vessels. To assess lymphatic vessel diameter, 70 measurements per meningeal whole-mount were taken by a blinded experimenter (40 along the TS and 30 along the SSS) and were averaged together. Lymphatic vessel loops and sprouts were calculated along both the TS and SSS by two independent blinded experimenters. Areas along the edge of the TSs were excluded for sprout quantification to avoid counting any lymphatic ends created by removal of the whole-mount. All meningeal whole mounts used for quantification of

lymphatic morphology were imaged with identical confocal settings and Fiji parameters (b/c-Lyve-1-18/149, thresh-Lyve-1-102/255). For microglia morphology analysis, high magnification (63x) images of microglia were taken and the filament function in Imaris software (9.5.1 Bitplane) was used to identify Iba1+ cells with their dendritic processes. Three fields of view in the peri-lesional area were imaged for each section, and two sections were imaged for each mouse. The microglia parameters for each section and field of view (cell number, filament dendrite length, filament dendrite volume, and number of dendritic branch points) were averaged together and plotted as one point per mouse. For the Sholl analysis, 4 microglia per field of view were quantified for five mice.

**Flow cytometry**. Mice were euthanized and blood was collected with cardiac puncture. Red blood cells were lysed, and then cells were then centrifuged for 5 min at 400 RCF and resuspended in 200 μl FACS buffer (pH 7.4; 0.1 M PBS; 1 mM EDTA, and 1% BSA). Fluorescence data were collected with a Gallios (Beckman Coulter), then analyzed using FlowJo software (Treestar). Single cells were gated using the height, area, and pulse-width of the forward and side scatter. Beads were gated based on their size and fluorescence.

**eGFP or VEGF-C AAV delivery**. For experiments using viral-mediated expression of mVEGF-C to enhance meningeal lymphatic vasculature growth, 2 μl of artificial CSF containing $10^{13}$ genome copies per ml of AAV1-CMV-mVEGF-C or control AAV1-CMV-eGFP (AAV1, adeno-associated virus serotype 1; CMV, cytomegalovirus promoter; eGFP, enhanced green fluorescent protein; purchased from Vector BioLabs, Philadelphia) were injected into the cisterna magna CSF at a rate of 2 μl/min, following the procedure described in the intra-cisterna magna injections methods section.

**Statistical analysis and reproducibility**. Sample sizes were chosen on the basis of standard power calculations (with $\alpha = 0.05$ and power of 0.8). Experimenters were blinded to the identity of experimental groups from the time of euthanasia until the end of data collection and analysis. One-way ANOVA, with Bonferroni's multiple comparison test, Tukey's multiple comparison test, Dunnett's multiple comparison test, or Holm-Sidak's post hoc test, was used to compare multiple independent groups. Two-group comparisons were made using unpaired Student's t-test. For comparisons of multiple factors (for example, age versus treatment), two-way ANOVA with Bonferroni's post hoc test or Tukey's multiple comparison test was used. Repeated-measures two-way ANOVA with Bonferroni's post hoc test was used for day versus treatment comparisons with repeated observations. Statistical analysis (data are always presented as mean ± s.e.m.) was performed using Prism 8.0 (GraphPad Software, Inc.).

**Reporting summary**. Further information on research design is available in the Nature Research Reporting Summary linked to this article.

## Data availability

All data and genetic material used for this paper are available from the authors on request. Raw and processed sequencing data can be accessed through the Gene Expression Omnibus (GEO) at [https://www.ncbi.nlm.nih.gov/geo/query/acc.cgi?acc=GSE155063]. Source data for all figures are provided with this paper.

## Code availability

All code used for analysis is available at [https://github.com/arun-b-dutta/TBI_Lymphatics_RNAseq-Analysis].

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

## Acknowledgements

We thank members of the Lukens lab and the Center for Brain Immunology and Glia (BIG) for valuable discussions. Graphical illustrations in Figs. 1a, e, 6h, and Supplementary Fig. 5a were made using BioRender (https://biorender.com/). This work was supported by The National Institutes of Health/National Institute of Neurological Disorders and Stroke (R01NS106383; awarded to J.R.L.), The Alzheimer's Association (AARG-18-566113; awarded to J.R.L.), The Owens Family Foundation (Awarded to J.R.L.), and The University of Virginia Research and Development Award (Awarded to J.R.L.). A.C.B., A.B.D., and M.A.K. were supported by a Medical Scientist Training Program Grant (5T32GM007267-38). A.C.B. and M.A.K. were supported by an Immunology Training Grant (5T32AI007496-25). A.C.B. was supported by a Wagner Fellowship. A.B.D. was supported by a Biomedical Data Sciences Training Grant (T32LM012416). H.E.E. was supported by a Cell and Molecular Biology Training Grant (T32GM008136). B.H.N. was supported by an R35 grant (5R35GM128635-02). C.R.L. was supported by a NIH National Institute of General Medical Sciences predoctoral training grant (3T32GM008328) and a Wagner Fellowship. E.L.F. was supported by a National Multiple Sclerosis Foundation Postdoctoral Fellowship (FG-1707-28590).

## Author contributions

A.C.B., J.K, and J.R.L. designed the study; A.C.B., M.E.H., I.S., E.L.F., C.R.L., C.A.M., M.A.K., H.E.E., D.S., and J.R.L. performed experiments. B.H.N. contributed to data analysis. A.C.B., A.B.D., and J.R.L. analyzed data and wrote the manuscript; J.R.L. oversaw the project.

## Competing interests

Competing interests: J.K. is an advisor to PureTech Health/Ariya. All other authors declare no competing interests.
