## [Peer Review File · Nature Communications]

Reviewers' Comments:

Reviewer #1:

Remarks to the Author:

In this manuscript, "Meningeal lymphatic dysfunction exacerbates traumatic brain injury pathogenesis" by Bolte et al., the authors explore how the meningeal lymphatics are impacted by a single mild traumatic brain injury (TBI). In this interesting study, the authors analyze the impact of TBI in the deep cervical lymph nodes, lymphatic vasculature (lyve1 +), gliosis, and expression of the inflammatory, neurodegenerative and complement genes. Also, they show how the pharmacological photoablation of the meningeal lymphatics during the first hours post-TBI leads to aggravating the neuropathology. An important question that remains unsolved is whether meningeal lymphatic dysfunction can be restored days or weeks post-TBI. The functions of uptake CSF, meningeal lymphatic dysfunction, or expression of inflammatory genes were observed in the first 2 to 24 hours after TBI. Still, the greatest detrimental consequences occur weeks or months after mild TBI. It would be very interesting to observe this phenomenon in later time points post-injury and at different severities. The design of therapeutic tools for this mTBI model, increasing the severity, could be incorporated to determine the meningeal lymphatic dysfunction at a chronic stage. The effect of chronic and repetitive TBI on the meningeal lymphatic dysfunction would be highly relevant for the field. The techniques used to study the morphology of the meningeal lymphatic vessels are appropriately applied. However, it is not clear whether the results of RNA sequencing are affected by pharmacological ablation of the vessels or by the effect of the impact of the injury. While altogether this study represents a comprehensive analysis, the authors should provide additional justification/context to the pathological consequences of TBI, and specifically with respect to meningeal dysfunction. A revised version of this manuscript including the incorporation of a model of the severity of TBI and a more in-depth neuropathological analysis would significantly increase the likelihood of this study having a high impact in the TBI field.

Major comments:

- In the Material and Methods section, the authors describe that mice are matched for sex and age, and were randomly assigned into experimental groups, however, the exact mice that were used in each experiment are not described in the text nor the legends. Both sex and age can regulate the inflammatory and cerebrovascular responses to brain injuries. The combination of mice into the experimental groups may be induced disparity in the results. Analyzing the groups by sex and age separately is necessary, or at least it should be specified in each experiment which animals were used.
- It is not clear how the quantitative analysis of the immunohistochemistry images was performed. The specific regions that were quantified, the software used, and the parameters used should all be clearly defined. Microglia activation and the morphological analysis for microglia/macrophages should be incorporated to establish a correlation with the quantitative analysis of Sup. Fig 1.
- It is well known that the severity of the TBI can be relevant to determining the pathology, as well as in the case of mTBI, the repetitive hits or the intensity of the parameters that control the impact can aggravate the damage of the meningeal lymphatics and their functioning. Is the meningeal lymphatic drainage function severity-dependent? Does it occur an irreversible damage of the meningeal lymphatic vessels at long term post-TBI?
- This study shows that a single mTBI causes impairment in the meningeal lymphatic drainage function up to two weeks post-TBI, although it was seen as an indicator of recovery (Fig. 1h). Is the function of the meningeal lymphatic restored at some time post-TBI, or is it permanently damaged? The inclusion of an experimental group of animals to study the effect of TBI at 2- or 3-months post-injury will clarify if the meningeal lymphatic dysfunction is really time reversible.

- When analyzing the presence of Lyve-1-expressing lymphatic vessels in sham mice and in TBI mice, it is only decreased at 2 hours post-TBI, but these alterations are not observed at 24 h. Is the uptake of CSF restored after 24 h post-TBI?
- The ICP by itself produces meningeal lymphatic dysfunction at 2 hours after being induced, and it is assumed that the increase of ICP after TBI is the main inducer of the lymphatic damage. What role does the inflammatory response triggered in the meningeal lymphatic dysfunction play? It would be helpful to use pharmacological inhibition of the microglia activation to see if it influences the meningeal lymphatic vessel dysfunction.
- The effect that preexisting meningeal lymphatic dysfunction caused by Visudyne photoablation was observed as mild TBI, and it increased the levels of GFAP expression and IL-6. What happens to the activation of the microglia? What regions were specifically measured? The areas circled in Figure 4a vary from the Piriform cortex to the primary somatosensory cortex, where the expression of GFAP varies greatly depending on the focal area of the impact. All of these details are not fully described in this study, and the immunohistochemical analysis should be clarified and carried out accordingly. Also, what happens in sham animals with or without Visudyne administration?
- RNA sequencing results at 24 h post-surgery, is it unclear whether the inflammatory, neuronal damage, disease-associated microglia, or complement regulatory genes (Fig. 4h, and Fig. 5) are upregulated only in the TBI + ablation group or also in the Sham + ablation group? At least this similarity between the two groups is observed in Fig 4d. This would determine whether it is really the effect of the TBI that modifies these genes or if it is the consequence of the pharmacological ablation of the lymphatic vessels. This same reasoning could be applied to the results in Figure 5.
- It would be extremely interesting to investigate in more detail if the meningeal lymphatic dysfunction affects the neuropathology and immune-inflammatory response of TBI. What happens to the activation, shape, and morphology of the microglia in the ablated + TBI group compared to the non-ablated + TBI group and in the sham brains? How does the ablation of lymphatic vessels affect neuronal or axonal damage? Does the meningeal lymphatic ablation affect to the T cell priming and recruitment after TBI?

Minor comments:

- Please, check duplications, misspellings and use the acronyms where appropriate, e.g., DAPI counterstain (1: 1000), traumatic brain injury, hrs or hours,... or types: "exisitng."
- It is not clear why the time of 2 h is replaced by 3 h post-TBI.
- A more detailed description of the type of BIT produced in terms of neuropathology should be reported or indicate the reference of the same model already

Reviewer #2:

Remarks to the Author:

These interesting pre-clinical studies led by Ashley Bolte and John Lukens at University of Virginia provide new insight into mild traumatic brain injury (mTBI) pathogenesis by demonstrating that mTBI causes deficits in meningeal lymphatic drainage within hours of brain trauma in mice. The mTBI-induced impairments in lymphatic function and blocked drainage of macromolecules are temporally linked to raised ICP levels, which is a major driver of mortality and poorer neurological outcomes in head injury patients. Using elegant photoablation models the authors also demonstrate that existing meningeal lymphatic dysfunction predisposes the brain to increased

complement signaling which is associated with exacerbated neuroinflammation during subsequent mTBI exposure. As such, these new studies provide new and important insight into meningeal lymphatic functional impairment during mTBI in mice, and provide clues about microglial-mediated neuroinflammatory mechanisms that may lead to delayed cognitive impairments following repetitive mTBI.

Importantly, the closed head mTBI model has been carefully developed to avoid possible confounding results due to direct impact on lymphatic vessels. The righting reflex data indicate that injury severity is mild and thus relevant to concussive-like mTBI. Sophisticated image analysis and ablation models are used to demonstrate impairments in meningeal lymphatic function after mTBI or jugular vein ligations to increase ICP levels in brain. Overall, these pre-clinical studies are performed with high degree of scientific rigor, and appropriate controls and analyses are presented.

The studies related to effects of ICP on meningeal lymphatic function are particularly important for acute TBI and suggest that dysfunction within meningeal lymphatics vessels leads to early evolution of secondary injury (neuroinflammation). Also, the identification of the complement pathway as a mediator of post-traumatic neuroinflammation and dysfunction in the mTBI + Visudyne experiments is exciting, and is certainly worthy of further detailed molecular investigation.

The authors should consider the following points identified during peer-review:

1) Does meningeal lymphatic function ever return to sham control levels after single mTBI? What happens months post-mTBI? Because impairments in meningeal lymphatic function slows paravascular influx of macromolecules into the brain after mTBI, and such impairments are associated with cognitive impairments in mice (Da Mesquita 2018, Science PMID: 30046111), what is the long-term impact mTBI on cognitive function in this model? Such an outcome measure is relevant given link between single and repetitive mTBI and chronic neurodegeneration/dementias (CTE, AD...).

2) How do morphological changes in meningeal lymphatic vasculature relate to dysfunction? It might be better to integrate morphological studies in Figure 2 with initial characterization of mTBI-induced meningeal lymphatic dysfunction.

3) Can the authors shed light into what macromolecules or immune cells are blocked when meningeal lymphatic dysfunction occurs after mTBI? How do they relate to the subsequent inflammatory response in brain?

4) A rescue experiment that improves meningeal function and brain perfusion after mTBI would add great value to the opening figures of this study. VEGF-C overexpression or similar strategy should be considered by the authors.

5) Given that mTBI rapidly increases ICP, which is associated impairments in meningeal lymphatic function (nicely demonstrated in figure 3), these pre-clinical studies provide novel insight into secondary injury responses acutely after brain trauma.

6) The RNAseq data identifying the complement pathway as a key inflammatory pathway up-regulated by meningeal lymphatic dysfunction following ablation/mTBI is very interesting because this pathway has been implicated in microglial-mediated synaptic stripping and cognitive decline in age-related neurodegeneration (Hong 2016, Science PMID: 27033548) and neuroinflammatory demyelinating disease (Wernerberg 2019, Immunity PMID: 31883839). These promising findings should be developed further to provide novel insight into DAM microglia-neuronal interactions, structure function analyses of neuroplasticity, and long-term cognitive outcomes. The authors are encouraged to use available transgenic models and physiological methods to pinpoint mechanisms

by which complement signaling and DAM microglia react to post-traumatic meningeal lymphatic functional impairments that disrupt neuronal health and cognitive function.

7) Minor - representative photomicrographs of microglia and astrocytes should be included in supplemental figure 1.

Reviewer #3:

Remarks to the Author:

This manuscript by Bolte and colleagues describes a series of experiments to address a key knowledge gap in brain research - the role of meningeal lymphatic drainage in neurological disease. Specifically, the authors employed an experimental model of traumatic brain injury (TBI) to demonstrate proof-of-concept evidence for meningeal lymphatic dysfunction in this context, as well as some evidence of potential mechanisms. This paper will be of particular importance to the neurotrauma field given recent interest in lymphatics/glymphatics/CSF transport and brain injury biomarkers, across a whole spectrum of TBI severity from mild through to severe. The principle findings provide novel insight into how the meningeal lymphatic system is impacted by TBI, as well as how pre-existing deficits in this system can exacerbate the neuroinflammatory consequences that are characteristic of a brain injury.

The methodology appears sound (i.e. blinded analyses, randomization, inclusion of both sexes, power calculations), with minor exceptions, as follows: (1) What age range of animals was used? This should be reported, particularly in light of how CNS lymphatic drainage function declines with age (as noted by the authors in the discussion). Similarly, TBI neuropathology is influenced by age. (2) For the TBI model, where any analgesics administered? The procedure used to generate 'sham' animals should also be clearly described (e.g. surgical preparation + anesthesia?). (3) Assessment of GFAP/Iba1 glial reactivity appears weak and subject to unintended selection bias (only 2 representative slides imaged and averaged per brain). However, this is somewhat mitigated by most interpretations of this data being supported by additional measures of neuroinflammation (e.g. gene expression analyses).

The strongest finding is the evidence of lymphatic drainage impairments across a time course after TBI (Figure 1), in both the cervical lymph nodes and meninges. Somewhat surprisingly, this impairment was still evident up to 2 weeks after this 'mild' TBI insult (defined based on minimal acute neurological outcomes and glial reactivity). This raises the question of when (or if) these impairments resolve. A later time point (e.g. 1 month post-injury or beyond) would be valuable here to clarify this time course.

The observed changes in lymphatic drainage after experimental TBI were coupled with some evidence of morphological changes in meningeal vasculature, although this was only performed at a single time point of 7 days post-injury. Given the more subtle effects here, one again wonders if a more appropriate time point (or a time course) would provide greater insight.

As TBI results in an acute ICP increase, and this change was proposed to be an underlying mechanism of meningeal lymphatic dysfunction, jugular vein ligation experiments were next performed to transiently increase ICP, to test the hypothesis that an acute rise in ICP would likewise disrupt meningeal lymphatic drainage.

A key, linking experiment is missing here - the combination of both jugular vein ligation plus experimental TBI (versus sham ligation + TBI), to address the question of whether increased ICP promotes or worsens lymphatic dysfunction. Alternatively, given that TBI is shown to increase ICP itself, the question could likewise be addressed by aggressively treating this ICP pharmacologically (or via ventricular drainage) to evaluate how this affects lymphatic drainage post-TBI. This missing link results in a lack of support for some of the conclusions drawn in the discussion (page 15: "Our

results suggest that aggressive management of elevated ICP after a TBI may be a therapeutic strategy to help limit meningeal lymphatic dysfunction after injury”).

My second main point is regarding the use of a photo-ablation model in isolation to induce pre-injury lymphatic dysfunction.

While the proof-of-concept experimental results are intriguing and convincing, the model is presumed to replicate lymphatic dysfunction as may occur with repetitive TBI or aging. This conclusion would be greatly strengthened by actually incorporating models of either repetitive TBI or aging – e.g. repeat the experiment in aged animals (presumably the current study was performed in young adult mice, although the age range this is not explicitly stated); or the induction of repetitive experimental TBI impacts (as is increasingly common in preclinical TBI research and seems feasible in this particular model).

Additional minor comment: the abstract should mention that findings are based on experimental/animal models as this is currently unclear.

Reviewer #4:

Remarks to the Author:

In this interesting paper, Bolte and colleagues describe an important role for the meningeal lymphatic in the pathogenesis of traumatic brain injury. The precise role of CNS lymphatics during neuroinflammation remains poorly understood. Additionally, the relative role of meningeal lymphatics versus other CNS lymphatics (such as those near the base of the brain, spinal cord, cribriform plate, and cervical lymph nodes) in the drainage of fluid, cells, and antigens remains controversial. While others have shown that the lack of meningeal lymphatic vessels in K14-VEGFR3-Ig mice led to diminished CD4 T cell infiltration into the CNS in the chronic phase of TBI (Wojciechowski, S et al. bioRxiv Oct 29, 2019), this paper is the first to show the relationship between meningeal lymphatics and TBI at an acute 24-hour time-point. The paper provides strong evidence that the meningeal lymphatics are critical for the management of neuroinflammation, suggesting that they are just as important as other lymphatics responsible for CNS drainage including basal meningeal lymphatics, cribriform plate lymphatics, and cervical lymph node lymphatics. This paper also suggests that meningeal lymphatics may undergo dynamic changes during TBI-induced neuroinflammation and/or elevated intracranial pressure, which may have functional consequences.

The major claims of the manuscripts are the followings:

- 1) Dorsal meningeal lymphatic function is impaired after TBI, which occurs as early as 2 hours post-TBI and persists for weeks.
- 2) TBI induces elevated intracranial pressure (ICP).
- 3) Elevated ICP due to TBI reduces dorsal meningeal lymphatic uptake of “CSF” from the subarachnoid space.
- 4) ICP alone is sufficient in promoting meningeal lymphatic dysfunction
- 5) Ablation of meningeal lymphatics prior to TBI promotes elevated neuroinflammation (complement cascade and “microglia” reactivity are elevated)

The data is convincing for most of their major claims, yet there are a couple claims that are not substantiated by sufficient data. Additionally, it remains to be seen whether the changes observed in meningeal lymphatics actually play a role in regulating neuroinflammation.

Some concerns:

1. The authors propose that dorsal meningeal lymphatics can directly uptake CSF. However, this is based upon injection of macromolecules/dyes that disrupts the arachnoid barrier separating meningeal lymphatics from the subarachnoid space. It is unknown if and how the lymphatic vessels in the dura uptake macromolecules and fluid across the arachnoid barrier from the

subarachnoid space in vivo with an intact arachnoid barrier. The authors should at least speculate how this may occur in vivo (do mice have arachnoid granulations)? Anatomically, what is it about these meningeal "hotspots" that allows these dorsal meningeal lymphatics increased access to the subarachnoid space relative to other regions within the superior sagittal sinus? Alternatively, injection through the dura into the subarachnoid space may cause leakage of Lyve-1 antibody or beads that collect near these "hotspots" which may simply represent a region of bottlenecking in the drainage of the dural parenchyma.

2. Basal meningeal lymphatics have been shown to undergo hyperplasia in aged mice, showing dysfunctional "lymphangiogenic" morphology (Ahn et al. Nature 2019.) Alternatively, lymphangiogenesis that occurs many days after chronic neuroinflammation can induce functional lymphangiogenesis (Hsu et al. Nature Communications 2019.) Are the dynamic changes in meningeal lymphatics 1 week after TBI functional lymphangiogenesis, or non-functional hyperplasia? This is an important point therapeutically because functional lymphangiogenesis may reduce cerebral edema during neuroinflammation, while hyperplasia may increase edema. It would be interesting to measure the functionality of these meningeal lymphatics at 1 week or post 1-week TBI, the same as the authors have done at 24 hours post-TBI. Additionally, these meningeal lymphatics regress to baseline levels after inducing lymphangiogenesis during steady-state conditions; it would be interesting to see long-term after TBI if these lymphatics also return to baseline levels or if the changes induced by TBI are long-term.

3. In Figure 4A -C, it is unknown if any of the changes are due to dysfunctional meningeal lymphatics prior to TBI, or if the Visudyne + Laser treatment alone is sufficient in inducing the changes in GFAP and IL-6 levels. Control groups having a "sham" procedure for the TBI with/without Visudyne + laser would be nice to exclude the possibility of the Visudyne + Laser causing unspecific effects. In fact, in Figure 4D it looks like ablation of meningeal lymphatics alone when comparing the two Sham groups is sufficient in generating unique gene clusters of the samples. The authors should potentially show as a supplementary figure which genes if any, and by how much are changing between the non-ablated + sham and ablated + sham groups as shown in Figure 4F -I.

4. The results for Figure 5 use the term "disease associated microglia" which includes a common gene signature, yet to attribute these genes to microglia in this specific data set is somewhat misleading. Many of these genes have been shown to be expressed by any combination of CNS cell types such as astrocytes, as well as peripheral myeloid cells that may infiltrate the CNS after neuroinflammation such as myeloid cells. Single cell RNA sequencing would be needed to truly attribute a gene signature to a specific cell type, or follow-up experiments such as flow cytometry showing increased expression of certain proteins by CD45intermediate CD11b+ microglia or immunohistochemistry showing increased expression by TMEM119+ microglia.

5. After functional and specific ablation of meningeal lymphatics using Visudyne + laser, the authors report a significant reduction in Lyve-1+ vessels but no change in CD31+ area (Supplementary Figure 3). This data is surprising considering lymphatic endothelial cells should also express CD31, and the Visudyne + Laser treatment should also slightly reduce CD31 expression area. Are the lymphoid vessels intact but lose their expression of Lyve-1?

6. Several times throughout the paper, the authors have cited that dorsal meningeal lymphatics play a central role in the drainage of CNS derived molecules, fluid, and cells. Yet the relative role of meningeal lymphatics in the drainage of fluid, cells, and antigens remains controversial (Ma et al. Nature Communications 2019; Ahn et al. Nature 2019). What is their relative contribution versus other CNS lymphatics such as those basal to the brain or near the cribriform plate?

7. The authors provide strong evidence that TBI can induce dynamic changes in meningeal lymphatics, yet it is unclear whether these altered meningeal lymphatics can then influence neuroinflammation after TBI. It would be interesting to do the opposite experiment of Figure 4; i.e.

induce meningeal lymphangiogenesis by administering VEGFC and then measuring TBI-induced pathology. This would add significance to the morphological changes of meningeal lymphatics after TBI.

Reviewers' comments:

Reviewer #1 (Remarks to the Author):

In this manuscript, "Meningeal lymphatic dysfunction exacerbates traumatic brain injury pathogenesis" by Bolte et al., the authors explore how the meningeal lymphatics are impacted by a single mild traumatic brain injury (TBI). In this interesting study, the authors analyze the impact of TBI in the deep cervical lymph nodes, lymphatic vasculature (lyve1 +), gliosis, and expression of the inflammatory, neurodegenerative and complement genes. Also, they show how the pharmacological photoablation of the meningeal lymphatics during the first hours post-TBI leads to aggravating the neuropathology. An important question that remains unsolved is whether meningeal lymphatic dysfunction can be restored days or weeks post-TBI. The functions of uptake CSF, meningeal lymphatic dysfunction, or expression of inflammatory genes were observed in the first 2 to 24 hours after TBI. Still, the greatest detrimental consequences occur weeks or months after mild TBI. It would be very interesting to observe this phenomenon in later times points post-injury and at different severities. The design of therapeutic tools for this mTBI model, increasing the severity, could be incorporated to determine the meningeal lymphatic dysfunction at a chronic stage. The effect of chronic and repetitive TBI on the meningeal lymphatic dysfunction would be highly relevant for the field. The techniques used to study the morphology of the meningeal lymphatic vessels are appropriately applied. However, it is not clear whether the results of RNA sequencing are affected by pharmacological ablation of the vessels or by the effect of the impact of the injury. While altogether this study represents a comprehensive analysis, the authors should provide additional justification/context to the pathological consequences of TBI, and specifically with respect to meningeal dysfunction. A revised version of this manuscript including the incorporation of a model of the severity of TBI and a more in-depth neuropathological analysis would significantly increase the likelihood of this study having a high impact in the TBI field.

We were pleased that the Reviewer found our work to be interesting and deemed that it has potential to be of high impact in the TBI field. As the Reviewer suggested, we have provided substantial new insights into when meningeal lymphatic function is fully restored post-injury, whether we observe this phenomenon with a more severe injury paradigm, and how TBI +/- meningeal lymphatic ablation affects neuropathology. Additionally, as suggested by the Reviewer we have clarified the effects of pharmacologic ablation of the vessels in the RNA sequencing data. Below we provide these new data along with detailed point-by-point responses to their comments. We thank the Reviewer for their valuable suggestions as we feel that they have greatly helped to improve this study.

Major comments:

- In the Material and Methods section, the authors describe that mice are matched for sex and age, and were randomly assigned into experimental groups, however, the exact mice that were used in each experiment are not described in the text nor the legends. Both sex and age can regulate the inflammatory and cerebrovascular responses to brain injuries. The combination of mice into the experimental groups may be induced disparity in the results. Analyzing the groups by sex and age separately is necessary, or at least it should be specified in each experiment which animals were used.

We thank the Reviewer for highlighting the need for this important technical clarification. As suggested by the Reviewer, we have revised the Materials and Methods section to more clearly address the age and sex of the

mice used in our studies. Specifically, each experimental paradigm included only one sex because, as the Reviewer mentioned, sex can result in differential inflammatory and cerebrovascular responses. All adult mice were between eight and sixteen weeks of age. For each independent experiment, adult mice were within 2 weeks of age of each other. The aged mice used for the experiments in Figure 7 and Supplementary Figure 8 were between 18 and 24 months of age.

- It is not clear how the quantitative analysis of the immunohistochemistry images was performed. The specific regions that were quantified, the software used, and the parameters used should all be clearly defined. Microglia activation and the morphological analysis for microglia/macrophages should be incorporated to establish a correlation with the quantitative analysis of Sup. Fig 1.

We appreciate this thoughtful recommendation by the Reviewer and have thoroughly clarified the parameters used to perform quantitative image analysis in the methods section of the manuscript. Quantitative analysis of the acquired images was performed using both Fiji (used for all image analysis except microglia morphology) and Imaris (used for microglia morphology analysis) software. For the assessment of gliosis in the injured and uninjured brains, 2 representative brain sections from the site of the lesion (approximately -0.74 to 0 bregma), or the corresponding area in sham animals, were fully imaged and at least 5 animals were included per experimental group. The full brain section was adjusted for brightness/contrast uniformly for each experiment (Experiments in Figure 6a-c: b/c- GFAP-32/190, b/c- Iba1-51/158, Experiments in Figure 7h-j: b/c- GFAP-41/224, b/c- Iba1-40/174), and then the percent area of coverage of each immunohistochemical marker was calculated per hemisphere for each brain section. Each hemisphere was traced, and then the threshold was uniformly set for each experiment to select for stained cells (Experiments in Figure 6a-c: thresh- GFAP-90/255, b/c- Iba1-115/255, Experiments in Figure 7h-j: b/c- GFAP- 115/255, b/c- Iba1-139/255). The mean percent area was calculated using Microsoft Excel. For Supplementary Figure 1, high magnification images (20x and 63x) were taken directly adjacent to the site of the injury, but the entire hemisphere ipsilateral to the injury was quantified for the levels of gliosis at each timepoint as specified above. For lymph nodes, the percent volume of microbead coverage in cleared dCLN was assessed by creating a 3D reconstruction of the node and then calculating the volume covered by beads divided by the total volume of the node using Fiji. The right and the left dCLN percent volume were averaged together for each mouse. For assessment of meningeal lymphatic vessel coverage and complexity, images of meningeal whole-mounts were acquired using a confocal microscope and Fiji was used for quantifications. The entire meningeal whole-mount was traced and used for quantification. The percent area coverage of Lyve-1 was used to determine the coverage of the lymphatic vessels. When applicable, the same images were used to assess the percentage of field coverage by Lyve-1⁺ CD31⁺ vessels. To assess lymphatic vessel diameter, 70 measurements per meningeal whole-mount were taken by a blinded experimenter (40 along the transverse sinus and 30 along the superior sagittal sinus) and were averaged together. Lymphatic vessel loops and sprouts were calculated along both the transverse sinus and superior sagittal sinus. Areas along the edge of the transverse sinuses were excluded for sprout quantification to avoid counting any lymphatic ends created by removal of the whole-mount. All meningeal whole mounts used for quantification of lymphatic morphology were imaged with identical confocal settings and Fiji parameters (b/c-Lyve-1-18/149, thresh-Lyve-1-102/255). For the microglia morphology analysis, high magnification (63x) images of microglia were taken on the confocal microscope and the filament function in Imaris software (9.5.1 Bitplane) was used to identify Iba1⁺ cells with their dendritic processes. Three fields of view in the peri-lesional area were imaged for each section, and two sections were imaged for each mouse. The microglia parameters for each section and field of view (cell number, filament dendrite length, filament dendrite volume, and number of dendritic branch points) were averaged together and plotted as one point per mouse. For the Sholl analysis, 4 microglia per field of view were quantified for 5 mice.

Additionally, we have included representative images of microglia and astrocytes from the lesion site to include with Supplementary Figure 1 and have performed a quantitative analysis of microglial morphology to determine whether mice with pre-existing lymphatic dysfunction have more highly activated microglia than mice who received TBI alone. For this analysis, high magnification (63x) images of microglia were taken on the confocal microscope and the filament function in Imaris software (9.5.1 Bitplane) was used to identify Iba1+ cells with their dendritic processes. Three fields of view in the peri-lesional area were imaged for each section, and two sections were imaged for each mouse. The microglia parameters for each section and field of view (cell number, filament dendrite length, filament dendrite volume, and number of dendritic branch points) were averaged together and plotted as one point per mouse. For the Sholl analysis, 4 microglia per field of view were quantified for 5 mice. Details of the Imaris analysis have been included in the methods section of the paper. Overall, we found that Iba1+ cells from the pre-existing lymphatic dysfunction TBI group (Ablated + TBI) had a lower number of dendritic branches at greater distances from the soma, indicating a less ramified, more highly activated state than in mice with TBI alone (Not Ablated + TBI), or other control groups (Reviewer Figure panels a-b below). Additionally, the Ablated + TBI mice had a higher number of Iba1+ cells (Reviewer Figure panel c below), and strong trends towards lower dendrite length, dendrite volume, and dendrite branch points compared to the other control groups (Reviewer Figure panels d-f below). These new data suggest that brain macrophages (Iba1+ cells) in TBI mice with pre-existing lymphatic dysfunction (Ablated + TBI) are more highly activated and less ramified than mice in the other control groups. This information has also been added to the methods section, Figure 6, and Supplementary Figure 7 in the manuscript.

• It is well known that the severity of the TBI can be relevant to determining the pathology, as well as in the case of mTBI, the repetitive hits or the intensity of the parameters that control the impact can aggravate the damage of the meningeal lymphatics and their functioning. Is the meningeal lymphatic drainage function severity-dependent? Does it occur an irreversible damage of the meningeal lymphatic vessels at long term post-TBI?

We agree with the Reviewer that this is an important consideration and therefore performed new experiments to determine how a different severity of TBI would affect the functioning of the meningeal lymphatic system. In order to determine how severity might contribute to lymphatic dysfunction, we increased the velocity of the TBI impactor from 5.2 m/s to 6.2 m/s to create an injury that was more severe than our original paradigm. We examined bead drainage through the meningeal lymphatic vasculature at 1-month post injury. At one month post injury, we found that there was a noticeable trend towards decreased drainage in the mice that had received the higher velocity injury (Reviewer Figure panels a-b below), suggesting that meningeal lymphatic drainage capacity is also likely affected by injury severity. We have included these data in Supplementary Figure 3 in the manuscript.

- This study shows that a single mTBI causes impairment in the meningeal lymphatic drainage function up to two weeks post-TBI, although it was seen as an indicator of recovery (Fig. 1h). Is the function of the meningeal lymphatic restored at some time post-TBI, or is it permanently damaged? The inclusion of an experimental group of animals to study the effect of TBI at 2- or 3-months post-injury will clarify if the meningeal lymphatic dysfunction is really time reversible.

We agree with the Reviewer that more long-term assessments of bead drainage after TBI would be useful for determining when drainage deficits have recovered. Therefore, we performed bead drainage experiments at more distant timepoints including 1 month and 2 months after injury. We see that by 2 months, the bead drainage levels have returned to normal. These new data have been incorporated into Figure 1f,g in the manuscript.

- When analyzing the presence of Lyve-1-expressing lymphatic vessels in sham mice and in TBI mice, it is only

decreased at 2 hours post-TBI, but these alterations are not observed at 24 h. Is the uptake of CSF restored after 24 h post-TBI?

We thank the reviewer for bringing it to our attention that we did not properly stress this point in the text. For the resubmission, we have clarified the text related to this data. In Supplementary Figure 4 of the revised manuscript we show that there are still deficits at 24 hours. There is also a strong trend towards decreased Lyve1-488 at 24 hours and a statistically significant difference in the amount of Lyve1-488 that has traveled along the lymphatic vasculature after TBI (Reviewer Figure panels a-c below). We have now also included the *P* values to better indicate this. Additionally, we have performed this same Lyve-1 in vivo injection experiment one week after injury, and we saw that there were no apparent differences in CSF uptake at the hotspots on the transverse sinuses between sham and TBI mice (Reviewer Figure panels d-f below), indicating that the lymphangiogenesis seen at one week post injury may have helped to restore proper CSF uptake into the meningeal lymphatic vasculature. This data is included in Supplementary Figure 4 in the revised manuscript.

- The ICP by itself produces meningeal lymphatic dysfunction at 2 hours after being induced, and it is assumed that the increase of ICP after TBI is the main inducer of the lymphatic damage. What role does the inflammatory response triggered in the meningeal lymphatic dysfunction play? It would be helpful to use pharmacological inhibition of the microglia activation to see if it influences the meningeal lymphatic vessel dysfunction.

We also find it a very intriguing idea to investigate how microglia specifically may be playing a role in this model of TBI, both in the context of meningeal lymphatic dysfunction as well as in the context of increased ICP. Interestingly, elegant recent studies that were just published in the last months indicate that depletion and repopulation of microglia can have beneficial outcomes after TBI both in terms of behavior and adult neurogenesis (Henry et al. *Journal of Neuroscience* 2020 and Willis et al. *Cell* 2020). Unfortunately, we were not able to get these experiments on our animal protocol and completed before we were forced to shut our lab down because of COVID-19. We are very interested in this area of investigation and we are thankful for this thoughtful suggestion. However, given the scope of these proposed studies coupled with our lab continuing to be shut down because of COVID-19 we believe that this could be a stand-alone follow-up study for our group and others.

- The effect that preexisting meningeal lymphatic dysfunction caused by Visudyne photoablation was observed as mild TBI, and it increased the levels of GFAP expression and IL-6. What happens to the activation of the microglia? What regions were specifically measured? The areas circled in Figure 4a vary from the Piriform cortex to the primary somatosensory cortex, where the expression of GFAP varies greatly depending on the focal area of the impact. All of these details are not fully described in this study, and the immunohistochemical analysis should be clarified and carried out accordingly. Also, what happens in sham animals with or without Visudyne administration?

We thank the Reviewer for this valuable feedback and these important suggestions. For the resubmission, we have performed new experiments to assess whether meningeal lymphatic dysfunction caused by Visudyne photoablation results in increased gliosis two weeks after injury, and have included all four experimental groups in order to adequately assess the effects of TBI and ablation alone. The experimental groups included in this experiment are as follows: Not Ablated + Sham, Ablated + Sham, Not Ablated + TBI and Ablated + TBI, where all groups received the drug Visudyne intra cisterna magna, but the “Not Ablated” group did not receive the red laser. It was assured that all injuries were delivered to the right hemisphere at the piriform region of the cortex. We found that two weeks after injury, the mice with pre-existing lymphatic dysfunction have significantly more GFAP immunoreactivity as compared to all other control groups (Reviewer Figure panels a-c below). We have

included the specific immunohistochemical parameters for this experiment above and have also updated the methods section to include these important details.

- RNA sequencing results at 24 h post-surgery, is it unclear whether the inflammatory, neuronal damage, disease-associated microglia, or complement regulatory genes (Fig. 4h, and Fig. 5) are upregulated only in the TBI + ablation group or also in the Sham + ablation group? At least this similarity between the two groups is observed in Fig 4d. This would determine whether it is really the effect of the TBI that modifies these genes or if it is the consequence of the pharmacological ablation of the lymphatic vessels. This same reasoning could be applied to the results in Figure 5.

We appreciate the Reviewer's suggestion to include the data comparing the Not Ablated + Sham and Ablated + Sham groups in order to determine whether the results for the sequencing data in the Not Ablated + TBI vs. Ablated + TBI groups could be attributed to the effects of the ablation alone. We found that although ablation alone resulted in 338 differentially expressed genes, only 29 of these were shared with the TBI +/- ablation group (Reviewer Figure panels a, b, & d below). Moreover, when we looked at the top 20 up- and down-regulated genes for the group with pre-existing lymphatic function alone (Ablated + Sham), we did not see the same trends in the Not Ablated + Sham vs. Ablated + Sham group (Reviewer Figure panel e below). Therefore at 24 hours post injury, we can conclude that while there are changes from the ablation alone, these changes are not the same or as significant as seen in Not Ablated + TBI vs. Ablated + TBI comparison. Moreover, by 1 week post TBI, there are almost no changes resulting from ablation alone (only 11 differentially expressed genes), indicating that 1 week after injury, the ablation procedure is playing a negligible role (Reviewer Figure panels a and c below). As suggested by Reviewer 4, these data have been included as a supplementary figure of the revised manuscript.

• It would be extremely interesting to investigate in more detail if the meningeal lymphatic dysfunction affects the neuropathology and immune-inflammatory response of TBI. What happens to the activation, shape, and morphology of the microglia in the ablated + TBI group compared to the non-ablated + TBI group and in the sham brains? How does the ablation of lymphatic vessels affect neuronal or axonal damage? Does the meningeal lymphatic ablation affect to the T cell priming and recruitment after TBI?

We thank the Reviewer for these thoughtful suggestions. We were also curious as to how meningeal lymphatic dysfunction affects the neuropathology and the inflammatory response after TBI so we performed additional experiments using Visudyne to ablate the meningeal lymphatics and looked at the brain 2 weeks post injury (as detailed above). We found that even two weeks after injury, the percent area of GFAP was significantly higher in the Ablated + TBI group compared to any other groups.

Additionally, to better characterize the immune-inflammatory response after TBI in an unbiased and comprehensive manner, we performed RNA sequencing at a later timepoint (1 week post injury). Interestingly, both the group with TBI alone and the group with pre-existing lymphatic dysfunction before TBI (Ablated + TBI) clustered separately (Reviewer Figure panel a below). While the TBI alone still results in many differentially regulated genes, we find that there are over 200 differentially regulated genes in the TBI group with pre-existing lymphatic dysfunction (Ablated + TBI) when compared to TBI alone (FDR<0.1, Reviewer Figure panels a-b below). When we performed a gene set enrichment analysis with the pathways in the Reactome database in TBI mice with pre-existing lymphatic dysfunction (Ablated + TBI) compared to those with TBI alone (Not Ablated + TBI), we found that the most highly enriched pathways were related to the innate and adaptive immune system and cytokine signaling (Reviewer Figure panel c below). We also used the Biocarta database to determine which signaling pathways were most highly enriched in our dataset. We found that many pathways important for innate immunity (e.g. complement pathway), leukocyte recruitment (e.g. CCR5), and cytokine signaling (e.g. IL-1 and TNF signaling) were more highly enriched in TBI mice that possessed preexisting meningeal lymphatic deficits (Ablated + TBI) than mice receiving TBI alone (Not Ablated + TBI) (Reviewer Figure panel d below). We were also interested in investigating whether the TBI mice with pre-existing lymphatic dysfunction (Ablated + TBI) shared common genes associated with neurodegenerative or psychiatric diseases. To this end, we used the GWAS Catalog, a database of genome-wide association studies, to find known gene associations with several common neurodegenerative and psychiatric diseases including Parkinson's disease, schizophrenia, amyotrophic lateral sclerosis (als), Alzheimer's disease, bipolar disease, depression, multiple sclerosis (ms), and obsessive compulsive disorder (ocd). We determined that the TBI mice with pre-existing lymphatic dysfunction, compared to the mice with TBI alone, shared many genes with these diseases, indicating that these mice share signatures with various neurological disease states (Reviewer Figure panels e-f below). This data has been incorporated into Figure 5 in the manuscript.

Additionally, we have performed a quantitative analysis of microglial morphology to determine whether mice with pre-existing lymphatic dysfunction have more highly activated microglia than mice who received TBI alone. More specifically, we performed a Sholl analysis and found that Iba1+ cells from the pre-existing lymphatic dysfunction (Ablated + TBI) group had a lower number of dendritic branches at greater distances from the soma, indicating a less ramified, more highly activated state than in mice with TBI alone (Not Ablated + TBI), or other control groups (Reviewer Figure panels a,b). Additionally, the Ablated + TBI group had a higher number of Iba1+ cells (Reviewer Figure panel c), and trends towards lower dendrite length, dendrite volume and dendrite branch points compared to the other control groups (Reviewer Figure panels d-f). These data suggest that brain

macrophages (Iba1+ cells) in TBI mice with pre-existing lymphatic dysfunction (Ablated + TBI) are more highly activated and less ramified than Iba1+ cells in the other control groups.

Minor comments:

- Please, check duplications, misspellings and use the acronyms where appropriate, e.g., DAPI counterstain (1: 1000), traumatic brain injury, hrs or hours,... or types: "existingg."
- It is not clear why the time of 2 h is replaced by 3 h post-TBI.
- A more detailed description of the type of BIT produced in terms of neuropathology should be reported or indicate the reference of the same model already

-We thank the Reviewer for pointing out these typos. They have been corrected within the text. We have abbreviated hours, days, months, and weeks in some of the figures to save space using accepted nomenclature (i.e. hrs, d, mo, wks). We have done a better job of defining and highlighting these abbreviations in the figure legends.

-The only time a 3-hours timepoint was used was in the jugular venous ligation (JVL) experiments. This was used instead of 2 hours because this JVL surgical procedure results in peak increases in intracranial pressure at 3 hours post JVL, allowing for a better comparison with the peak intracranial pressure resulting from TBI.

-We have also updated the methods section to include a more detailed description of the traumatic brain injury, and have cited the original paper from which we adapted our TBI protocol in the methods section (Ren et al., *J Cereb Blood Flow Metab* 2013). We use the same impactor parameters and a very similar location of impact,

but have modified the procedure so the mouse is stabilized in the prone position on the nosecone instead of hanging by the teeth to create a more reproducible injury.

Reviewer #2 (Remarks to the Author):

These interesting pre-clinical studies led by Ashley Bolte and John Lukens at University of Virginia provide new insight into mild traumatic brain injury (mTBI) pathogenesis by demonstrating that mTBI causes deficits in meningeal lymphatic drainage within hours of brain trauma in mice. The mTBI-induced impairments in lymphatic function and blocked drainage of macromolecules are temporally linked to raised ICP levels, which is a major driver of mortality and poorer neurological outcomes in head injury patients. Using elegant photoablation models the authors also demonstrate that existing meningeal lymphatic dysfunction predisposes the brain to increased complement signaling which is associated with exacerbated neuroinflammation during subsequent mTBI exposure. As such, these new studies provide new and important insight into meningeal lymphatic functional impairment during mTBI in mice, and provide clues about microglial-mediated neuroinflammatory mechanisms that may lead to delayed cognitive impairments following repetitive mTBI.

Importantly, the closed head mTBI model has been carefully developed to avoid possible confounding results due to direct impact on lymphatic vessels. The righting reflex data indicate that injury severity is mild and thus relevant to concussive-like mTBI. Sophisticated image analysis and ablation models are used to demonstrate impairments in meningeal lymphatic function after mTBI or jugular vein ligations to increase ICP levels in brain. Overall, these pre-clinical studies are performed with high degree of scientific rigor, and appropriate controls and analyses are presented.

The studies related to effects of ICP on meningeal lymphatic function are particularly important for acute TBI and suggest that dysfunction within meningeal lymphatics vessels leads to early evolution of secondary injury (neuroinflammation). Also, the identification of the complement pathway as a mediator of post-traumatic neuroinflammation and dysfunction in the mTBI + Visudyne experiments is exciting, and is certainly worthy of further detailed molecular investigation.

We were pleased that the Reviewer found our studies to be interesting and performed with a high degree of scientific rigor.

The authors should consider the following points identified during peer-review:

1) Does meningeal lymphatic function ever return to sham control levels after single mTBI? What happens months post-mTBI? Because impairments in meningeal lymphatic function slows paravascular influx of macromolecules into the brain after mTBI, and such impairments are associated with cognitive impairments in mice (Da Mesquita 2018, Science PMID: 30046111), what is the long-term impact mTBI on cognitive function in this model? Such an outcome measure is relevant given link between single and repetitive mTBI and chronic neurodegeneration/dementias (CTE, AD...).

We agree with the Reviewer that more long-term assessments of bead drainage after TBI would be useful for determining when drainage deficits had recovered. Therefore, we performed bead drainage experiments at more distant timepoints including 1 month and 2 months after injury. We see that by 2 months, the bead drainage levels have returned to normal. These new data have been incorporated into Figure 1f,g in the manuscript.

We agree with the Reviewer that determining how pre-existing lymphatic dysfunction before mTBI affects behavioral outcomes is indeed important for linking this paradigm with neurodegeneration and dementia. Therefore, we performed several cognitive studies. First, we wished to determine whether the mice with pre-existing lymphatic dysfunction exhibited any motor learning deficits on the rotarod. Interestingly, while all four experimental groups exhibited similar performance on the accelerating rotarod at 24 hours post-TBI, the mice possessing deficits in meningeal lymphatic function before TBI (Ablated + TBI) showed impaired motor learning over days 2 and 3 of the accelerating rotarod test and consistently had a shorter latency to fall than the other control groups (Reviewer Figure panel a below). Indeed, the percent performance increase over three days in the ablated + TBI group was lower than any of the other control groups indicating that undergoing meningeal lymphatic photoablation before TBI results in impaired motor learning (Reviewer Figure panel a below). Additionally, we performed the Novel Location Recognition Test (NLRT) to assess memory function at 2 weeks post injury. The NLRT relies on hippocampal memory to recall the location of two objects over a 24-hour period and to recognize when one object has changed positions (Reviewer Figure panel b below). We found that mice that underwent meningeal lymphatic photoablation before TBI also performed worse in the novel object recognition test (NORT) at two weeks post-brain injury, suggesting that meningeal lymphatic dysfunction prior to TBI results in impaired memory (Reviewer Figure panel c below). These data have been added to Figure 6 of the manuscript.

2) How do morphological changes in meningeal lymphatic vasculature relate to dysfunction? It might be better

to integrate morphological studies in Figure 2 with initial characterization of mTBI-induced meningeal lymphatic dysfunction.

We agree with the Reviewer that a more extensive characterization of meningeal lymphatic vasculature morphology changes in TBI was needed. Therefore, to more fully evaluate the lymphangiogenic response in the meninges after TBI we conducted new studies for the resubmission to add more timepoints to Figure 2. These results show that lymphangiogenesis peaks between 1 week and 2 weeks after TBI and returns to baseline sham levels by 2 months post injury.

3) Can the authors shed light into what macromolecules or immune cells are blocked when meningeal lymphatic dysfunction occurs after mTBI? How do they relate to the subsequent inflammatory response in brain?

In our studies, we have found that mTBI results in marked deficits in the drainage of beads that are 0.5 μm in diameter. These beads are much smaller in size than immune cells (7-15 μm in diameter) that drain from the injured brain. Our lab and others have also shown that TBI incites the generation of inflammasome-derived ASC specks in the brain (Reviewer Figure panel a below). ASC specks are large multiprotein oligomers (1-2 μm in diameter; Sahillioglu AC et al., *Structure* 2014; Stutz A et al., *Methods Mol Biol* 2013) that are potent inducers of inflammatory responses and have also recently been shown to seed amyloid beta spread (Venegas C et al., *Nature* 2017). In our unpublished studies, we have recently found that ASC specks drain from the brain to the deep cervical lymph nodes via the meningeal lymphatics (Reviewer Figure panel b below; red arrows denote ASC specks within the lymphatic vasculature). Collectively, these findings raise the possibility that ASC speck drainage can be impaired when meningeal lymphatic dysfunction occurs after mTBI. Interestingly, recent studies in other models of disease have also shown that deficits in meningeal lymphatic drainage that occur as a result of aging or pharmacological lymphatic ablation can hinder the drainage of protein aggregates commonly associated with neurodegenerative disease including alpha-synuclein, amyloid beta, and tau (Da Mesquita S et al., *Nature* 2018; Patel TK et al., *Mol Neurodegener* 2019; Zou W et al., *Transl Neurodegener* 2019). Given that immune cells and protein aggregates like ASC speck and amyloid beta plaques are much larger in size than the 0.5 μm beads used in our studies, we expect that they will all exhibit impaired clearance from the brain as a result of TBI-induced disruptions to meningeal lymphatic system. We hope to further investigate how TBI-induced meningeal lymphatic dysfunction impacts CNS drainage of immune cells, inflammasome-derived ASC specks, and amyloid beta in follow-up studies once we are allowed back in the lab following COVID-19. In the meantime, we have modified text in the revised manuscript to better highlight the types of macromolecules and immune cells that can be impacted by impairments in meningeal lymphatic drainage.

4) A rescue experiment that improves meningeal function and brain perfusion after mTBI would add great value to the opening figures of this study. VEGF-C overexpression or similar strategy should be considered by the authors.

We thank the Reviewer for this thoughtful suggestion and have performed a new study utilizing VEGF-C to enhance meningeal lymphatic growth in aged mice. Multiple recent studies have shown that aging leads to severe impairments in meningeal lymphatic function (Da Mesquita et al., *Nature* 2018; Ma et al., *Nat Comms* 2018; Ahn et al., *Nature* 2019). Moreover, it is also known that even mild-to-moderate forms of brain trauma can have especially devastating consequences in elderly individuals. Therefore, we were interested in investigating whether recuperating meningeal lymphatic function in aged mice would be effective in limiting TBI disease pathogenesis. To this end, we utilized viral delivery of VEGF-C, which has previously been shown to successfully increase the diameter of the meningeal lymphatic vessels and rejuvenate meningeal lymphatic drainage function in aged mice (Louveau et al., *Nature* 2015; Da Mesquita et al., *Nature* 2018). Accordingly, aged mice (18-24 months of age) and young mice (8-10 weeks of age) received either AAV1-CMV-mVEGF-C or control AAV1-CMV-eGFP by intracisterna magna (i.c.m.) delivery. We confirmed that stable expression of the viral vector along the transverse sinus (TS) and superior sagittal sinus (SSS) in the aged meninges was still observed 1 month

after injection (Reviewer Figure panel a below). Two weeks after viral vector delivery, we subjected these aged and young mice to TBI or sham treatment. Consistent with previous studies, the aged mice who experienced head trauma had a significantly longer loss-of-consciousness and performed significantly worse on the neuroscore behavioral tests than their younger counterparts with the same injury parameters (Reviewer Figure panels b and c below). There were no differences in righting time or neuroscore between the groups that had received AAV1-CMV-mVEGF-C or AAV1-CMV-eGFP delivery (Reviewer Figure panels b and c below). Mice that received sham procedures instead of a TBI and either AAV1-CMV-mVEGF-C or AAV1-CMV-eGFP delivery also showed no differences in righting times or neuroscore behavioral tests, although predictably, both measurements were lower than in mice that had received TBI (Supplementary Figure 8a,b in the revised manuscript). Two weeks after injury and one month after viral vector delivery, we assessed measurements of gliosis (GFAP and Iba1 immunoreactivity) to determine whether treatment with VEGF-C improved neuroinflammatory measures in the injured brains. Interestingly, we saw that aged TBI mice that received viral-mediated VEGF-C treatment (Aged- VEGFC + TBI) had significantly lower levels of Iba1 in the hemisphere contralateral to the injury site, and a trend towards lower levels of Iba1 in the hemisphere ipsilateral to the injury site when compared to mice that had received the control viral vector (Aged- GFP + TBI) (Reviewer Figure panels d-f below). Notably, the levels of Iba1 in the aged TBI mice that were treated with VEGF-C (Aged- VEGFC + TBI) were more similar to the levels of Iba1 immunoreactivity in a young TBI mouse (Young- GFP + TBI) (Reviewer Figure panels d-f below). In contrast, VEGF-C pretreatment was not found to influence GFAP immunoreactivity in aged mice (Reviewer Figure panels d, g, & h below). These data indicate that boosting meningeal lymphatic function after TBI may aid in decreasing levels of Iba1 gliosis in aged mice. These new data have been incorporated into the revised manuscript and now appear in Figure 7 and Supplementary Figure 8 (sham data).

5) Given that mTBI rapidly increases ICP, which is associated impairments in meningeal lymphatic function (nicely demonstrated in figure 3), these pre-clinical studies provide novel insight into secondary injury responses acutely after brain trauma.

We were pleased that the Reviewer found our pre-clinical studies to provide novel insight into secondary injury responses acutely after brain trauma.

6) The RNAseq data identifying the complement pathway as a key inflammatory pathway up-regulated by meningeal lymphatic dysfunction following ablation/mTBI is very interesting because this pathway has been implicated in microglial-mediated synaptic stripping and cognitive decline in age-related neurodegeneration (Hong 2016, Science PMID: 27033548) and neuroinflammatory demyelinating disease (Wernerberg 2019, Immunity PMID: 31883839). These promising findings should be developed further to provide novel insight into DAM microglia-neuronal interactions, structure function analyses of neuroplasticity, and long-term cognitive outcomes. The authors are encouraged to use available transgenic models and physiological methods to pinpoint mechanisms by which complement signaling and DAM microglia react to post-traumatic meningeal lymphatic functional impairments that disrupt neuronal health and cognitive function.

We thank the Reviewer for these thoughtful suggestions and great ideas. Before our lab was shut down because of COVID-19, we were able to examine several aspects more thoroughly. First, we performed several behavioral studies to determine whether long-term cognitive outcomes were different in the mice with pre-existing lymphatic dysfunction compared to mice with TBI alone. We wished to determine whether the mice with pre-existing lymphatic dysfunction exhibited any motor learning deficits on the rotarod. Interestingly, while all four experimental groups exhibited similar performance on the accelerating rotarod at 24 hours post-TBI, the mice possessing deficits in meningeal lymphatic function before TBI (Ablated + TBI) showed impaired motor learning over days 2 and 3 of the accelerating rotarod test and consistently had a shorter latency to fall than the other control groups (Reviewer Figure panel a below). Indeed, the percent performance increase over 3 days in the Ablated + TBI group was lower than any of the other control groups indicating that undergoing meningeal lymphatic photoablation before TBI results in impaired motor learning (Reviewer Figure panel a below). Additionally, we performed the Novel Location Recognition Test (NLRT) to assess memory function at 2 weeks post injury. The NLRT relies on hippocampal memory to recall the location of two objects over a 24 hour period and to recognize when one object has changed positions (Reviewer Figure panel b below). We found that mice that underwent meningeal lymphatic photoablation before TBI also performed worse in the novel object recognition test (NORT) at two weeks post-brain injury, suggesting that meningeal lymphatic dysfunction prior to TBI results in impaired memory (Reviewer Figure panel c below). These data have been added to Figure 6 of the manuscript.

Second, we have performed a quantitative analysis of microglial morphology to determine whether mice with pre-existing lymphatic dysfunction have more highly activated microglia than mice who received TBI alone. For this analysis, high magnification (63x) images of microglia were taken on the confocal microscope and the filament function in Imaris software (9.5.1 Bitplane) was used to identify Iba1+ cells with their dendritic processes. Three fields of view in the peri-lesional area were imaged for each section, and two sections were imaged for each mouse. The microglia parameters for each section and field of view (cell number, filament dendrite length, filament dendrite volume, and number of dendritic branch points) were averaged together and plotted as one point per mouse. For the Sholl analysis, 4 microglia per field of view were quantified for 5 mice. Details of the Imaris analysis have been included in the methods section of the paper. Overall, we found that Iba1+ cells from the pre-existing lymphatic dysfunction TBI group (Ablated + TBI) had a lower number of dendritic branches at greater distances from the soma, indicating a less ramified, more highly activated state than in mice with TBI alone (Not Ablated + TBI), or other control groups (Reviewer Figure panels a-b below). Additionally, the Ablated + TBI group had a higher number of Iba1+ cells (Reviewer Figure panel c below), and trends towards lower dendrite length, dendrite volume and dendrite branch points compared to the other control groups (Reviewer Figure panels d-f below). These data suggest that the Iba1+ cells in mice with pre-existing lymphatic dysfunction (Ablated + TBI) are more highly activated and less ramified than mice in the other control groups. This new information has also been added to the methods section, Figure 6, and Supplementary Figure 7 in the revised manuscript. We also agree that exploring the role of complement signaling in our model will be interesting. However, given the scope of these proposed complement studies and the continued closure of our lab because of COVID-19, we believe that these future complement studies could be a stand-alone study that follows up on the manuscript under review here. We think it is a great idea for the next stage of this work and we plan to take this thoughtful advice and work these experiments into our next TBI grant submission.

7) Minor - representative photomicrographs of microglia and astrocytes should be included in supplemental figure 1.

We thank the Reviewer for this suggestion and have added representative images to Supplementary Figure 1.

Reviewer #3 (Remarks to the Author):

This manuscript by Bolte and colleagues describes a series of experiments to address a key knowledge gap in brain research - the role of meningeal lymphatic drainage in neurological disease. Specifically, the authors employed an experimental model of traumatic brain injury (TBI) to demonstrate proof-of-concept evidence for meningeal lymphatic dysfunction in this context, as well as some evidence of potential mechanisms. This paper will be of particular importance to the neurotrauma field given recent interest in lymphatics/glymphatics/CSF transport and brain injury biomarkers, across a whole spectrum of TBI severity from mild through to severe. The principle findings provide novel insight into how the meningeal lymphatic system is impacted by TBI, as well as how pre-existing deficits in this system can exacerbate the neuroinflammatory consequences that are characteristic of a brain injury.

The methodology appears sound (i.e. blinded analyses, randomization, inclusion of both sexes, power calculations), with minor exceptions, as follows: (1) What age range of animals was used? This should be reported, particularly in light of how CNS lymphatic drainage function declines with age (as noted by the authors in the discussion). Similarly, TBI neuropathology is influenced by age. (2) For the TBI model, where any analgesics administered? The procedure used to generate 'sham' animals should also be clearly described (e.g. surgical preparation + anesthesia?). (3) Assessment of GFAP/Iba1 glial reactivity appears weak and subject to unintended selection bias (only 2 representative slides imaged and averaged per brain). However, this is somewhat mitigated by most interpretations of this data being supported by additional measures of neuroinflammation (e.g. gene expression analyses).

We were pleased to learn that the Reviewer believes our principal findings offer novel insights into how the meningeal lymphatic system is impacted by TBI, as well as how pre-existing deficits in this system can exacerbate the neuroinflammatory consequences that are characteristic of a brain injury.

We also appreciate their suggestions to clarify the Materials and Methods section of the paper. As suggested by the Reviewer, we have made the following changes to the Materials and Methods section of the revised manuscript:

- 1) We have updated our Materials and Methods section to indicate that the mice used were between 8-16 weeks of age for all the experiments except for the aged mice experiments in Figure 7, in which the mice were 18-24 months of age.
- 2) We have clarified the sham procedure in the methods section. No analgesic was administered for this relatively mild injury. Mice were provided with soft food and were allowed to recover on the heating pad for at least 6 hours.
- 3) As suggested by the Reviewer, we have performed new experiments to assess whether meningeal lymphatic dysfunction caused by Visudyne photoablation results in increased gliosis two weeks after injury, and have included all four experimental groups in order to adequately assess the effects of TBI and ablation alone. The experimental groups included in this experiment are as follows: Not Ablated + Sham, Ablated + Sham, Not Ablated + TBI and Ablated + TBI, where the Ablated or Not Ablated groups both received the drug Visudyne intra cisterna magna, but the “Not Ablated” groups did not receive the red laser. We found that two weeks after injury, the mice with pre-existing lymphatic dysfunction have significantly more GFAP immunoreactivity as compared to all other control groups (Reviewer Figure panels a-c below). We have included the specific immunohistochemical parameters for this experiment above and have also updated the methods section to include these important details. The two representative whole brain sections were taken from the area of the lesion (approximately -0.74 to 0 bregma).

For the resubmission, we have also performed new studies evaluating changes in microglia morphology (Figure 6d-f of the revised manuscript) and cognitive performance (Figure 6g-i of the revised manuscript, as well as new RNA-seq experiments at one week post-injury (Figure 5 of the revised manuscript).

The strongest finding is the evidence of lymphatic drainage impairments across a time course after TBI (Figure 1), in both the cervical lymph nodes and meninges. Somewhat surprisingly, this impairment was still evident up to 2 weeks after this 'mild' TBI insult (defined based on minimal acute neurological outcomes and glial reactivity). This raises the question of when (or if) these impairments resolve. A later time point (e.g. 1 month post-injury or beyond) would be valuable here to clarify this time course.

We agree with the Reviewer that more long-term assessments of bead drainage after TBI would be useful for determining when drainage deficits had recovered. Therefore, we performed bead drainage experiments at more distant timepoints including 1 month and 2 months after injury. We see that by 2 months, the bead drainage levels have returned to normal. These new data have been incorporated into Figure 1f,g in the manuscript.

The observed changes in lymphatic drainage after experimental TBI were coupled with some evidence of morphological changes in meningeal vasculature, although this was only performed at a single time point of 7 days post-injury. Given the more subtle effects here, one again wonders if a more appropriate time point (or a time course) would provide greater insight.

We agree with the Reviewer that a more extensive characterization of meningeal lymphatic vasculature morphology changes in TBI is needed. Therefore, to more fully evaluate the lymphangiogenic response in the meninges after TBI we conducted new studies for the resubmission to add more timepoints to Figure 2. These results show that lymphangiogenesis peaks between 1 week and 2 weeks after TBI and returns to baseline sham levels by 2 months.

As TBI results in an acute ICP increase, and this change was proposed to be an underlying mechanism of

meningeal lymphatic dysfunction, jugular vein ligation experiments were next performed to transiently increase ICP, to test the hypothesis that an acute rise in ICP would likewise disrupt meningeal lymphatic drainage.

A key, linking experiment is missing here – the combination of both jugular vein ligation plus experimental TBI (versus sham ligation + TBI), to address the question of whether increased ICP promotes or worsens lymphatic dysfunction. Alternatively, given that TBI is shown to increase ICP itself, the question could likewise be addressed by aggressively treating this ICP pharmacologically (or via ventricular drainage) to evaluate how this affects lymphatic drainage post-TBI. This missing link results in a lack of support for some of the conclusions drawn in the discussion (page 15: “Our results suggest that aggressive management of elevated ICP after a TBI may be a therapeutic strategy to help limit meningeal lymphatic dysfunction after injury”).

We agree with the Reviewer that our data does not support this aforementioned conclusion drawn in the discussion (i.e., “Our results suggest that aggressive management of elevated ICP after a TBI may be a therapeutic strategy to help limit meningeal lymphatic dysfunction after injury”). Therefore, we have removed this text from the discussion. We thank the Reviewer for raising this important point.

My second main point is regarding the use of a photo-ablation model in isolation to induce pre-injury lymphatic dysfunction.

While the proof-of-concept experimental results are intriguing and convincing, the model is presumed to replicate lymphatic dysfunction as may occur with repetitive TBI or aging. This conclusion would be greatly strengthened by actually incorporating models of either repetitive TBI or aging – e.g. repeat the experiment in aged animals (presumably the current study was performed in young adult mice, although the age range this is not explicitly stated); or the induction of repetitive experimental TBI impacts (as is increasingly common in preclinical TBI research and seems feasible in this particular model).

We were pleased that the Reviewer found our proof-of-concept experimental results involving photoablation to be intriguing and convincing. Moreover, we thank the Reviewer for their thoughtful suggestion here and have performed a new study utilizing VEGF-C to enhance meningeal lymphatic growth in aged mice. Multiple recent studies have shown that aging leads to severe impairments in meningeal lymphatic function (Da Mesquita et al., *Nature* 2018; Ma et al., *Nat Comms* 2018; Ahn et al., *Nature* 2019). Moreover, it is also known that even mild-to-moderate forms of brain trauma can have especially devastating consequences in elderly individuals. Therefore, we were interested in investigating whether recuperating meningeal lymphatic function in aged mice would be effective in limiting TBI disease pathogenesis. To this end, we utilized viral delivery of VEGF-C, which has previously been shown to successfully increase the diameter of the meningeal lymphatic vessels and rejuvenate meningeal lymphatic drainage function in aged mice (Louveau et al., *Nature* 2015; Da Mesquita et al., *Nature* 2018). Accordingly, aged mice (18-24 months of age) and young mice (8-10 weeks of age) received either AAV1-CMV-mVEGF-C or control AAV1-CMV-eGFP by intracisterna magna (i.c.m.) delivery. We confirmed that stable expression of the viral vector along the transverse sinus (TS) and superior sagittal sinus (SSS) in the aged meninges was still observed 1 month after injection (Reviewer Figure panel a below). Two weeks after viral vector delivery, we subjected these aged and young mice to TBI or sham treatment. Consistent with previous studies, the aged mice who experienced head trauma had a significantly longer loss-of-consciousness and performed significantly worse on the neuroscore behavioral tests than their younger counterparts with the same injury parameters (Reviewer Figure panels b and c below). There were no differences in righting time or neuroscore between the groups that had received AAV1-CMV-mVEGF-C or AAV1-CMV-eGFP delivery (Reviewer Figure panels b and c below). Mice that received sham procedures instead of a TBI and either AAV1-CMV-mVEGF-C or AAV1-CMV-eGFP delivery also showed no differences in righting times or neuroscore behavioral tests, although predictably, both measurements were lower than in mice that had received TBI (Supplementary Figure 8a,b in the revised manuscript). Two weeks after injury and one month after viral vector

delivery, we assessed measurements of gliosis (GFAP and Iba1 immunoreactivity) to determine whether treatment with VEGF-C improved neuroinflammatory measures in the injured brains. Interestingly, we saw that aged TBI mice that received viral-mediated VEGF-C treatment (Aged- VEGFC + TBI) had significantly lower levels of Iba1 in the hemisphere contralateral to the injury site, and a trend towards lower levels of Iba1 in the hemisphere ipsilateral to the injury site when compared to mice that had received the control viral vector (Aged- GFP + TBI) (Reviewer Figure panels d-f below). Notably, the levels of Iba1 in the aged TBI mice that were treated with VEGF-C (Aged- VEGFC + TBI) were more similar to the levels of Iba1 immunoreactivity in a young TBI mouse (Young- GFP + TBI) (Reviewer Figure panels d-f below). In contrast, VEGF-C pretreatment was not found to influence GFAP immunoreactivity in aged mice (Reviewer Figure panels d, g, & h below). These data indicate that boosting meningeal lymphatic function after TBI may aid in decreasing levels of Iba1 gliosis in aged mice. These new data have been incorporated into the revised manuscript and now appear in Figure 7 and Supplementary Figure 8 (sham data).

Additional minor comment: the abstract should mention that findings are based on experimental/animal models as this is currently unclear.

We thank the Reviewer for this important suggestion and have updated the abstract accordingly.

Reviewer #4 (Remarks to the Author):

In this interesting paper, Bolte and colleagues describe an important role for the meningeal lymphatic in the pathogenesis of traumatic brain injury. The precise role of CNS lymphatics during neuroinflammation remains poorly understood. Additionally, the relative role of meningeal lymphatics versus other CNS lymphatics (such as those near the base of the brain, spinal cord, cribriform plate, and cervical lymph nodes) in the drainage of fluid, cells, and antigens remains controversial. While others have shown that the lack of meningeal lymphatic vessels in K14-VEGFR3-Ig mice led to diminished CD4 T cell infiltration into the CNS in the chronic phase of TBI (Wojciechowski, S et al. bioRxiv Oct 29, 2019), this paper is the first to show the relationship between meningeal lymphatics and TBI at an acute 24-hour time-point. The paper provides strong evidence that the meningeal lymphatics are critical for the management of neuroinflammation, suggesting that they are just as important as other lymphatics responsible for CNS drainage including basal meningeal lymphatics, cribriform plate lymphatics, and cervical lymph node lymphatics. This paper also suggests that meningeal lymphatics may undergo dynamic changes during TBI-induced neuroinflammation and/or elevated intracranial pressure, which may have functional consequences.

The major claims of the manuscripts are the followings:

- 1) Dorsal meningeal lymphatic function is impaired after TBI, which occurs as early as 2 hours post-TBI and persists for weeks.
- 2) TBI induces elevated intracranial pressure (ICP).
- 3) Elevated ICP due to TBI reduces dorsal meningeal lymphatic uptake of “CSF” from the subarachnoid space.
- 4) ICP alone is sufficient in promoting meningeal lymphatic dysfunction
- 5) Ablation of meningeal lymphatics prior to TBI promotes elevated neuroinflammation (complement cascade and “microglia” reactivity are elevated)

The data is convincing for most of their major claims, yet there are a couple claims that are not substantiated by sufficient data. Additionally, it remains to be seen whether the changes observed in meningeal lymphatics actually play a role in regulating neuroinflammation.

We are happy that the Reviewer found our paper to be interesting and that our work provides strong evidence that the meningeal lymphatics are critical for the management of neuroinflammation.

Some concerns:

1. The authors propose that dorsal meningeal lymphatics can directly uptake CSF. However, this is based upon injection of macromolecules/dyes that disrupts the arachnoid barrier separating meningeal lymphatics from the subarachnoid space. It is unknown if and how the lymphatic vessels in the dura uptake macromolecules and fluid across the arachnoid barrier from the subarachnoid space in vivo with an intact arachnoid barrier. The authors should at least speculate how this may occur in vivo (do mice have arachnoid granulations)? Anatomically, what is it about these meningeal “hotspots” that allows these dorsal meningeal lymphatics increased access to the subarachnoid space relative to other regions within the superior sagittal sinus? Alternatively, injection through the dura into the subarachnoid space may cause leakage of Lyve-1 antibody or beads that collect near these “hotspots” which may simply represent a region of bottlenecking in the drainage of the dural parenchyma.

We appreciate the Reviewer’s suggestions, and have updated our Discussion section to address current theories of how CSF is taken up into the lymphatic vasculature both in humans and in mice.

2. Basal meningeal lymphatics have been shown to undergo hyperplasia in aged mice, showing dysfunctional “lymphangiogenic” morphology (Ahn et al. Nature 2019.) Alternatively, lymphangiogenesis that occurs many

days after chronic neuroinflammation can induce functional lymphangiogenesis (Hsu et al. Nature Communications 2019.) Are the dynamic changes in meningeal lymphatics 1 week after TBI functional lymphangiogenesis, or non-functional hyperplasia? This is an important point therapeutically because functional lymphangiogenesis may reduce cerebral edema during neuroinflammation, while hyperplasia may increase edema. It would be interesting to measure the functionality of these meningeal lymphatics at 1 week or post 1-week TBI, the same as the authors have done at 24 hours post-TBI. Additionally, these meningeal lymphatics regress to baseline levels after inducing lymphangiogenesis during steady-state conditions; it would be interesting to see long-term after TBI if these lymphatics also return to baseline levels or if the changes induced by TBI are long-term.

The Reviewer brings up several interesting and important points. First, the Reviewer inquires whether the lymphangiogenesis seen after TBI is functional or non-functional hyperplasia. To address this question, we have performed the same Lyve-1 in-vivo injection experiment 1 week after injury that was performed at 2 and 24 hours post TBI. We saw that there were no apparent differences in CSF uptake at the hotspots on the transverse sinuses between sham and TBI mice (Reviewer Figure panels a-c below), indicating that the lymphangiogenesis seen at one week post injury may have helped to restore proper CSF uptake into the meningeal lymphatic vasculature. This new data now appears in Supplementary Figure 4 of the revised manuscript.

Additionally, we were also interested in the question of whether the lymphatic morphology changes seen at 1 week post-injury eventually return to normal levels. Therefore, we have added more timepoints to Figure 2 to more fully characterize the lymphangiogenic response in the meninges after TBI. These results show that lymphangiogenesis peaks between 1 week and 2 weeks after TBI and returns to baseline sham levels by 2 months.

3. In Figure 4A -C, it is unknown if any of the changes are due to dysfunctional meningeal lymphatics prior to TBI, or if the Visudyne + Laser treatment alone is sufficient in inducing the changes in GFAP and IL-6 levels. Control groups having a “sham” procedure for the TBI with/without Visudyne + laser would be nice to exclude the possibility of the Visudyne + Laser causing unspecific effects. In fact, in Figure 4D it looks like ablation of meningeal lymphatics alone when comparing the two Sham groups is sufficient in generating unique gene clusters of the samples. The authors should potentially show as a supplementary figure which genes if any, and by how much are changing between the non-ablated + sham and ablated + sham groups as shown in Figure 4F -I.

We appreciate the Reviewer’s suggestion to include the data comparing the Not Ablated + Sham and Ablated + Sham groups in order to determine whether the results for the sequencing data in the Not Ablated + TBI vs. Ablated + TBI groups could be attributed to the effects of the ablation alone. We found that although

ablation alone resulted in 338 differentially expressed genes, only 29 of these were shared with the TBI +/- ablation group (Reviewer Figure panels a, b, & d below). Moreover, when we looked at the top 20 up- and down-regulated genes for the group with pre-existing lymphatic function alone (Ablated + Sham), we did not see the same trends in the Not Ablated + Sham vs. Ablated + Sham group (Reviewer Figure panel e below). Therefore at 24 hours post injury, we can conclude that while there are changes from the ablation alone, these changes are not the same or as significant as seen in Not Ablated + TBI vs. Ablated + TBI comparison. Moreover, by 1 week post TBI, there are almost no changes resulting from ablation alone (only 11 differentially expressed genes), indicating that 1 week after injury, the ablation procedure is playing a negligible role (Reviewer Figure panels a and c below). These additional new data have been incorporated into the manuscript as Supplementary Figure 6.

4. The results for Figure 5 use the term “disease associated microglia” which includes a common gene signature, yet to attribute these genes to microglia in this specific data set is somewhat misleading. Many of these genes have been shown to be expressed by any combination of CNS cell types such as astrocytes, as well as peripheral myeloid cells that may infiltrate the CNS after neuroinflammation such as myeloid cells. Single cell RNA sequencing would be needed to truly attribute a gene signature to a specific cell type, or follow-up experiments such as flow cytometry showing increased expression of certain proteins by CD45intermediate CD11b+ microglia or immunohistochemistry showing increased expression by TMEM119+ microglia.

We completely agree with the Reviewer that this term is misleading since the experiment includes bulk brain tissue and is not enriched for microglia. Therefore, we have removed this heatmap and discussion of disease associated microglia from the manuscript altogether.

5. After functional and specific ablation of meningeal lymphatics using Visudyne + laser, the authors report a significant reduction in Lyve-1+ vessels but no change in CD31+ area (Supplementary Figure 3). This data is surprising considering lymphatic endothelial cells should also express CD31, and the Visudyne + Laser treatment should also slightly reduce CD31 expression area. Are the lymphoid vessels intact but lose their expression of Lyve-1?

We thank the Reviewer for inquiring about this. We have revised the text and figure to show the percent of Lyve-1 negative CD31 positive area in order to avoid including the lymphatic vessels in our analysis. We believe that our original dataset did not exhibit an overall decrease in CD31 levels because the Lyve-1 vessels make up a relatively small proportion of the total percent area of the meningeal whole mounts, and because the lymphatic vessels are in an area (the transverse and superior sagittal sinuses) that strongly labels for CD31 regardless of Visudyne photoablation of the lymphatic vasculature. However, our new analysis of CD31⁺Lyve1⁻ percent area allows for measurement of the blood vasculature without the lymphatic vasculature. Moreover, this Visudyne photoablation approach has also been extensively shown to robustly destroy lymphatic vessels and impair lymphatic drainage function (Tammela et al., *Science Translational Medicine* 2011 & Da Mesquita et al., *Nature* 2018).

6. Several times throughout the paper, the authors have cited that dorsal meningeal lymphatics play a central role in the drainage of CNS derived molecules, fluid, and cells. Yet the relative role of meningeal lymphatics in the drainage of fluid, cells, and antigens remains controversial (Ma et al. *Nature Communications* 2019; Ahn et al. *Nature* 2019). What is their relative contribution versus other CNS lymphatics such as those basal to the brain or near the cribriform plate?

This is a very interesting question that we hope to investigate further in future studies. We anticipate that the basal lymphatics are also playing important roles in our TBI model given recent studies demonstrating their involvement in CSF drainage to the dCLN (Ahn et al., *Nature* 2019). The basal lymphatics have only been characterized very recently and there are unfortunately few groups in the world that currently possess the necessary expertise to perform the technically challenging procedures needed to rigorously explore this area. We hope to establish a collaboration with a group that can teach us techniques to examine the basal lymphatics in future studies. Unfortunately, COVID-19 related lab shutdowns and bans on travel has made it difficult to acquire the necessary training in these specialized techniques or obtain the emerging tools needed to investigate this area. We do, however, hope to further explore the role of basal lymphatics in TBI in our future studies.

7. The authors provide strong evidence that TBI can induce dynamic changes in meningeal lymphatics, yet it is unclear whether these altered meningeal lymphatics can then influence neuroinflammation after TBI. It would be interesting to do the opposite experiment of Figure 4; i.e. induce meningeal lymphangiogenesis by administering VEGFC and then measuring TBI-induced pathology. This would add significance to the morphological changes of meningeal lymphatics after TBI.

We thank the Reviewer for this thoughtful suggestion and have performed a new study utilizing VEGF-C to enhance meningeal lymphatic growth in aged mice. Multiple recent studies have shown that aging leads to severe impairments in meningeal lymphatic function (Da Mesquita et al., *Nature* 2018; Ma et al., *Nat Comms* 2018; Ahn et al., *Nature* 2019). Moreover, it is also known that even mild-to-moderate forms of brain trauma can have especially devastating consequences in elderly individuals. Therefore, we were interested in investigating whether recuperating meningeal lymphatic function in aged mice would be effective in limiting TBI disease pathogenesis. To this end, we utilized viral delivery of VEGF-C, which has previously been shown to successfully increase the diameter of the meningeal lymphatic vessels and rejuvenate meningeal lymphatic drainage function in aged mice (Louveau et al., *Nature* 2015; Da Mesquita et al., *Nature* 2018). Accordingly, aged mice (18-24 months of age) and young mice (8-10 weeks of age) received either AAV1-CMV-mVEGF-C or control AAV1-CMV-eGFP by intracisterna magna (i.c.m.) delivery. We confirmed that stable expression of the viral vector along the transverse sinus (TS) and superior sagittal sinus (SSS) in the aged meninges was still observed 1 month after injection (Reviewer Figure panel a below). Two weeks after viral vector delivery, we subjected these aged and young mice to TBI or sham treatment. Consistent with previous studies, the aged mice who experienced head trauma had a significantly longer loss-of-consciousness and performed significantly worse on the neuroscore behavioral tests than their younger counterparts with the same injury parameters (Reviewer Figure panels b and c below). There were no differences in righting time or neuroscore between the groups that had received AAV1-CMV-mVEGF-C or AAV1-CMV-eGFP delivery (Reviewer Figure panels b and c below). Mice that received sham procedures instead of a TBI and either AAV1-CMV-mVEGF-C or AAV1-CMV-eGFP delivery also showed no differences in righting times or neuroscore behavioral tests, although predictably, both measurements were lower than in mice that had received TBI (Supplementary Figure 8a,b in the revised manuscript). Two weeks after injury and one month after viral vector delivery, we assessed measurements of gliosis (GFAP and Iba1 immunoreactivity) to determine whether treatment with VEGF-C improved neuroinflammatory measures in the injured brains. Interestingly, we saw that aged TBI mice that received viral-mediated VEGF-C treatment (Aged- VEGFC + TBI) had significantly lower levels of Iba1 in the hemisphere contralateral to the injury site, and a trend towards lower levels of Iba1 in the hemisphere ipsilateral to the injury site when compared to mice that had received the control viral vector (Aged- GFP + TBI) (Reviewer Figure panels d-f below). Notably, the levels of Iba1 in the aged TBI mice that were treated with VEGF-C (Aged- VEGFC + TBI) were more similar to the levels of Iba1 immunoreactivity in a young TBI mouse (Young- GFP + TBI) (Reviewer Figure panels d-f below). In contrast, VEGF-C pretreatment was not found to influence GFAP immunoreactivity in aged mice (Reviewer Figure panels d, g, & h below). These data indicate that boosting meningeal lymphatic function after TBI may aid in decreasing levels of Iba1 gliosis in aged mice. These new data have been incorporated into the revised manuscript and now appear in Figure 7 and Supplementary Figure 8 (sham data).

Reviewers' Comments:

Reviewer #1:

Remarks to the Author:

I thank the authors for the detailed answers, in incorporating new data and following all the points of my comments in the first submission.

Indeed, adding several post-injury times helps to understand the neuropathology associated with damage to the meninges.

Although a new time point at two months post-TBI was added, experiments suggested to inhibit the microglia have not been performed due to lab shutdown because of COVID-19.

I understand that the experiments on this resubmission, including the Visudyne photoablation, were performed during/before the first submission, but had not been incorporated into the study despite this. I would strongly encourage the authors to follow up with these studies in the future. The issue of sex was partially resolved, but readers will be left a bit in the dark whether males or females were used in these studies, even if only one sex was used per experiment, which sex? And why?

In summary, the authors have responded well to the reviewer's and editor's suggestions, editing the text, and incorporated new experiments and figures, although not completely responding to all of my concerns. I recommend for publication after the authors clarify the points raised above.

Reviewer #2:

Remarks to the Author:

In revision, the authors have significantly improved this new preclinical investigation into the causes and consequences of meningeal lymphatic dysfunction in mTBI. Several open questions from the original manuscript have been addressed with new data and relevant discussion, and the long-term effects of deficits in meningeal lymphatic drainage are now more clearly defined. In particular, the revised manuscript now confirms that impairments in brain drainage after mTBI return to baseline levels by 2 months post-injury. New data on meningeal lymphatic vascular remodeling over time reveal the dynamic of lymphanogenesis after mTBI and a return to baseline sham levels at 2 months post-injury as well. The neurological consequences of meningeal lymphatic dysfunction following mTBI have been established with increased cognitive impairments in mice that had pre-existing meningeal lymphatic impairments (induced by visudyne) prior to mTBI. The additional analysis in aged mTBI mice increases the translational relevance of these preclinical studies, and a rescue experiment with VEGF-C in aged mice reduces microglial activation when meningeal lymphatics are functionally restored. This intervention study provides proof-of-concept support for therapeutic targeting of meningeal lymphatic system to improve outcomes in TBI. Overall, these novel studies by Bolte and colleagues advance basic understanding about the contribution of meningeal lymphatic impairments to secondary injury and recovery following mTBI. Congratulations to the authors on completing very thorough and very interesting preclinical TBI studies.

David Loane

Reviewer #3:

Remarks to the Author:

The authors have comprehensively addressed all comments and suggestions recommended by the reviewers, and the study + manuscript are significantly improved as a result. Incorporation of an aged cohort and targeting of VEGF-C in particular provide novel insight into mechanisms of meningeal lymphatic drainage dysfunction.

Reviewer #4:

Remarks to the Author:

Overall, the authors made a substantial effort to reply to the concerns of the reviewers and the manuscript is significantly improved. Considering that the lab was closed due to Covid19, the progress on the manuscript data is commendable. In spite of the strengths in the rebuttal, some concerns are still valid.

The authors make a point that at 24 hours after TBI there are in fact changes due to ablation alone, however, it is not the same changes as in the ablated + TBI group and thus they consider the effects of ablation alone negligible. This is reinforced by the claim that at 1-week post-TBI there are virtually no changes in the bulk RNA-seq data in the control groups. While this reviewer agrees with the rebuttal in making a supplementary figure with the title saying photoablation alone is sufficient to induce changes but disagrees with the text saying the 338 differentially expressed genes are "negligible" since they are different and less significant than the experimental groups: It is not reasonable to say that 338 differentially expressed genes represent a "negligible" change, especially since the data in the main figure 4 is focused on the 24-hour time-point where these changes in ablation alone can be seen by the bulk RNA-seq data. The authors do say in the title of Supplementary Figure 6 that there are indeed changes, however, they claim that they did not see the same "trend" of genes in the top 20 up and down-regulated genes between the ablated and non-ablated sham groups. The trend does in fact look similar in panel e in Supplementary Figure 6 with the ablated group showing a general increase in similar complement-related genes as the data shown in Figure 4D. It is still unclear if the top 20 genes in supplementary figure 6 is what is shown in panel e? It's likely not, so some of the top 20 genes in supplementary figure 6B should be identified similar to Figure 4C to see if the top 20 genes are in fact different. The authors also claim that these changes due to photo-ablation are "not as significant," yet this cannot be seen as the scale bars in Figure 4D and Supplementary Figure 6E are not the same and/or the p-values are not shown. Additionally, even if the photoablation control group's genes are not as significant, the question is if it is significant enough to make the actual data in Figure 4D not significant. The authors should also look at the photo-ablated SHAM versus the photo-ablated TBI to determine how much of an actual difference TBI alone induces. This may have been shown in the middle panel of Figure 4B, but it is unclear if the authors are comparing the ablated sham versus ablated TBI or unablated TBI. Alternatively, if it is possible the authors should compare the non-ablated TBI to ablated TBI after normalizing or subtracting the differences due to photoablation alone in the sham controls. Some comments addressing these concerns would be helpful.

Another issue is that it is possible that differences in astrocytes are masked by other cell types in the bulk RNA-seq such as microglia or neurons, or by cells in the ipsilateral side of the brain; in fact, Figure 6D looks like ablation alone had an effect on GFAP area similar to TBI alone but not so much in microglia. Comment regarding this limitation would be useful.

Visudyne specificity for lymphatic endothelial ablation: The authors address the previous concern regarding CD31 expression by doing percent area of CD31+ Lyve1- vessels in a supplementary figure, but this is not sufficient in addressing the reviewer's concerns. To count CD31+ blood vessels across the entire meninges even as a percent area is misleading when photoablation only occurred in a specific region. Any differences in CD31+ blood vessels in the area affected by photoablation would be masked by the unaffected CD31+ blood vessels in the rest of the meninges (the authors even state this in their rebuttal). The authors should analyze the percent area of CD31+ Lyve1- vessels in specifically the photo-ablated area to ensure that Visudyne is specific for lymphatic endothelial cells and not blood endothelial cells. The authors also say that the lymphatic vessels in photo-ablated area strongly label for CD31 even with photoablation; this data should be shown in the supplementary and discussed as it's contradictory to their conclusion that photoablation doesn't affect blood vessels. If this is true, then the authors cannot distinguish a photoablated lymphatic vessel from a blood-vessel and the conclusion that photoablation doesn't affect blood-vessels would be difficult to make. Two citations are provided describing the specificity

of Visudyne for LECs, however, the first citation shows photoablation also induces smooth muscle cell necrosis, and the second citation uses the same analysis as here with presumably the same issue. An alternative is to stain for other lymphatic markers along with CD31; it's possible that photoablation reduces Lyve-1 expression but like CD31 does not reduce the expression of VEGFR3, Podoplanin, and/or Prox1. Some explanation regarding Visudyne specificity for LECs would be important.

Is there lymphangiogenesis in both the contralateral and ipsilateral sides of the TBI? This should be clarified.

The choice of behavioral tests needs additional explanation. The author's model of TBI focuses on the piriform cortex, which is classically associated with the sense of smell and not motor coordination or learning/memory. How ablation of meningeal lymphatics coupled with TBI in the piriform cortex affects motor learning/memory (Figure 6) is not addressed at all. It's even more puzzling in that ablation of meningeal lymphatics alone shows no behavioral deficits, TBI alone shows no behavioral deficits, but both together show deficits in motor learning/memory. This would imply that you need a really exacerbated pathology of both lymphatic inhibition and TBI to see a behavioral deficit; for me this data says the behavioral test doesn't fit the model. The authors should've done a behavioral test in which TBI alone can induce a difference, that way you can see if manipulation of meningeal lymphatics improves or worsens behavior in both directions with either inhibition or "rejuvenation" of meningeal lymphatics. There are several behavioral tests that would more accurately measure the pathology of TBI in the piriform cortex by measuring olfaction impairments. Additional experiments are not necessary but an explanation for the behavioral model selection in the discussion would be important.

We would like to thank the Reviewers for their continued support of this manuscript and for their insightful comments. We are grateful for the opportunity to continue to improve this manuscript and believe we have addressed all additional suggestions raised by the Reviewers. Below, you will find our point-by-point responses addressing remaining questions raised by the Reviewers.

Reviewer #1 (Remarks to the Author):

I thank the authors for the detailed answers, in incorporating new data and following all the points of my comments in the first submission.

Indeed, adding several post-injury times helps to understand the neuropathology associated with damage to the meninges.

Although a new time point at two months post-TBI was added, experiments suggested to inhibit the microglia have not been performed due to lab shutdown because of COVID-19. I understand that the experiments on this resubmission, including the Visudyne photoablation, were performed during/before the first submission, but had not been incorporated into the study despite this. I would strongly encourage the authors to follow up with these studies in the future. The issue of sex was partially resolved, but readers will be left a bit in the dark whether males or females were used in these studies, even if only one sex was used per experiment, which sex? And why? In summary, the authors have responded well to the reviewer's and editor's suggestions, editing the text, and incorporated new experiments and figures, although not completely responding to all of my concerns. I recommend for publication after the authors clarify the points raised above.

We are pleased that the Reviewer highlighted that we responded thoroughly to the Reviewer's and Editor's suggestions. We are also encouraged that the Reviewer believes that this work would be suitable for publication following additional clarification about the sex of the animals used in our experiments. We thank the Reviewer for highlighting the important need to better clarify the sex of the mice used in our experiments. Males and females were used for drainage and lymphangiogenesis studies, as sex has not been shown to influence lymphatic drainage at baseline (Da Mesquita et al. *Nature* 2018). Both males and females were also used to study the effects of increased ICP on lymphatic flow. Males were used for all pre-existing lymphatic dysfunction studies (Figures 4-6) for consistency with behavioral readouts and for consistency within the RNA sequencing data, as sex can influence both of these readouts. Male mice were also used for the aged mice experiments. We have added additional text to the revised manuscript in the methods section to more clearly indicate this. We also agree with the Reviewer that it will be exciting in future studies to evaluate how depletion of microglia affects the meningeal lymphatic system in TBI. We thank the Reviewer for this thoughtful suggestion and also their encouragement to follow up on this novel research direction in future studies.

Reviewer #2 (Remarks to the Author):

In revision, the authors have significantly improved this new preclinical investigation into the causes and consequences of meningeal lymphatic dysfunction in mTBI. Several open questions from the original manuscript have been addressed with new data and relevant discussion, and the long-term effects of deficits in meningeal lymphatic drainage are now more clearly defined. In particular, the revised manuscript now confirms that impairments in brain drainage after mTBI return to baseline levels by 2 months post-injury. New data on meningeal lymphatic vascular remodeling over time reveal the dynamic of lymphangiogenesis after mTBI and a return to baseline sham levels at 2 months post-injury as well. The neurological consequences of meningeal lymphatic dysfunction following mTBI have been established with increased cognitive impairments in mice that had pre-existing meningeal lymphatic impairments (induced by visudyne) prior to mTBI. The additional analysis in aged mTBI mice increases the translational relevance of these preclinical studies, and a rescue experiment with VEGF-C in aged

mice reduces microglial activation when meningeal lymphatics are functionally restored. This intervention study provides proof-of-concept support for therapeutic targeting of meningeal lymphatic system to improve outcomes in TBI. Overall, these novel studies by Bolte and colleagues advance basic understanding about the contribution of meningeal lymphatic impairments to secondary injury and recovery following mTBI. Congratulations to the authors on completing very thorough and very interesting preclinical TBI studies.

David Loane

We thank Dr. Loane for his support of this manuscript and for his comments that the manuscript is “significantly improved” and that our studies are “novel”, “very thorough”, and “very interesting”.

Reviewer #3 (Remarks to the Author):

The authors have comprehensively addressed all comments and suggestions recommended by the reviewers, and the study + manuscript are significantly improved as a result. Incorporation of an aged cohort and targeting of VEGF-C in particular provide novel insight into mechanisms of meningeal lymphatic drainage dysfunction.

We were pleased to find that the Reviewer felt that we had comprehensively addressed all comments and suggestions recommended by the reviewers. We agree with them that addressing these comments have led to significant improvements in this work and the manuscript.

Reviewer #4 (Remarks to the Author):

Overall, the authors made a substantial effort to reply to the concerns of the reviewers and the manuscript is significantly improved. Considering that the lab was closed due to Covid19, the progress on the manuscript data is commendable. In spite of the strengths in the rebuttal, some concerns are still valid.

We are pleased that the Reviewer highlighted that the revised work was “significantly improved”, and that they found the progress made on this project to be “commendable”. We also greatly appreciate the thoughtful feedback provided by the Reviewer on both submissions and believe that their suggestions have helped to substantially improve this work. Below we have provided a point-by-point response to address the Reviewer’s remaining critiques.

The authors make a point that at 24 hours after TBI there are in fact changes due to ablation alone, however, it is not the same changes as in the ablated + TBI group and thus they consider the effects of ablation alone negligible. This is reinforced by the claim that at 1-week post-TBI there are virtually no changes in the bulk RNA-seq data in the control groups. While this reviewer agrees with the rebuttal in making a supplementary figure with the title saying photoablation alone is sufficient to induce changes but disagrees with the text saying the 338 differentially expressed genes are “negligible” since they are different and less significant than the experimental groups: It is not reasonable to say that 338 differentially expressed genes represent a “negligible” change, especially since the data in the main figure 4 is focused on the 24-hour time-point where these changes in ablation alone can be seen by the bulk RNA-seq data. The authors do say in the title of Supplementary Figure 6 that there are indeed changes, however, they claim that they did not see the same “trend” of genes in the top 20 up and down-regulated genes between the ablated and non-ablated sham groups. The trend does in fact look similar in panel e in Supplementary Figure 6 with the ablated group showing a general increase in similar complement-related genes as the data shown in Figure 4D. It is still unclear if the top 20 genes

in supplementary figure 6 is what is shown in panel e? It's likely not, so some of the top 20 genes in supplementary figure 6B should be identified similar to Figure 4C to see if the top 20 genes are in fact different.

The authors also claim that these changes due to photo-ablation are “not as significant,” yet this cannot be seen as the scale bars in Figure 4D and Supplementary Figure 6E are not the same and/or the p-values are not shown. Additionally, even if the photoablation control group's genes are not as significant, the question is if it is significant enough to make the actual data in Figure 4D not significant. The authors should also look at the photo-ablated SHAM versus the photo-ablated TBI to determine how much of an actual difference TBI alone induces. This may have been shown in the middle panel of Figure 4B, but it is unclear if the authors are comparing the ablated sham versus ablated TBI or unablated TBI.

Alternatively, if it is possible the authors should compare the non-ablated TBI to ablated TBI after normalizing or subtracting the differences due to photoablation alone in the sham controls. Some comments addressing these concerns would be helpful.

We agree with the Reviewer that words like “negligible” and “trend” are misleading in the context of this data and have revised the wording in the text accordingly. We would also like to clarify that in the previous Figure 4B, we showed both the Not Ablated + Sham vs. Not Ablated + TBI comparison and the Ablated + Sham vs. Ablated + TBI comparison (left to right). Thanks to the Reviewer we also now realize that the wording above the volcano plots was confusing and we apologize for that. As a result, we have re-labeled the volcano plots and figure panels so that now 4b denotes the Not Ablated + Sham vs. Not Ablated + TBI comparison, 4c denotes the Ablated + Sham vs. Ablated + TBI comparison, and 4d denotes the Not Ablated + TBI vs. Ablated + TBI comparison.

We agree with the Reviewer that based on the presentation of our data shown in the previous submission that we could not conclude that the changes due to photoablation alone are “not as significant” when compared to the Ablation + TBI group. Therefore, we have provided a table that includes the negative log of the adjusted p value for the top 20 up- and downregulated genes for the Not Ablated + TBI vs. Ablated + TBI group as shown in revised Figure 4e (Figure 4d in the previous submission), but for both requested comparisons (Not Ablated + Sham vs. Ablated + Sham comparison and Not Ablated + TBI vs. Ablated + TBI comparison) (please see Reviewer Table below). With this presentation, the significance of the adjusted p-values can now be more easily compared between the Ablated + TBI and Ablated alone groups. Upon re-examination of Arc, which we discussed in the manuscript, it is apparent that this gene is multiple orders of magnitude more significantly downregulated in the Not Ablated + TBI vs. Ablated + TBI comparison than in the Not Ablated + Sham vs. Ablated + Sham comparison. Moreover, when we look at some of the complement genes that were upregulated in the Ablation + TBI group (C1qc, C4b, Ctsh, Irf7, Itgam), it can now be determined that they are at least an order of magnitude more significantly upregulated in the Ablated + TBI group than in the Ablation alone comparison. There are some genes that were equally significantly upregulated with ablation alone including H19 and genes related to collagen synthesis (Col1a1, Col1a2, Col6a2), indicating that the ablation procedure results in some shared signatures, such as collagen production. We have included this new table in the manuscript to provide better clarity to the readers on which genes were more significantly changed in the Ablation + TBI group as compared to Ablation alone group. Moreover, we have revised the text in the results section to discuss these considerations.

Finally, as the Reviewer pointed out, the scale bars on the heatmaps are not helpful when attempting to compare the heatmap in Figure 4 (now Figure 4e) with Supplementary Figure 6e, as these scale bars are automatically generated to fit the data based on the directed comparisons. Since the new Table 1 provides an easier method of interpretation for the same genes as featured in Supplementary Figure 6e, we have removed Supplementary Figure 6e from the manuscript. We appreciate the Reviewer for highlighting that our conclusions regarding significance were not directly supported with

the presented data, and believe that the addition of this new quantitative table will allow for direct comparison and now offers better support for our original claims.

Gene	Not Ablated + TBI vs. Ablated + TBI (-log2padj)	Not Ablated + Sham vs. Ablated + Sham (-log2padj)	Fold Change of comparisons (Column 2 / Column 3)
X9630013A20Rik	26.970626	2.815854066	9.578133443
H19	21.1141876	22.06175671	0.957049245
Col1a1	15.43023984	30.38196062	0.507875052
Col1a2	10.75533593	30.98605305	0.347102482
C1qc	10.32893875	2.391067707	4.319801869
Mpeg1	6.757430717	1.756788511	3.846467959
C4b	5.984917191	4.616960161	1.296289546
Col6a1	5.984917191	3.999476138	1.496425278
Pld4	5.940281437	6.187779078	0.960002185
Tcirg1	5.456945942	4.717892655	1.156649026
Bgn	5.360353995	10.40616731	0.515113186
Ctsh	5.14985031	3.157904469	1.630780906
Itgam	5.14985031	3.936637228	1.308185137
Gjc2	4.938855873	1.674230563	2.949925765
Ltbp4	4.938855873	1.657847813	2.979076749
Trem2	4.841619029	3.181598672	1.521756679
Col6a2	4.806953128	5.819750401	0.825972387
Fcrls	4.806953128	1.824303203	2.634952962
Irf7	4.806953128	2.971062138	1.617924131
Sema6a	4.806953128	0.991155235	4.849848902
Arc	13.03848763	0.227624649	57.28064907
Gng4	4.797345775	0.494613486	9.69918109
Hspa5	3.798782603	4.253261616	0.893145766
Gm43980	3.79364208	0.058467469	64.88466372
Pcsk1	3.79364208	2.981308322	1.272475595
Mras	3.645202498	2.149743978	1.695644939
Etv5	3.490604725	10.8326543	0.322229864
Zbtb7a	3.490604725	3.151038462	1.107763287
Rgs8	3.238182681	0.402979251	8.035606483
Serpinb8	3.187780695	0.046380808	68.73059805
Zfp324	3.168039956	0.104637618	30.27630047
Gfod1	3.153122178	7.999403905	0.394169643
Pdzd2	3.143305133	0.660051173	4.7622143
Spry4	3.06569035	1.54188296	1.988276951
Hyou1	3.052478864	0.436498851	6.993097131
Slc6a3	3.006420929	0.14153804	21.24108066
Cbln4	2.944778846	0.303388055	9.706311115
Unc13c	2.875692664	11.36028247	0.253135666
Mex3c	2.811678116	0.494613486	5.684596546
Ppp1r10	2.590577645	0.032459754	79.80891294
Mean	6.010426967	4.830226859	1.244336372
Median	4.806953128	2.603460886	1.707103073

Activated	
0-1.99	
2-3.99	
4-5.99	
6-9.99	
10+	
Repressed	
0-1.99	
2-3.99	
4-5.99	
6-9.99	
10+	

Another issue is that it is possible that differences in astrocytes are masked by other cell types in the bulk RNA-seq such as microglia or neurons, or by cells in the ipsilateral side of the brain; in fact, Figure 6D looks like ablation alone had an effect on GFAP area similar to TBI alone but not so much in microglia. Comment regarding this limitation would be useful.

We agree with the Reviewer that there are limitations with bulk RNA sequencing, and we may be missing more nuanced changes in individual cell populations. We hope to follow up in the future with

other techniques that will show these changes, such as single-cell RNA sequencing. We have added text to the discussion section of the revised manuscript to describe these limitations.

Visudyne specificity for lymphatic endothelial ablation: The authors address the previous concern regarding CD31 expression by doing percent area of CD31+ Lyve1- vessels in a supplementary figure, but this is not sufficient in addressing the reviewer's concerns. To count CD31+ blood vessels across the entire meninges even as a percent area is misleading when photoablation only occurred in a specific region. Any differences in CD31+ blood vessels in the area affected by photoablation would be masked by the unaffected CD31+ blood vessels in the rest of the meninges (the authors even state this in their rebuttal). The authors should analyze the percent area of CD31+ Lyve1- vessels in specifically the photo-ablated area to ensure that Visudyne is specific for lymphatic endothelial cells and not blood endothelial cells. The authors also say that the lymphatic vessels in photo-ablated area strongly label for CD31 even with photoablation; this data should be shown in the supplementary and discussed as it's contradictory to their conclusion that photoablation doesn't affect blood vessels. If this is true, then the authors cannot distinguish a photoablated lymphatic vessel from a blood-vessel and the conclusion that photoablation doesn't affect blood-vessels would be difficult to make.

Two citations are provided describing the specificity of Visudyne for LECs, however, the first citation shows photoablation also induces smooth muscle cell necrosis, and the second citation uses the same analysis as here with presumably the same issue. An alternative is to stain for other lymphatic markers along with CD31; it's possible that photoablation reduces Lyve-1 expression but like CD31 does not reduce the expression of VEGFR3, Podoplanin, and/or Prox1. Some explanation regarding Visudyne specificity for LECs would be important.

We thank the Reviewer for bringing forward these important considerations. We agree with the Reviewer that the CD31+ blood vessels surrounding the meningeal lymphatics are more likely to be affected by the Visudyne ablation approach than those further away and that our previous analysis approach could have missed changes to the blood vessels surrounding the meningeal lymphatics. As suggested by the Reviewer, we have analyzed the percent area of CD31+Lyve-1- vessels specifically in the regions surrounding the areas of photoablation and have included this data, as well as the total CD31+Lyve-1- data in Supplementary Figure 5 (See Reviewer Figure below, panel e). This new analysis demonstrates that our Visudyne ablation approach does not significantly impact the percent area covered by CD31+Lyve-1- cells in areas that are in close proximity to the specific regions targeted with photoablation. Instead, we only observe significant ablation of Lyve-1+ cells in these photoablated regions. In further support of our findings, other studies show that this photoablation procedure has no impact on blood flow or blood oxygenation in this area four days after treatment when measured by photoacoustic imaging (Louveau et al., Nat Neuroscience, 2018).

We would also like to clarify that we did not mean to indicate that lymphatic vessels in the photo-ablated area strongly label for CD31, because we do not see lymphatic vessels in the region of photoablation. Instead, we hoped to convey that the area where the meningeal lymphatic vessels lie (along the transverse and superior sagittal sinuses) strongly labels for CD31 since there are large blood vessels in this region. We do not, observe the expression of CD31 in areas previously labeled by Lyve-1 following Visudyne-induced photoablation. We apologize for any confusion that our wording may have caused in the previous submission. For the resubmission we have rewritten this section to ensure better clarity.

Regarding photoablation specificity for lymphatic endothelial cells, previous studies have shown that the Visudyne photoablation procedure results in the loss of Lyve-1, Prox-1, and Podoplanin expression in the region of Visudyne-induced photoablation, indicating that other markers of lymphatic endothelial cells are also being disrupted along with Lyve-1 (Tammela et al., Sci Trans Medicine, 2011, Louveau et al., Nat Neuroscience, 2018). Collectively, these findings suggest that intracisterna magna injection of Visudyne followed by photoablation selectively ablates the meningeal lymphatic vessels and does

not appreciably ablate the blood vasculature surrounding this region. We appreciate the Reviewer for bringing these considerations to light, and have included additional text in the results section to better describe and support this approach.

Is there lymphangiogenesis in both the contralateral and ipsilateral sides of the TBI? This should be clarified.

Thanks for pointing out that this was unclear. The lymphangiogenesis was quantified for the entire lymphatic vasculature overlying both hemispheres. This is now more clearly stated in the methods section.

The choice of behavioral tests needs additional explanation. The author's model of TBI focuses on the piriform cortex, which is classically associated with the sense of smell and not motor coordination or learning/memory. How ablation of meningeal lymphatics coupled with TBI in the piriform cortex affects motor learning/memory (Figure 6) is not addressed at all. It's even more puzzling in that ablation of meningeal lymphatics alone shows no behavioral deficits, TBI alone shows no behavioral deficits, but both together show deficits in motor learning/memory. This would imply that you need a really exacerbated pathology of both lymphatic inhibition and TBI to see a behavioral deficit; for me this data says the behavioral test doesn't fit the model. The authors should've done a behavioral test in which TBI alone can induce a difference, that way you can see if manipulation of meningeal lymphatics improves or worsens behavior in both directions with either inhibition or "rejuvenation" of meningeal lymphatics. There are several behavioral tests that would more accurately measure the pathology of TBI in the piriform cortex by measuring olfaction impairments. Additional experiments are not necessary but an explanation for the behavioral model selection in the discussion would be important.

We appreciate the Reviewer's questions regarding the behavioral tests. While the injury in our experimental paradigm is directed at the piriform cortex, injury to one region of the brain has been shown to have widespread and even systemic effects (Cai et al., Nat Neuroscience, 2019). Previous studies have shown that there are changes in neurons even in the contralateral hemisphere years after injury, indicating that a regional hit can impact the entire brain (Erturk., J Neuroscience, 2016). Additionally, we have shown in our studies that this TBI model does affect the contralateral region of the brain (Figure 7), so although the injury is directed, the effects can be widespread. Furthermore, injuries to this specific region of the brain in other more severe models of TBI have been shown to cause disruptions in cognitive function (Ren et al., J Cereb Blood Flow Metab, 2013).

For our behavior studies, we were particularly interested in investigating how combining pre-existing meningeal lymphatic impairments with TBI affects neurological function. We decided on a mild form of TBI that does not by itself cause obvious behavioral abnormalities to see if pre-existing meningeal lymphatic dysfunction can synergize with TBI to cause exacerbated behavioral defects. Likewise, selecting behavioral tasks that are not appreciably impacted by pharmacological disruption of the meningeal lymphatics alone further helps to illustrate how coupling pre-existing meningeal lymphatic dysfunction with TBI can lead to deleterious outcomes. We thank the Reviewer for highlighting that we have not effectively articulated this in the text. As suggested by the Reviewer we have added text to the results section of the revised manuscript to better explain the selection of behavioral tests in our studies.

Reviewers' Comments:

Reviewer #4:

Remarks to the Author:

All comments have been addressed. This is an important contribution to the field.